# The nuclear factor ID3 endows macrophages with a potent anti-tumour activity

Zihou Deng[1], Pierre-Louis Loyher[1], Tomi Lazarov[1,2], Li Li[3], Zeyang Shen[4,5], Bhavneet Bhinder[6], Hairu Yang[1], Yi Zhong[7], Araitz Alberdi[1], Joan Massague[8], Joseph C. Sun[1], Robert Benezra[8], Christopher K. Glass[4], Olivier Elemento[6], Christine A. Iacobuzio-Donahue[7] & Frederic Geissmann[1,2✉]

Macrophage activation is controlled by a balance between activating and inhibitory receptors[1–7], which protect normal tissues from excessive damage during infection[8,9] but promote tumour growth and metastasis in cancer[7,10]. Here we report that the Kupffer cell lineage-determining factor *ID3* controls this balance and selectively endows Kupffer cells with the ability to phagocytose live tumour cells and orchestrate the recruitment, proliferation and activation of natural killer and CD8 T lymphoid effector cells in the liver to restrict the growth of a variety of tumours. ID3 shifts the macrophage inhibitory/activating receptor balance to promote the phagocytic and lymphoid response, at least in part by buffering the binding of the transcription factors ELK1 and E2A at the *SIRPA* locus. Furthermore, loss- and gain-of-function experiments demonstrate that ID3 is sufficient to confer this potent anti-tumour activity to mouse bone-marrow-derived macrophages and human induced pluripotent stem-cell-derived macrophages. Expression of ID3 is therefore necessary and sufficient to endow macrophages with the ability to form an efficient anti-tumour niche, which could be harnessed for cell therapy in cancer.

Molecular understanding of mechanisms that control the growth of tumour cells within target tissues helps in the identification of therapeutic targets and strategies[7,10–14]. Macrophages are an important component of these niches[15–17], and can recognize, bind to and phagocytose tumour cells[2,17–19], but frequently fail to do so and can even support tumour growth and dissemination[15,16].

Macrophage activation is tightly controlled by a balance of activating and inhibitory receptors, which protect normal tissues[4,6,8,9,20] but allow tumoural cells to escape[7,10,14]. Binding of the tyrosine-based inhibitory motif (ITIM)-containing inhibitory receptors signal regulatory protein-α (SIRPA)[3] or sialic-acid-binding Ig-like lectin 10 (SIGLEC10)[7] to their respective ligands CD47 and CD24 on tumour cells prevents activating receptors such as dectin-1[1], dectin-2[21] and the calreticulin receptor LRP1, which bind to glycoproteins on tumoural cells[2,18], from initiating macrophage activation[4,5,7]. The importance of this mechanism is underscored by the promising results of targeting the SIRPA–CD47 axis for cancer treatment[10,22,23].

Resident macrophages from different tissues express sets of lineage-determining factors (LDFs) that are important for their embryonic development and the establishment and maintenance of tissue-specific transcriptional programs[24–26]. This suggests that the expression of LDFs may endow macrophages from distinct anatomical niches with specific functions. In the context of cancer, tissue-specific LDF expression by macrophages could therefore contribute to variations in the local resistance to tumour growth. The liver filters venous blood from the gastrointestinal tract, carrying microbial products[27] and metastatic cells from colorectal and pancreatic cancer[28], and is therefore a major site for tumour haematogenous dissemination. Notably, a number of studies suggest that, in contrast to other organs, macrophages in the liver represent a robust innate immune barrier against tumour progression[17,21,29], yet the underlying mechanisms remain poorly understood. The identification of such mechanisms is of general interest to both basic and clinical tumour immunology, as they could be harnessed for the purpose of cellular therapies in cancer.

Kupffer cells (KCs), the resident macrophages of the liver, are highly phagocytic and are a good candidate to mediate resistance to metastasis[17,21,29]. Here we took advantage of genetic tools for the selective targeting of KCs and of human induced pluripotent stem (hiPS) cell macrophages to investigate the role of KCs and the KC-specific LDF ID3 in cancer. We report that ID3 expression by KCs endows them with the ability to orchestrate a potent anti-tumour response by establishing a peritumoural phagocytic and activated lymphoid effector niche. Furthermore, we show that ectopic expression of ID3 in mouse bone-marrow-derived macrophages (BMDMs) and human hiPS cell-derived macrophages (hiPSC-Macs) is sufficient to endow them with the ability to orchestrate this vigorous phagocytic and lymphoid anti-tumoural activity in a variety of tumour models in vitro and in vivo. Mechanistically, we demonstrate that ID3 shifts the macrophage inhibitory/activating receptor balance at least in part by buffering the binding of the transcription factors ELK1 and E2A at the

[1]Immunology Program, Sloan Kettering Institute, Memorial Sloan Kettering Cancer Center, New York, NY, USA. [2]Weill Cornell Graduate School of Medical Sciences, New York, NY, USA. [3]Graduate Center, City University of New York, New York, NY, USA. [4]Department of Bioengineering, University of California, San Diego, La Jolla, CA, USA. [5]Department of Cellular and Molecular Medicine, University of California, San Diego, La Jolla, CA, USA. [6]Department of Physiology and Biophysics, Institute for Computational Biomedicine, Weill Cornell, New York, NY, USA. [7]Department of Pathology, Memorial Sloan Kettering Cancer Center, New York, NY, USA. [8]Cancer Biology and Genetics Program, Sloan Kettering Institute, Memorial Sloan Kettering Cancer Center, New York, NY, USA. ✉e-mail: geissmaf@mskcc.org

*Sirpa* locus under steady-state and inflammatory conditions, lowering SIRPA expression and, therefore, enabling the formation of a potent anti-tumour immune niche.

## KCs restrict tumour growth

Depletion of macrophages in C57BL/6 mice with the CSF1R inhibitor PLX5622 increased liver engraftment of the pancreatic adenocarcinoma cell lines KPC-1 (*P48^cre*, *Kras^LSL-G12D*, *Trp53^LSL-R172H*), Pan02, the melanoma cell line B16F10 and Lewis lung carcinoma LLC1 in comparison to control untreated mice (Extended Data Fig. 1a,b), consistent with the proposed anti-tumour role of liver macrophages[17,19,21,29]. We therefore performed an analysis in genetic models of macrophage-deficient mice of the roles of liver macrophage subsets in long-term syngeneic pancreatic adeno-carcinoma models. After 8 weeks, littermate control mice developed large pancreatic and splenic tumours, and around half of the mice developed detectable liver and lung metastases (59 ± 3% and 52 ± 7%) (Fig. 1a–e and Supplementary Data 1). The same tumoural phenotype was observed in *Flt3^cre Csf1r^f/f* (Fig. 1a) and *Ccr2^-/-* mice (Fig. 1b), which carry a normal number of TIM4⁺ KCs (Extended Data Fig. 1c,d) but are deficient in CSF1R-dependent BMDMs and monocyte-derived mac-rophages, respectively[30,31].

By contrast, specific targeting of KCs (Extended Data Fig. 1e–g) in *Clec4f^cre Csf1r^f/f* (Fig. 1c and Extended Data Fig. 1h) or *Clec4f^cre Spi1^f/f* (Fig. 1d and Extended Data Fig. 1i) mice, or the inducible depletion of KCs after diphtheria toxin (DT) administration to tumour-bearing *Clec4f^cre R26^LSL-DTR* mice[25] (Fig. 1e and Extended Data Fig. 1j,k) resulted in the development of larger liver, lung and (less reproducibly) spleen metastases in all mice (100% (liver and lung) and 95% ± 7 (spleen)), while the size of pancreatic tumours was unchanged in comparison to *cre*-negative littermate controls (Fig. 1c–e and Supplementary Data 1). As lung macrophages do not express *Clec4f* (Extended Data Fig. 1f) and are not depleted in *Clec4f^cre R26^LSL-DTR* mice treated with DT (Extended Data Fig. 1j), the increase in lung metastases in KC-deficient mice is probably a consequence of the higher tumour burden in the liver, as the hepatic veins drain into the lung through the right ventricle of the heart (Fig. 1f). Moreover, survival experiments after intraportal injection of KPC cells showed that *Clec4f^cre R26^LSL-DTR* mice treated with DT have a reduced survival in comparison to *cre*-negative littermate controls (Fig. 1g). KC depletion in *Clec4f^cre R26^LSL-DTR* mice also increased the number of tumour cells present in the liver 24 h after portal-vein injection of KPC cells by more than threefold (Fig. 1h). Moreover, KC depletion before, as well as 3 days after, tumour injection increased the tumour burden in the liver after 2 weeks by greater than fivefold (Fig. 1i,j). Finally, flow cytometry analysis showed that a subset of tumour cells coexpressing CD47^bright and markers previously associ-ated with metastatic potential such as CD9 and CD133[32,33] (Extended Data Fig. 1l) and endowed with metastatic potential in vivo and in vitro (Extended Data Fig. 1m) is increased by around tenfold in the liver of KC-deficient *Clec4f^cre R26^LSL-DTR* mice in comparison to in their littermate controls (Extended Data Fig. 1n). These data indicate that KCs, in con-trast to BMDMs, represent a potent barrier to the liver engraftment of tumour cells circulating in the portal vein, and exert a strong and long-lasting inhibitory effect on their subsequent growth in the liver and the lungs.

## KCs nucleate a peritumoural niche

TIM4⁺CLEC4F⁺ KCs were always located outside and around the liver tumour nodules in an endogenous tumour model with spontaneous metastases (KPC mice) (Fig. 2a), as well as in the orthotopic graft model (Extended Data Fig. 2a), and in short-term models after intraportal injection of five different carcinoma and melanoma cell lines (Extended Data Fig. 2b–g). An analysis of CD45.1–CD45.2 parabionts in which the CD45.2 partner received intraportal injection of KPC cells confirmed

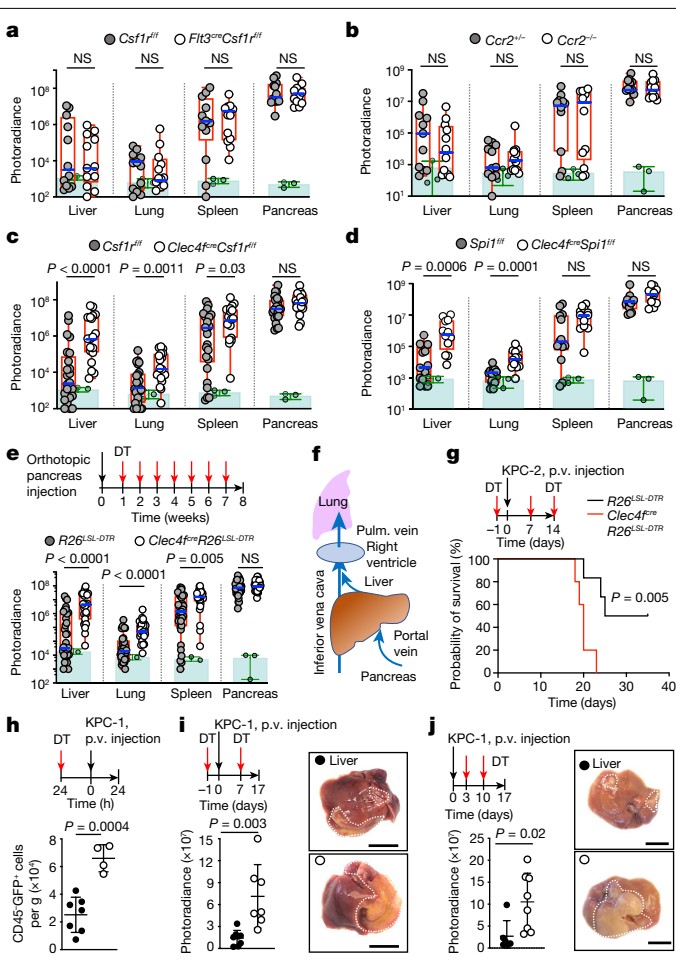

**Fig. 1 | KCs restrict tumour cell engraftment and metastasis.**
**a**–**e**, Bioluminescence analysis of the tumour burden in the liver, lungs, spleen and pancreas of the indicated mice 8 weeks after orthotopic pancreas injection of 2 × 10⁵ KPC-2-luciferase (KPC-2-luci) cells: *Flt3^cre Csf1r^f/f* mice (*n* = 12) and control *Csf1r^f/f* littermates (*n* = 12) (**a**); *Ccr2^-/-* mice (*n* = 12) and *Ccr2^+/-* littermates (*n* = 11) (**b**); *Clec4f^cre Csf1r^f/f* mice (*n* = 19) and *Csf1r^f/f* littermates (*n* = 25) (**c**); and *Clec4f^cre Spi1^f/f* mice (*n* = 12) and *Spi1^f/f* littermates (*n* = 13) (**d**). *Clec4f^cre R26^LSL-DTR* mice (*n* = 26) and *R26^LSL-DTR* littermates (*n* = 33) (**e**) received weekly intraperitoneal injection of DT (Methods) from week 1 to 7. The circles represent individual mice, boxes represent the 25–75% confidence intervals and the whiskers indicate the extreme values. The blue lines indicate the median. The green histograms represent the background bioluminescence imaging signal from wild-type C57BL/6J mice that did not receive tumours (*n* = 3 mice per group). Results are from at least three independent experiments per genotype. Statistical analysis was performed using two-tailed Mann–Whitney *U*-tests; *P* < 0.05 was considered to be significant. **f**, Schematic of pancreas venous drainage. Pulm., pulmonary. **g**, *Clec4f^cre R26^LSL-DTR* mice (*n* = 5) and *R26^LSL-DTR* littermates (*n* = 6) received intraportal injection of 3 × 10⁵ KPC-2-luci cells (day 0 (D0)) and DT injections (D−1, D7 and D14). Survival was analysed using log-rank (Mantel–Cox) tests. p.v., portal vein. **h**, *Clec4f^cre R26^LSL-DTR* (*n* = 7) and *R26^LSL-DTR* control (*n* = 4) mice received DT injection 24 h before intraportal injection of 1 × 10⁶ KPC-1-luci-GFP cells. The numbers of CD45⁻GFP⁺ tumour cells per g of liver were analysed 24 h later using flow cytometry. **i**, *Clec4f^cre R26^LSL-DTR* mice (*n* = 7) and *R26^LSL-DTR* littermates (*n* = 8) received 1 × 10⁶ KPC-1-luci cells (D0) and DT injections (D−1 and D7), and the livers were analysed at D14 by bioluminescence imaging. Representative liver micrographs are shown on the right. **j**, Bioluminescence imaging analysis as described in **i**, with DT injections at D3 and D10 (*n* = 8 and 6). The circles represent individual mice. Data are mean ± s.d. Statistical analysis was performed using two-tailed unpaired *t*-tests; *P* < 0.05 was considered to be significant. For **i** and **j**, scale bars, 1 cm.

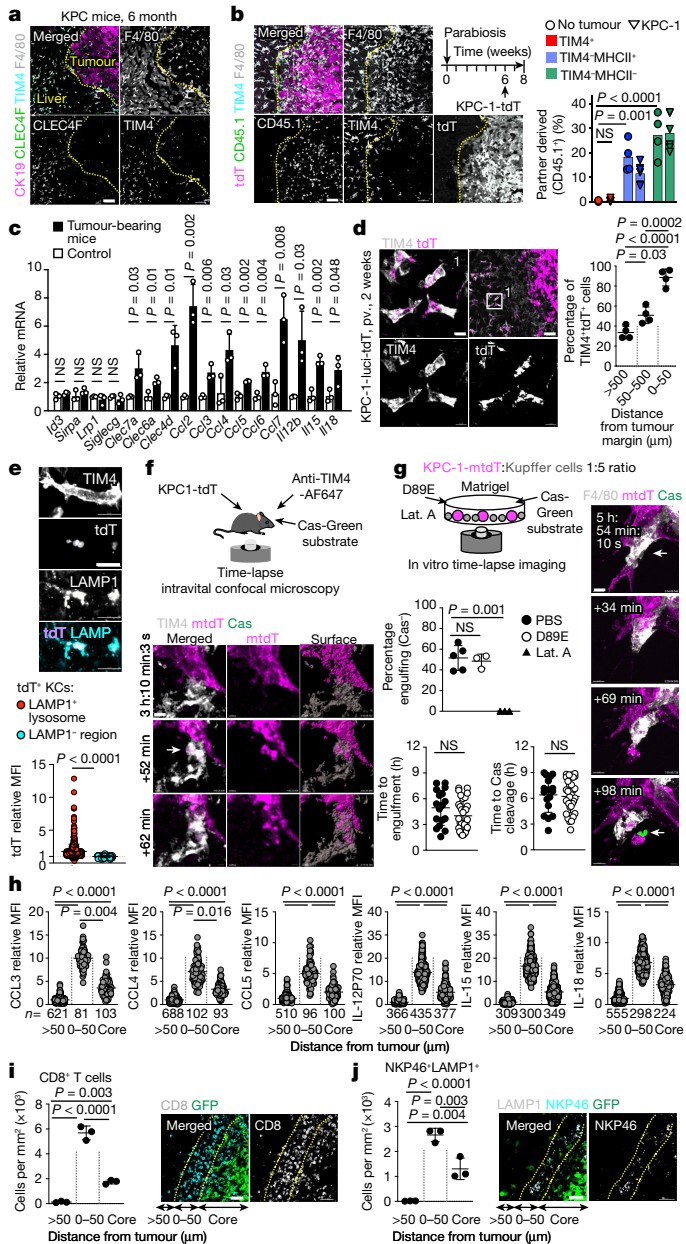

**Fig. 2 | Peritumoural niche. a**, Immunofluorescence staining of F4/80, TIM4, CLEC4F and CK19 on liver sections from 6-month-old KPC mice. $n = 3$ independent experiments. **b**, Immunofluorescence (left) and flow cytometry (right) analysis of CD45.1$^+$ macrophages in the livers of the CD45.2 partners from CD45.1–CD45.2 parabiotic pairs, 2 weeks after intraportal injection of $1 \times 10^6$ KPC-1-tdT cells in the CD45.2 partner ($n = 5$), or without tumour injection ($n = 4$). For **a** and **b**, scale bars, 50 µm. **c**, Analysis using quantitative PCR with reverse transcription (RT–qPCR) of the indicated genes in KCs, 2 weeks after intraportal injection of $1 \times 10^6$ KPC-1-tdT cells or in control mice. $n = 3$ mice per group. **d**, Immunofluorescence staining and the percentage of TIM4$^+$ KCs containing tdT in mice from **c**. $n = 4$. Scale bars, 10 µm (left) and 50 µm (right). **e**, tdT expression in LAMP1$^+$ ($n = 713$) and LAMP1$^-$ ($n = 237$) areas in KCs from **c**. For **e**–**g**, scale bars, 10 µm. **f**, Engulfment of KPC-1-tdT cells by KCs in vivo was analysed using intravital imaging. KCs and dying cells were labelled by i.v. injection of F4/80-AF647 antibodies and Cas-Green, respectively. The arrow indicates engulfing KCs. $n = 3$. **g**, In vitro analysis of KPC-1 engulfment by KCs in the presence of PBS control ($n = 5$ experiments), phosphatidylserine blockade (MFG-E8(D89E); $n = 3$) or actin inhibitor (latrunculin A (lat. A); $n = 3$). KCs and dying cells are labelled as in **f**. The open arrow shows engulfing KCs. The closed arrow shows caspase-3/7 cleavage. The plots show the percentage of KCs engulfing Cas-Green$^-$ KPC-1-tdT cells, and the time from stable interaction between individual KCs ($n = 17$ (PBS) and $n = 31$ (MFG-E8(D89E))) and tumour cells to engulfment and caspase-3/7 cleavage. **h**, Expression of chemokines and cytokines by TIM4$^+$ KCs ($n$ values are shown) in tumour core and peritumoural liver (0–50 µm from tumour and >50 µm from tumour) 2 weeks after injection of $1 \times 10^6$ KPC-1-GFP cells. **i,j**, representative staining and the number of CD8$^+$ T cells (**i**) and LAMP1$^+$NKP46$^+$ cells (**j**) in the liver from **h**. $n = 3$ mice. For **i** and **j**, scale bars, 50 µm. Statistical analysis was performed using one-way analysis of variance (ANOVA) (**b**, **d**, **g**, **i** and **j**), two-tailed Mann–Whitney $U$-tests (**e**), Kruskal–Wallis tests (**h**) and unpaired two-tailed $t$-tests (**c** and **g**). Data are mean ± s.d. NS, not significant.

the increased expression of several receptors that are involved in macrophage activation and phagocytosis such as the activating receptors dectins that recognize carbohydrate antigens on tumour cells[37], and C–C chemokines CCL2, CCL3, CCL4, CCL5, CCL6 and CCL7, and interleukins IL-12, IL-15 and IL-18, which are involved in the recruitment and activation of effector lymphoid cells at tumour sites[38–42] (Fig. 2c and Extended Data Fig. 3b,c). Moreover, the most differentially expressed mRNA in KCs from tumour-bearing mice included epithelial mRNA such as cytokeratins 8 (*Krt8*) and 19 (*Krt19*) (Extended Data Fig. 3b), which is compatible with the phagocytosis of KPC tumour cells, although a contamination cannot be excluded. The transcriptional response to tumour cells of the main and minor KCs was similar (Extended Data Fig. 3d). These data indicated that resident KCs surround the tumours and suggested several mechanisms for the KC-mediated restriction of tumour growth.

In favour of phagocytosis of tumour cells, KCs contained abundant tumour-derived material in short-term (Fig. 2d and Extended Data Fig. 3e) and long-term (Extended Data Fig. 3f) orthotopic models and the endogenous KPC tumour model with spontaneous metastases (Extended Data Fig. 3g), as visualized by tdTomato (tdT) (Fig. 2d and Extended Data Fig. 3f) or KRT19 staining (Extended Data Fig. 3e,g). The percentage of KCs containing tumour material increased over time from 40% to around 100% over 2 months in the orthotopic model (Extended Data Fig. 3f), and was around 100% in the endogenous model (Extended Data Fig. 3g). Spatially, in the short-term models, around 90% of KCs contained tumour material at the tumour margin, while only approximately 60% and 30% did between 50 to 500 µm and more than 500 µm away from the tumour margin, respectively (Fig. 2d), and tumour material (tdT) in KCs was colocalized with LAMP1$^+$ phagolysosomes (Fig. 2e). Twenty-four hours after intraportal injection of KPC cells, around 35% of liver KCs contained tumour material as assessed by flow cytometry, independently of phosphatidylserine blockade with MFG-E8(D89E) (Extended Data Fig. 3h). Intravital microscopy in the liver of wild-type mice using the CellEvent caspase-3/7 cleavage reporter (Cas-Green) to

the location of TIM4$^+$ KCs around the metastatic nodules (Fig. 2b). Although parabiosis experiments underestimate the contribution of blood-circulating cells to tissues, our results also showed that the contribution of partner-derived cells to TIM4$^+$ KCs was below 0.5%, compared with around a 25% contribution of partner-derived cells to CD45$^+$TIM4$^-$ cells (Fig. 2b and Extended Data Fig. 2h–j), suggesting that up to 99% TIM4$^+$ KCs remain host derived (CD45.2$^+$) in the tumour-bearing liver, whether they belong to the main KC CD206$^+$ subset or the smaller CD206$^{bright}$ subset[34–36] (Extended Data Fig. 2h–j). By contrast, partner-derived F4/80$^+$TIM4$^-$ macrophages accumulated within the metastatic nodules (Fig. 2b). Furthermore, genetic labelling of bone-marrow-derived cells from tumour-free and tumour-bearing mice using three genetic models (*Cx3cr1*$^{gfp}$ mice, *Cxcr4*$^{gfp}$ mice and *Cxcr4*$^{creERT2}$*R26*$^{LSL-tdT}$ mice pulsed with 4-hydroxytamoxifen at 6 weeks of age), confirmed that most TIM4$^+$ cells (KCs) from both CD206$^+$ and CD206$^{bright}$ subsets are not labelled (Extended Data Fig. 2k–o).

RNA-sequencing (RNA-seq) analysis of sorted KCs from tumour-bearing liver in comparison to the control showed macrophage activation, an inflammatory response and a cellular chemotaxis profile (Extended Data Fig. 3a and Supplementary Data 2). This profile included

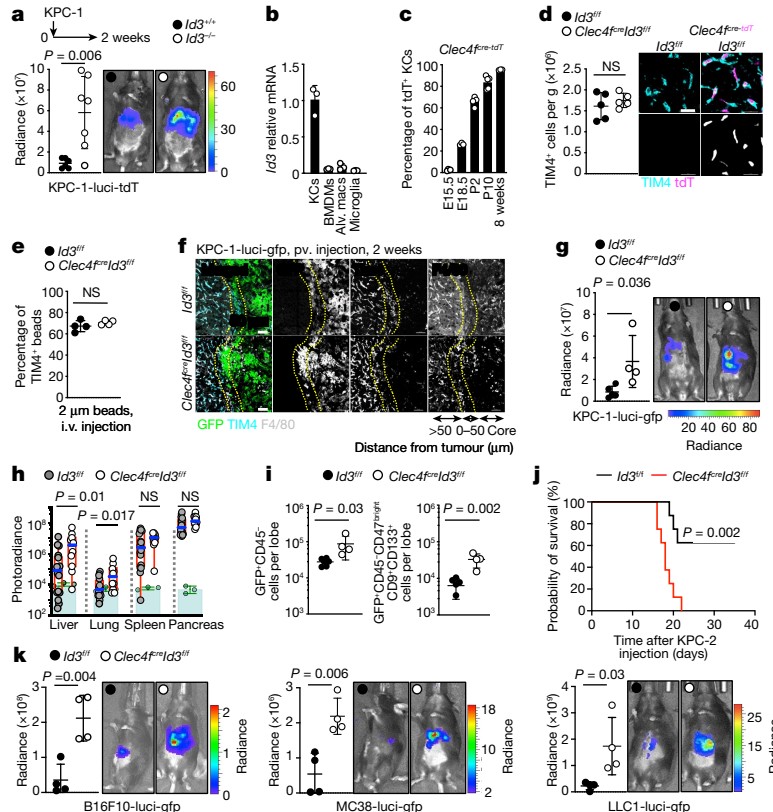

**Fig. 3 | ID3-expressing KCs are required for the restriction of tumour growth.**
**a**, Bioluminescence analysis of the tumour burden in livers from *Id3*[−/−] mice
(*n* = 7) and *Id3*[+/+] littermates (*n* = 6) 2 weeks after intraportal injection of 1 × 10[6]
KPC-1-luci cells. Results are from two independent experiments. **b**, RT–qPCR
analysis of *Id3* mRNA in macrophage populations from three C57BL/6J mice.
Alv. macs, alveolar macrophages. **c**, Flow cytometry analysis of tdT expression
by TIM4[+] KCs from *Clec4f*[cre-tdT] mice at embryonic day 15.5 (E15.5; *n* = 4), E18.5
(*n* = 3), postnatal day 2 (P2; *n* = 5) and P10 (*n* = 5) and at 8 weeks old (*n* = 3). **d**, Flow
cytometry and immunofluorescence analysis of KC numbers and morphology
in the livers of 6-week-old *Clec4f*[cre-tdT]*Id3*[f/f] mice and *Id3*[f/f] littermates. *n* = 5 per
group. Scale bars, 20 μm. **e**, Flow cytometry analysis of the uptake of 2 μm beads
injected i.v. 2 h before analysis of KCs from *Clec4f*[cre-tdT]*Id3*[f/f] mice and *Id3*[f/f]
littermates. *n* = 4 per group. **f**, Representative immunofluorescence staining
for GFP, TIM4 and F4/80 in the liver from *Clec4f*[cre-tdT]*Id3*[f/f] mice (*n* = 4) and *Id3*[f/f]
littermates (*n* = 5) 2 weeks after intraportal injection of 1 × 10[6] KPC-1-luci-GFP
cells. Scale bars, 50 μm. **g**, Bioluminescence analysis of the tumour burden
in the livers from the mice in **f**. **h**, Bioluminescence analysis of the tumour
burden in *Clec4f*[cre]*Id3*[f/f] mice (*n* = 10) and *Id3*[f/f] littermates (*n* = 16) 8 weeks after
orthotopic pancreas injection of 2 × 10[5] KPC-2-luci cells. **i**, The number of
GFP[+]CD45[−], GFP[+]CD45[−]CD47[bright] and CD9[+]CD133[+] tumour cells per liver lobe
from the mice in **f** and **g** was determined using flow cytometry. **j**, Survival of
*Clec4f*[cre]*Id3*[f/f] mice (*n* = 8) and *Id3*[f/f] littermates (*n* = 8) after intraportal injection
of 3 × 10[5] KPC-2-luci cells. **k**, Bioluminescence analysis of the tumour burden in
the livers from *Clec4f*[cre]*Id3*[f/f] mice (*n* = 4) and *Id3*[f/f] littermates (*n* = 4) 2 weeks after
intraportal injection of 5 × 10[5] B16F10-luci cells, 1 × 10[6] MC38-luci cells or 1 × 10[6]
LLC1-luci cells. Statistical analysis was performed using unpaired two-tailed
*t*-tests (**a**, **d**, **e**, **g**, **i** and **k**), log-rank (Mantel–Cox) tests (**j**) and Mann–Whitney
*U*-tests (**h**). Data are mean ± s.d.

monitor tumour cell apoptosis and death, documented the engulfment
of live KPC tumour cells by KCs (Fig. 2f, Extended Data Fig. 4a and Sup-
plementary Video 1); however the phagocytic process spanned several
hours, which made quantification difficult. We therefore developed a
two-cell in vitro time-lapse imaging assay in which KCs and KPC cells
were cultivated together in Matrigel with the Cas-Green reporter, and
with PBS control, MFG-E8(D89E) or the inhibitor of actin polymeriza-
tion latrunculin A (Fig. 2g, Extended Data Fig. 4b and Supplementary
Video 2). The results showed that around 50% of wild-type KCs actively
engulf at least 1 Cas-green[−] KPC cell in the course of a 20 h observation,
independently of phosphatidylserine blockade, whereas latrunculin
A blocked this process (Fig. 2g). Moreover, caspase-3/7 cleavage in
tumour cells followed rather than preceded engulfment by KCs (Fig. 2g,
Extended Data Fig. 4b and Supplementary Video 2). The time from
contact between KCs and tumour cells to engulfment was around 4 h
on average, while the time from contact between KCs and tumour cells
to caspase-3/7 cleavage was around 6 h in this assay (Fig. 2g).

Immunofluorescence staining in tumour-bearing liver from KPC
mice and littermate controls confirmed that the CCR5 ligands CCL3,
CCL4 and CCL5, and the cytokines IL-12, IL-15 and IL-18 were pro-
duced by KCs in peritumoural liver (Extended Data Fig. 5a). Similarly,

immunofluorescence staining in tumoural liver 2 weeks after intrapor-
tal injection of KPC cells in wild-type mice indicated that CCL3, CCL4
and CCL5, and IL-12, IL-15 and IL-18 are most prominently produced by
KCs present at the tumour margin (Fig. 2h and Extended Data Fig. 5b).
Consistently, quantification of activated natural killer (NK) cells and
CD8[+] T cells in the metastatic liver showed that they were also prefer-
entially enriched at the tumour margin, next to KCs expressing CCL3,
CCL4, CCL5, IL-12, IL-15 and IL-18 (Fig. 2i,j). Together, these data sug-
gested that resident KCs that surround the tumour cells may exert their
anti-tumour activity through sustained phagocytosis of live tumour
cells and recruitment and activation of lymphoid effectors cells.

## KC anti-tumour activity is ID3 dependent

IDs[43] are early genes that regulate cell fate determination during devel-
opment and cellular functions in differentiated cells[44]. ID3 was shown to
be a KC lineage-determining nuclear factor because embryonic premac-
rophages lacking ID3 expression do not differentiate into KCs during
organogenesis, resulting in a selective KC deficiency[24] (Extended Data
Fig. 5c). As expected for resident macrophages, KCs are not replaced
by wild-type bone-marrow-derived cells in *Id3*-deficient parabiotic

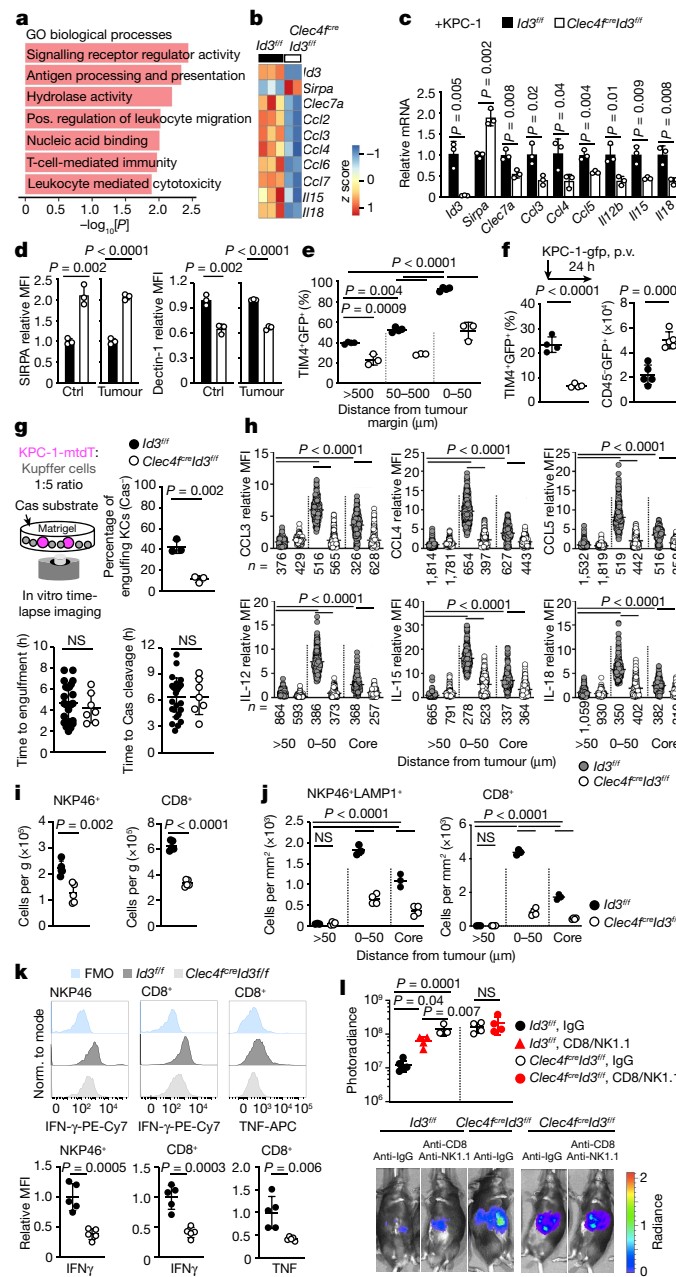

**Fig. 4 | ID3-dependent peritumoural niche. a,b**, RNA-seq analysis of KCs from *Clec4f^cre^Id3^f/f^* mice (*n* = 2) and *Id3^f/f^* littermates (*n* = 3). **a**, The pathways downregulated in *Clec4f^cre^Id3^f/f^* mice. Pos., positive. **b**, Selected differentially expressed genes. **c**, RT–qPCR analysis of selected genes in KCs from *Clec4f^cre^Id3^f/f^* mice and *Id3^f/f^* littermates. *n* = 3 per group. **d**, Flow cytometry analysis of the expression of SIRPA and dectin-1 by KCs from *Clec4f^cre^Id3^f/f^* mice and *Id3^f/f^* littermates 2 weeks after intraportal injection or not of 1 × 10^6 KPC-1-GFP cells. *n* = 3 per group. Ctrl, control. **e**, The percentage of TIM4^+ KCs containing GFP in the livers from *Clec4f^cre^Id3^f/f^* mice (*n* = 4) and *Id3^f/f^* littermates (*n* = 3) treated as described in **d**. **f**, Immunofluorescence analysis of the percentage of TIM4^+ KCs stained with GFP in *Clec4f^cre^Id3^f/f^* mice and *Id3^f/f^* littermates 24 h after intraportal injection of 1 × 10^6 KPC-1-GFP cells (left). *n* = 4 per group. Right, the number of CD45^− GFP^+ tumour cells per liver lobe was analysed using flow cytometry in the same mice. *n* = 5 per group. **g**, Engulfment of KPC-1-tdT cells by *Clec4f^cre^Id3^f/f^* or *Id3^f/f^* KCs in vitro, as in Fig. 2g. The plots show the percentage of KCs engulfing Cas-Green^− KPC-1-tdT cells (*n* = 3 per group), and the time from stable interaction between KCs (*n* = 25 (*Id3^f/f^*) and *n* = 7 (*Clec4f^cre^Id3^f/f^*)) and tumour cells to engulfment or caspase-3/7 cleavage. **h**, Expression of chemokines and cytokines by TIM4^+ KCs from *Id3^f/f^* and *Clec4f^cre^Id3^f/f^* mice treated as in **d**. *n* values are indicated. **i,j**, Flow cytometry (**i**) and immunofluorescence (**j**) analysis of CD8^+ T cells and LAMP1^+NKP46^+ cell numbers per g of liver or per mm^2 in mice from **h**. *n* = 5 mice per group. **k**, IFNγ and TNF expression by NKP46^+ and CD8^+ T cells from **i** and **j**. **l**, The liver tumour burden was analysed using photoradiance 2 weeks after injection of 1 × 10^6 KPC-1-luci cells in *Id3^f/f^* mice treated with IgG (*n* = 6) or anti-CD8/NK1.1 (*n* = 3) and *Clec4f^cre^Id3^f/f^* mice treated with IgG or anti-CD8/NK1.1. *n* = 4 per group. Statistical analysis was performed using unpaired two-tailed *t*-tests (**c**, **d**, **f**, **g**, **i**, **k** and **l**), one-way ANOVA (**e**, **j** and **l**) and Kruskal–Wallis tests (**h**). Data are mean ± s.d. Norm., normalized.

to the control (Fig. 3i and Extended Data Fig. 5f). Survival experiments after intraportal injection of KPC cells showed that *Clec4f^cre^Id3^f/f^* mice have reduced survival in comparison to the controls (Fig. 3j), comparable to that of *Clec4f^cre^R26^LSL-DTR^* mice treated with DT (Fig. 1g). Finally, *Clec4f^cre^Id3^f/f^* mice also developed larger liver metastases after intraportal injection of B16F10 melanoma, MC38 colon adenocarcinoma and LLC1 lung carcinoma (Fig. 3k). These data therefore suggest that, in addition to being required during embryonic development for KC differentiation, expression of ID3 is also necessary in adult KCs for their anti-tumour activity.

## ID3 controls the KC peritumoural niche

Differential gene expression analysis of RNA-seq data of KCs from *Clec4f^cre^Id3^f/f^* and control mice showed the downregulation of pathways associated with signalling receptor activity, leukocyte-mediated cytotoxicity, leukocyte migration and T-cell-mediated immunity in *Id3*-deficient KCs (Fig. 4a and Supplementary Data 3). *Id3* deficiency shifted the activatory/inhibitory receptor balance towards inhibitory receptor expression (Extended Data Fig. 3c). Notably the activating receptor dectin-1 (also known as CLEC7A)[1,37] was downregulated in *Id3*-deficient KCs from control and tumoural liver; by contrast, the macrophage inhibitory receptor *Sirpa* was overexpressed in the same KCs (Fig. 4b–d and Extended Data Fig. 6a). This analysis also identified ID3-independent receptors, including the activating receptor LRP1, which binds to tumour-expressed calreticulin[18] and the inhibitory receptor SIGLECG (also known as SIGLEC10), which binds to CD24 on tumour cells[7], which were expressed in *Id3*-deficient and control KCs (Extended Data Figs. 3c and 6b). Moreover, expression of the C–C chemokines CCL3, CCL4 and CCL5, and the cytokines IL-12, IL-15 and IL-18 was also downregulated in *Id3*-deficient KCs from tumoural liver (Fig. 4b,c).

Despite normal KC numbers and peritumoural location, the percentage of KCs carrying tumour material 2 weeks after intraportal injection of KPC cells was decreased by around half in *Id3*-deficient

mice (Extended Data Fig. 5d). Mice with ID3 deficiency during embryogenesis (*Id3^−/−^*) developed larger liver tumours and lung metastases in comparison to their littermate controls 2 weeks after intraportal injection of KPC cells (Fig. 3a), comparable to the phenotype of other KC-deficient mice (Fig. 1 and Extended Data Fig. 1). ID3 remains preferentially expressed at high levels in KCs after birth[24] (Fig. 3b), but its role in the function of adult KCs is unclear. KCs acquire expression of CLEC4F after birth (Fig. 3c), a time at which KC specification has been completed[24], and we therefore examined the consequences of *Id3* deletion in adult KCs in *Clec4f^cre^Id3^f/f^* mice. *Clec4f^cre^Id3^f/f^* mice presented with normal numbers of KCs, and normal KC morphology and ability to uptake 2 μm latex beads after intravenous (i.v.) injection as compared to the wild-type controls (Fig. 3d,e and Extended Data Fig. 5e). Moreover, *Id3*-deficient KCs were normally located outside and around metastatic tumours, similar to the controls (Fig. 3f). However, *Clec4f^cre^Id3^f/f^* mice still developed large liver and lung metastases comparable to that of KC-deficient mice in the short-term (Fig. 3g) and orthotopic (Fig. 3h) models. Flow cytometry analysis confirmed that liver tumours as well as phenotypic metastasis-initiating cells were increased in comparison

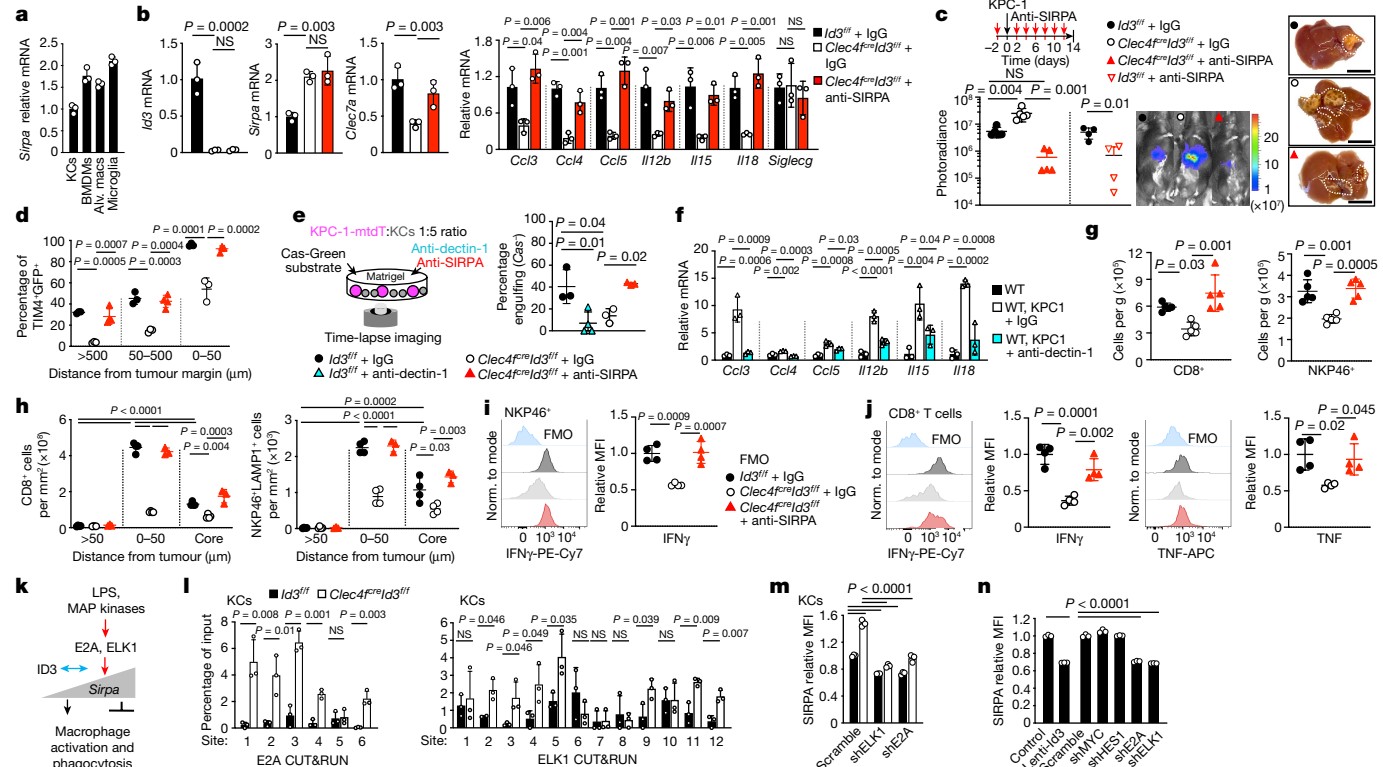

**Fig. 5 | SIRPA and dectin-1 mediate, in part, ID3 function. a**, RT–qPCR analysis of *Sirpa* mRNA in macrophages from three C57BL/6J mice. **b**–**d**, Analysis of *Clec4f^cre^Id3^f/f^* mice and *Id3^f/f^* littermates 2 weeks after intraportal injection of 1 × 10⁶ KPC-1-luci cells and treatment with anti-SIRPA or IgG control antibodies. **b**, RT–qPCR analysis of the indicated gene mRNA in KCs. *n* = 3 mice per group. **c**, Photoradiance and histology analysis of the liver tumour burden (*Id3^f/f^* + IgG, *Clec4f^cre^Id3^f/f^* + IgG or *Clec4f^cre^Id3^f/f^* + anti-SIRPA (*n* = 5 per group); *Id3^f/f^* + IgG and *Id3^f/f^* + anti-SIRPA (*n* = 4 per group)). Scale bars, 1 cm. **d**, The percentage of TIM4⁺ KCs containing GFP⁺ material in the peritumoural niche. *n* = 4 (*Id3^f/f^* + IgG and *Clec4f^cre^Id3^f/f^* + IgG) and *n* = 3 (*Clec4f^cre^Id3^f/f^* + anti-SIRPA). **e**, In vitro engulfment of live KPC-1-membrane-tdT (KPC-1-mtdT) cells (Fig. 2g) by KCs from *Id3^f/f^* mice treated with IgG (*n* = 3 independent experiments) or anti-dectin-1 antibodies (*n* = 4) and *Clec4f^cre^Id3^f/f^* littermates treated with IgG and anti-SIRPA antibodies (*n* = 3 per group). **f**, RT–qPCR analysis of mRNA gene expression by wild-type (WT) KCs that were or were not cocultured with KPC-1

cells for 12 h in the presence of anti-dectin-1 or control IgG antibodies. *n* = 3 per group. **g**–**j**, Flow cytometry analysis of the numbers of CD8⁺ T cells and NK cells (**g**; *n* = 5 per group), immunofluorescence analysis (**h**; *n* = 4 per group) and analysis of the production of cytokines by CD8⁺ T cells (**j**) and NK cells (**i**) (*n* = 4 per group) in tumoural liver from mice treated as described in **b**–**d**. **k**, Hypothesis for the regulation by ID3 of *Sirpa* transactivation in macrophages. **l**, CUT&RUN analysis of E2A and ELK1 binding to *Sirpa* predicted promoter/ enhancer regions in KCs from *Clec4f^cre^Id3^f/f^* and *Id3^f/f^* littermates. *n* = 3 per group. **m**, Flow cytometry analysis of SIRPA expression by KCs from *Id3^+/+^* mice and *Id3^−/−^* littermates, expressing scramble, *E2a* or *Elk1* shRNAs (*n* = 3 per group). **n**, Flow cytometry analysis of SIRPA expression by mouse BMDMs expressing lenti-Id3, lenti-control or the indicated shRNAs. *n* = 3 per group. Statistical analysis was performed using one-way ANOVA (**b**–**j**, **m** and **n**) and unpaired two-tailed *t*-tests (**c** and **l**). The dots represent individual mice (**a**–**d** and **g**–**j**). Data are mean ± s.d.

mice in comparison to the control (Fig. 4e). The percentage of KCs carrying tumour material in the liver of *Clec4f^cre^Id3^f/f^* mice 24 h after portal injection of KPC cells was also decreased by around threefold (Fig. 4f), and the number of live tumour cells (CD45⁻tdT⁺) was increased 2.5-fold (Fig. 4f). The percentage of *Id3*-deficient KCs engulfing one or more Cas-Green⁻ KPC cells in the course of the 20 h in vitro time-lapse imaging assay was decreased by fivefold (Fig. 4g), although the average delay between contact and engulfment or caspase-3/7 cleavage in *Id3*-deficient and control KCs was similar (Fig. 4g).

Immunofluorescence staining confirmed that the expression of chemokines CCL3, CCL4 and CCL5 and the cytokines IL-12, IL-15 and IL-18 by KCs in the liver of *Clec4f^cre^Id3^f/f^* mice was reduced or abolished at the tumour margin and in the tumour in comparison to in the littermate controls (Fig. 4h and Extended Data Fig. 6c,d). NK and CD8⁺ T cell numbers were selectively reduced in the livers of *Clec4f^cre^Id3^f/f^* tumour-bearing mice as shown by flow cytometry analysis (Fig. 4i and Extended Data Fig. 6e–g) and, specifically, from the peritumoural zone and the tumours as shown by immunofluorescence analysis (Fig. 4j and Extended Data Fig. 6h). Moreover, although NK and CD8⁺ T cells recruited to liver tumours of wild-type

mice produce IFNγ and TNF (Fig. 4k), the production of IFNγ and TNF was reduced in the remaining NK and CD8⁺ T cells present in the liver of tumour-bearing *Clec4f^cre^Id3^f/f^* mice (Fig. 4k). Furthermore, in vitro co-culture of fluorescence-activated cell sorting (FACS)-sorted KCs from *Clec4f^cre^Id3^f/f^* mice and *Id3^f/f^* (wild-type) littermates with or without KPC cells showed that tumour cells induced high expression by KCs of CCL3, CCL4 and IL-18 in an ID3-dependent manner and, to a lesser extent, of CCL5, IL-15 and IL-12 (Extended Data Fig. 6i). Moreover, the supernatants of *Id3^f/f^* (wid-type) KC/KPC cocultures, but not of ID3-deficient KC/KPC cocultures, were sufficient to stimulate IFNγ expression by NK cells (Extended Data Fig. 6j).

Depletion of CD8 and NK cells with antibodies increased tumour growth in wild-type mice, but not to the level of *Clec4f^cre^Id3^f/f^* mice, and did not further increase tumour growth in *Clec4f^cre^Id3^f/f^* mice (Fig. 4l), suggesting that both phagocytosis and the recruitment and/ or activation of effector lymphoid cells contribute to the anti-tumour activity of KCs. Together, these data indicate that ID3 deficiency in KCs impairs their activation by tumour cells, possibly through dys-regulating expression by KCs of macrophage inhibitory and activating receptors, resulting in impaired phagocytosis of tumour cells, and

decreased recruitment and non-cognate activation of effector CD8⁺ T cells and NK cells.

## SIRPA blockade rescues ID3-deficient KCs

Wild-type KCs express higher levels of *Id3* (Fig. 3b), and lower levels of *Sirpa* compared with other macrophage subsets including microglia, alveolar macrophages and BMDMs (Fig. 5a). SIRPA binding to its ligand CD47 inhibits macrophage activation and phagocytosis[4,5]. By contrast, dectin-1, which recognizes tumour cell antigens, activates macrophages in tumours[37], and can prime cytotoxic T cell responses[45,46]. We therefore reasoned that ID3 may regulate the inhibitory/activating receptor balance in macrophages, and that the control of SIRPA and dectin-1 expression may underlie at least part of the anti-tumour activities of wild-type KCs. We performed rescue experiments to test this hypothesis. In vivo blockade of SIRPA with antibodies rescued the expression of dectin-1, CCL3, CCL4, CCL5, IL-12, IL-15 and IL-18 by *Id3*-deficient KCs (Fig. 5b and Extended Data Fig. 7a–d). SIRPA blockade restricted the development of liver metastases in *Clec4f*^cre^*Id3*^f/f^ mice (Fig. 5c), and rescued the phagocytosis of tumour cells by *Id3*-deficient KCs to wild-type levels in vivo (Fig. 5d) and in vitro (Fig. 5e). Conversely, dectin-1 blocking antibodies abolished phagocytosis of tumour cells by wild-type KCs (Fig. 5e), and also decreased their production of chemokines and cytokines (Fig. 5f).

SIRPA blockade also rescued the numbers of CD8⁺ T cells and NKP46⁺ NK cells in tumour-bearing livers of *Clec4f*^cre^*Id3*^f/f^ mice (Fig. 5g and Extended Data Fig. 7e), the formation of the peritumoural CD8⁺ T cell and NKP46⁺ NK-rich zone (Fig. 5h and Extended Data Fig. 7f) and the production of IFNγ and TNF by CD8⁺ T cells and NK cells (Fig. 5i,j). Moreover, genetic deletion of the SIRPA ligand *Cd47* on tumour cells also restricted tumour growth in *Clec4f*^cre^*Id3*^f/f^ mice (Extended Data Fig. 7g–i) and rescued CD8 T cell and NK cell recruitment to wild-type levels (Extended Data Fig. 7j). Thus, the regulation of SIRPA and dectin-1 expression by ID3 in KCs underlies, at least in part, the mediation of phagocytosis, inflammatory chemokine production, and recruitment and activation of NK and CD8⁺ T cells. Notably, our results suggest that signalling by SIRPA itself controlled, in part, the expression of the activating receptor dectin-1 by KCs.

## ID3 buffers SIRPA transactivation

We therefore sought to identify the molecular-level mechanism by which ID3 may control *Sirpa* expression in KCs, test its potential physiological importance and investigate whether the same mechanism can endow other macrophages with anti-tumour activity. ID proteins exert their biological effects by blocking the DNA-binding activity of class I bHLH E-proteins, Pax-, Ets- and Ets-domain transcription factors from the ternary complex factor (TCF) family[43,47–49]. Among these, the E protein E2A encoded by *Tcf3* (also known as *Tcfe2a*), as well as the TCF factor ELK1, are highly expressed in macrophages in general and in KCs in particular (Extended Data Fig. 8a,b). ELK1 links gene transcription to RAS/MAPK/ERK signalling in response to cellular stress and environmental cues such as LPS. ELK1 DNA binding and transcriptional activity are stimulated by the phosphorylation of its C-terminal domain (C-box) by ERK1/2. E2A is a transcriptional activator conserved from yeast to humans and its expression and DNA-binding activity is also induced by LPS[50]. The liver drains blood from the gut through the portal circulation and is therefore constantly and directly exposed to stress signals including bacterial products[27]. From a physiological perspective, we therefore hypothesized that the expression of ID3 by KCs may allow downregulation of *Sirpa* expression by limiting the binding of E proteins and ELK1 to *Sirpa* promoter/enhancer regions to maintain KC phagocytic activity within an inflammatory environment (Fig. 5k).

A deep learning analysis identified 12 putative ELK1-binding sites and 6 putative E-box-binding sites at upstream enhancer regions, intronic enhancers and the *Sirpa* promoter in mouse KCs and BMDMs (Extended Data Fig. 8c,d). These regions are all in proximity or connection to the *Sirpa* promoter according to the H3K4me3 HiChIP data of BMDMs (Extended Data Fig. 8c). We therefore performed cleavage under targets and release using nuclease (CUT&RUN) analyses of E2A and ELK1 binding to DNA at these sites in KCs from control and *Clec4f*^cre^*Id3*^f/f^ littermates. These experiments indicated that ID3 prevents the binding of E2A and ELK1 at the *Sirpa* promoter and the upstream and intronic *Sirpa* enhancer regions (Fig. 5l). Moreover, shRNAs targeting E2A and ELK1 both reduced *Sirpa* expression in *Id3*-deficient KCs to wild-type levels (Fig. 5m), indicating that E2A and ELK1 are required for upregulation of *Sirpa* expression in *Id3*-deficient KCs. BMDMs express low levels of *Id3* and high levels of *Sirpa* (Figs. 3b and 5a) but share active regulatory regions at the *Sirpa* locus with KCs (Extended Data Fig. 8c). Accordingly, we found that overexpression of *Id3*, as well as short-hairpin RNAs (shRNAs) targeting *Tcf3* and *Elk1*, all reduce *Sirpa* expression in BMDMs (Fig. 5n). As expected, LPS (2 mg per kg) further increased ELK1 and E2A binding to *Sirpa* enhancer/promoter regions in *Id3*-deficient but not wild-type KCs (Extended Data Fig. 8e,f). Consistently, LPS increased *Sirpa* expression in *Id3*-deficient KCs and in wild-type BMDMs, but not in wild-type KCs or BMDMs overexpressing ID3 (Extended Data Fig. 8g,h). Finally, shRNA against *Tcf3* or *Elk1* in BMDMs was sufficient to abrogate the LPS-mediated increase in *Sirpa* expression in BMDMs (Extended Data Fig. 8h). Although the comparison of human and mice non-coding sequences is difficult, a simple mouse/human BLASTn alignment for *Sirpa* regulatory elements identified conserved ELK1-binding motifs in the *Sirpa* enhancer and promoter regions (Extended Data Fig. 9a), suggesting that the role of ID3 may be conserved in human macrophages. Accordingly, we found that lentiviral-mediated expression of mouse *Id3* in mouse BMDMs and human *ID3* in hiPSC-Macs resulted in the selective downregulation of *Sirpa* expression as well as the upregulation of *Clec7a* in the two cell types (Fig. 6a,b). Together, these results strongly suggest that ID3 represses *Sirpa* expression by preventing DNA binding of E2A and ELK1 to *Sirpa* enhancer/promoter regions, a property that characterizes wild-type KCs but can also be transferred to other macrophages such as BMDMs or human macrophages, through enforced expression of *Id3* or knockdown of *Tcf3* or *Elk1*.

## Conserved features of human KCs

Human KCs express higher levels of *ID3* and lower levels of *SIRPA* compared with other monocytes and macrophage subsets[51,52] (https://www.proteinatlas.org) (Extended Data Fig. 9b). Immunofluorescence analysis of liver metastasis samples from three patients with pancreatic ductal adenocarcinoma (PDAC) showed that human TIM4⁺ KCs are enriched in the peritumoural liver in comparison to in tumour nodules (Extended Data Fig. 9c), as observed in mouse models. Peritumoural human KCs also contained CK19⁺ tumour material (Extended Data Fig. 9d) suggesting engulfment of tumour cells. Moreover, immunofluorescence analysis indicated that peritumoural human KCs expressed CCL3, CCL4 and CCL5 as well as IL-12, IL-15 and IL-18 (Extended Data Fig. 9e). To confirm these findings, we next reanalysed two single-cell RNA sequencing (scRNA-seq) datasets from human PDAC metastatic liver, and human colorectal carcinoma (CRC) metastatic liver (Extended Data Fig. 9f). In both cases, KCs represented the majority of macrophages in normal liver, but a minor subset in tumoural liver (Extended Data Fig. 9f,g), and expressed higher *ID3*, lower *SIRPA* and higher chemokine (*CCL4*, *CCL3*) and cytokines genes (*IL18*) in comparison to CD14⁺TIM4⁻ tumour-associated macrophages (Extended Data Fig. 9h).

## ID3 confers anti-tumour activity

In the context of cancer, the above data together suggested that ectopic expression of ID3 in macrophages with low ID3, high SIRPA and low anti-tumour activity, such as mouse BMDMs or hiPSC-Macs, may endow

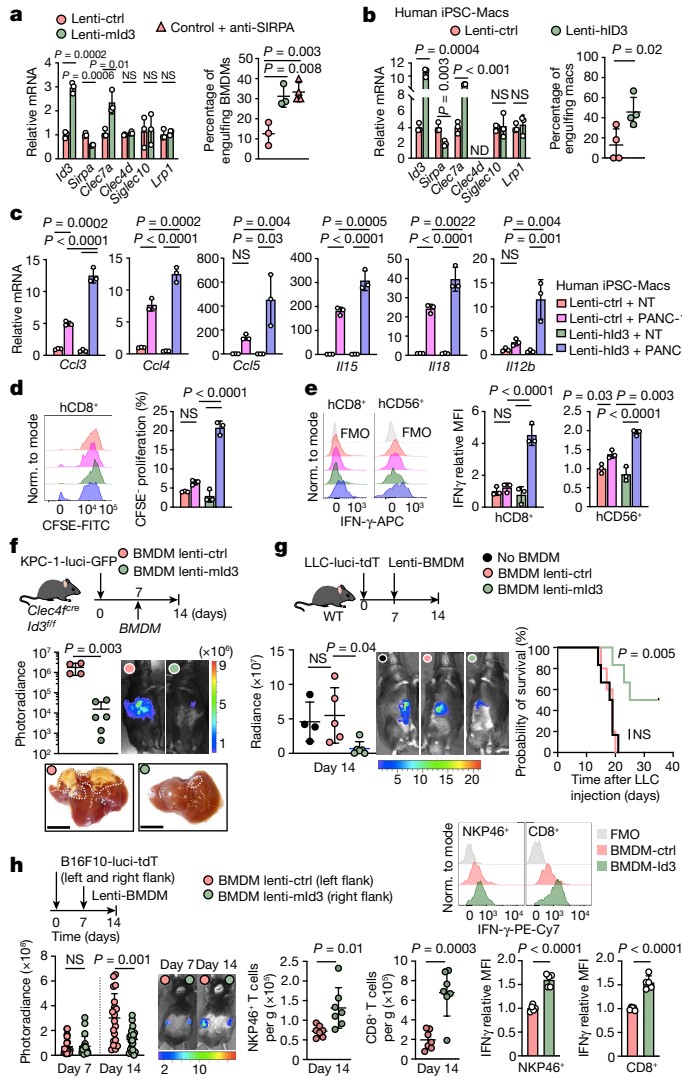

**Fig. 6 | ID3 expression endows macrophages with anti-tumour activity.**
**a**, Left, RT–qPCR analysis of the indicated genes in mouse BMDMs expressing lenti-mouse-Id3 (Lenti-mId3) or lenti-control (Lenti-ctrl). $n = 3$ independent experiments. Right, in vitro engulfment of Cas-Green$^-$ KPC-1-mtdT cells by mouse BMDMs expressing lenti-mId3 ($n = 3$) or lenti-control with or without anti-SIRPA blocking antibodies ($n = 3$ and 4). **b**, RT–qPCR analysis of the indicated genes. Right, engulfment of Cas-Green$^-$ PANC-1-mtdT cells by hiPSC-Macs expressing lenti-human-ID3 (lenti-hID3) or lenti-control. $n = 4$ experiments. **c**, RT–qPCR analysis of chemokines and cytokines in hiPSC-Macs expressing lenti-hID3 or lenti-control, cultured alone or with PANC-1 cells for 48 h. NT denotes macrophages that were not cultured with PANC-1 cells. $n = 3$ experiments. **d**,**e**, Flow cytometry analysis of the proliferation of human CD8$^+$ T cells (**d**) and IFNγ expression by human CD8$^+$ T cells and CD56$^+$ NK cells (**e**) that were cultured with supernatants from hiPSC-Mac–PANC-1 cell cocultures in **c**. $n = 3$ experiments. **f**, Liver pictures and bioluminescence analysis of the liver tumour burden of *Clec4f$^{cre}$Id3$^{f/f}$* mice 2 weeks after intraportal injection of $1 × 10^6$ KPC-1-luci cells and intraportal injection of $1 × 10^6$ BMDMs expressing lenti-control ($n = 4$) or lenti-mId3 cells ($n = 6$) at D7. Scale bars, 1 cm. **g**, C57BL/6J mice received $1 × 10^6$ LLC1-luci cells by intraportal injection at D0, and $1 × 10^6$ BMDMs expressing lenti-control or lenti-mId3, or PBS at D7. Left, bioluminescence analysis of liver tumour burden at D14. $n = 4$ (PBS), $n = 5$ (BMDMs + lenti-control) and $n = 6$ (BMDMs + lenti-mId3). Right, survival analysis. $n = 6$, 5 and 6 mice per group, respectively. **h**, C57BL/6J mice received subcutaneous injection of $1 × 10^6$ B16F10-luci-tdT cells into the left and right flanks at D0 and intratumour injection of $5 × 10^5$ BMDMs expressing lenti-control or lenti-mId3 cells at D7. At D14, the flank tumour burden was analysed by bioluminescence ($n = 20$ mice per group; left), and the recruitment of NKP46$^+$ NK and CD8$^+$ T cells ($n = 7$ mice per group; middle) and IFNγ production ($n = 5$ mice per group; right) were analysed using flow cytometry. Statistical analysis was performed using one-way ANOVA (**a**, **c**–**e** and **g** (left)), log-rank (Mantel–Cox) tests (**g** (right)), two-tailed Mann–Whitney $U$-tests (**h**) and unpaired two-tailed $t$-tests (**a**, **b**, **f** and **h**). Data are mean ± s.d.

treatment or with lenti-control BMDMs, wild-type mice required euthanasia within 3 weeks, whereas around half of the mice were alive at 5 weeks after treatment with ID3-expressing BMDMs (Fig. 6g). Finally, in a melanoma model in which B16F10 cells are injected subcutaneously in the two flanks of a wild-type mice, intratumoural injection of ID3-expressing BMDMs, but not control BMDMs, blocked tumour growth in the corresponding flank, but not the contralateral flank (Fig. 6h), and triggered accumulation of activated CD8$^+$ T cells and NK cells producing IFNγ and TNF to the B16F10 tumours in the corresponding flank (Fig. 6h and Extended Data Fig. 10c,d). These results show that, in addition to being required for the anti-tumour activity of KCs, expression of *ID3* is also sufficient to endow mouse and human macrophages with a potent, local and innate anti-tumour activity, in vitro and in vivo, against epithelial and melanocytic cancers (Extended Data Fig. 10e).

## Discussion

The role of macrophages in cancer growth and metastasis has been obscured by the transcriptional and functional diversity of tissue macrophages, between and within tissue microenvironments. The recent genetic dissection of tissue macrophage diversity led to the identification of LDFs that may control the differentiation and functions of tissue-specific subsets[24]. The helix-loop-helix transcriptional repressor ID3 is a KC LDF that is preferentially expressed by embryonic macrophage precursors in the liver and is necessary for their differentiation into KCs[24]. ID3 remains expressed at high levels in adult KCs and we show here that it is dispensable for the maintenance or general functions of adult KCs such as survival, anatomical distribution or the ability to uptake small particles such as latex beads from the circulation, but that it controls their activation by live tumour cells and anti-tumour immunity. These results support the hypothesis that expression of

them with high anti-tumour activity. In support, we found that in vitro ectopic lentiviral-mediated expression of mouse *Id3* in mouse BMDMs resulted in their ability to phagocytose KPC tumour cells comparable to that of SIRPA blockade or of wild-type mouse KCs (Fig. 6a and Extended Data Fig. 10a). Similarly, hiPSC-Macs poorly phagocytose human PANC-1 tumoural cells, but expression of human *ID3* endowed them with the ability to phagocytose PANC-1 cells comparable to that of mouse ID3-expressing macrophages (Fig. 6b). Furthermore, ectopic expression of ID3 in hiPSC-Macs also increased their production of the chemokines CCL3, CCL4 and CCL5, and the cytokines IL-12, IL-15 and IL-18 in response to tumour cells in vitro (Fig. 6c). Finally, supernatants from cocultures of lenti-ID3 hiPSC-Macs and tumour cells were sufficient to trigger the proliferation of CD8$^+$ T cells (Fig. 6d), the production of IFNγ by CD8$^+$ T cells and NK cells (Fig. 6e), and the production of TNF by CD8$^+$ T cells (Extended Data Fig. 10b), whereas the supernatant of lenti-control iPSC-Mac/tumour cell cocultures had little or no effect (Fig. 6d,e and Extended Data Fig. 10b).

We therefore tested the ability of ID3-expressing BMDMs to limit tumour growth in vivo. In the KPC model, we found that intraportal injection of $10^6$ ID3-expressing BMDMs 7 days after intraportal injection of $10^6$ KPC cells in *Clec4f$^{cre}$Id3$^{f/f}$* mice prevented the growth of liver tumours after 2 weeks (Fig. 6f). Intraportal injection of ID3-expressing BMDMs also prevented the growth of liver tumours in wild-type mice after 2 weeks in the more aggressive LLC model (Fig. 6g). Moreover, a survival analysis in this model indicated that ID3-expressing BMDMs improved the survival of wild-type mice. In the absence of BMDM

macrophage LDFs is not only important for the specification of tissue macrophages from embryonic precursors, but also controls essential tissue-specific functions in adult tissues. Here we show that expression of ID3 by liver-resident KCs is necessary to orchestrate the formation of a peritumoural niche, characterized by a potent phagocytic activity against tumour cells and the recruitment and activation of a lymphoid anti-tumour immune response, which restricts engraftment and growth of a variety of cancer lines in the liver. Moreover, we show that enforced expression of ID3 in other mouse and human macrophages such as iPSC-Macs is also sufficient to endow them with the ability to mount this anti-tumour response.

Mechanistically, ID3 controls the activatory/inhibitory receptor balance, which, in turn, controls KC activation by tumour cells. ID3 directly represses transactivation of the inhibitory receptor *Sirpa* by inhibiting binding of the E-box E2A and the MAP-kinase target ELK1 to the promoter/enhancer regions of the *Sirpa* gene, to a level low enough to allow KC activation by CD47-expressing cancer cells, at least in part through the activating receptor dectin-1. KC activation results in phagocytosis of live tumour cells, recruitment of NK cell and CD8+ T cells to the tumour and peritumoural niche through the production of chemokines, and activation of these lymphoid effector cells to proliferate and produce IFNγ and TNF, at least in part through the production of the cytokines IL-12, IL-15 and IL-18 (Extended Data Fig. 10e). As IFNγ stimulates the production of IL-12 by KCs in tumour-bearing mice[41], a feed-forward loop between ID3-expressing macrophages and effector lymphoid cells may contribute to the anti-tumour effect driven by macrophages. In addition to this non-cognate, or innate, activation of lymphoid cells, ID3-dependent engulfment of tumour cells may also regulate cross-presentation of tumour antigens by macrophages to T cells, although KCs have been consistently shown to be poor cognate antigen-presenting cells[53,54]. Finally, the anti-tumour response of ID3-expressing macrophages is local in a subcutaneous tumour model, although lung and spleen metastases are reduced in a liver metastasis model, which can be attributed to a debulking effect. We also found that ID3 buffers *Sirpa* upregulation by LPS in macrophages, which may be relevant to the phagocytic activity of KCs as the liver drains microbial products from the gut[27], but may also be of interest for the function of ID3-expressing macrophages in the inflammatory tumour microenvironment.

In summary, here we show that ID3 is a master regulator of the immune response to cancer that is necessary and sufficient to orchestrate a potent local macrophage-driven anti-tumour response, and could be harnessed for the treatment of cancer. Our results suggest that the engineering of ID3-expressing macrophages could be taken into consideration in the design of future therapeutic strategies.

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

## Methods

### Materials

**Mice.** Animal procedures were performed in adherence with the Institutional Review Board (IACUC 15-04-006) at Memorial Sloan Kettering Cancer Center (MSKCC). *Rosa26*[LSL-YFP] (ref. 55), *Rosa26*[LSL-tdT] (ref. 56), *Flt3*[cre] (ref. 57), *Tnfrsf11a*[cre] (ref. 58), *Id3*[f/f] (ref. 59), *Id3*[−/−] (ref. 60), *Cx3cr1*[gfp/+] (ref. 61), *CCR2*[−/−] (ref. 30), *Csf1r*[f/f] (ref. 62), *Spi1*[f/f] (ref. 63), *Cxcr4*[gfp/+] (ref. 64), *Cxcr4*[creERT2] (ref. 65), *Clec4f*[cre-tdT] (ref. 25), *Rosa26*[LSL-DTR] (ref. 66), *p48*[cre] (ref. 67), *Trp53*[LSL-R172H] (ref. 68), *Kras*[LSL-G12D] (ref. 69), CD45.1 mice and C57BL/6J mice were purchased from Jackson Lab. Mice were bred under SPF conditions, under a 12 h–12 h light–dark cycle, at around 21–22 °C with 30–70% humidity. A list of mouse strains and the genotyping protocol is provided in Supplementary Data 4.

**Human tissue samples.** All of the procedures performed in studies involving human participants were conducted according to the Declaration of Helsinki. Human tissues were obtained with patient-informed consent and used under approval by the Institutional Review Boards from Memorial Sloan Kettering Cancer Center (IRB protocols, 15-021).

A list of the reagents, plasmids, antibodies and qPCR primers purchased and used in this study is provided in Supplementary Data 5.

**Mouse cell lines.** KPC-1 and KPC-2 cell lines[70] were obtained from pancreas tissue from *p48*[cre]*Trp53*[LSL-R172H]*Kras*[LSL-G12D] mice[71]. Pan02[72,73] (DCTD Tumour Repository) cells were provided by D. Lyden. The Colon adenocarcinoma cell line MC38[73,74] and melanoma cell line B16F10[75] (ATCC, CRL-6475 was a gift from J. D. Wolchok). Lewis lung carcinoma line LLC1 cells[76] (ATCC, CRL-1642) were purchased from ATCC. KPC-1, KPC-2 and PANC-1 cells were cultured in DMEM (Gibco) supplemented with 10% fetal bovine serum, 100 U ml[−1] penicillin and 100 μg ml[−1] streptomycin (Invitrogen). Panc02 cells, LLC1 cells, MC38 cells and B16F10 cells were cultured in RPMI 1640 (Gibco) supplemented with 10% fetal bovine serum, 100 U ml[−1] penicillin and 100 μg ml[−1] streptomycin (Invitrogen) and incubated at 37 °C in 5% $CO_2$.

**Mouse primary cells.** Mouse BMDMs were obtained as follows. Femur, tibia and iliac bones from C57BL/6J mice were flushed with PBS, red blood cells were lysed using red blood cell lysis buffer (eBioscience) and bone marrow cells were seeded per 15 cm non-tissue culture plate in DMEM (Thermo Fisher Scientific) with 10% FBS (Thermo Fisher Scientific), 20 ng ml[−1] M-CSF (315-02-50ug, PeproTech), 100 U ml[−1] penicillin–streptomycin (Thermo Fisher Scientific) for 7 days.

**Mouse splenic NK cells.** Mouse splenic NK cells were obtained as follows. Splenic cell suspensions from C57BL/6J mice were dissociated and passed through a 100 μm cell strainer (BD), red blood cells were lysed using red blood cell lysis buffer (eBioscience) and resuspended in 50 μl of blocking buffer containing anti-mouse CD16/32 antibodies (1:100) for 15 min at 4 °C, followed by staining with PE-anti-NKP46 antibodies for 30 min at 4 °C and anti-PE microbeads (Miltenyi Biotec) for 30 min at 4 °C, respectively. NKP46[+] NK cells were isolated by passing stained samples through the Miltenyi Biotec Magnetic separation system using LS columns according to the manufacturer's instructions.

**Human cell lines.** PANC-1 cells[77] (ATCC, CRL-1469) were purchased from ATCC. Human macrophages were obtained from hiPS cell lines derived from frozen peripheral blood mononuclear cells of two independent healthy donors. Written informed consent was obtained according to the Helsinki convention. The study was approved by the Institutional Review Board of St Thomas' Hospital; Guy's hospital; the King's College London University and the Memorial Sloan Kettering Cancer Center. hiPS cells were derived according to published protocols[78] using Sendai viral vectors (Thermo Fisher Scientific; A16517).

Newly derived iPS cell clones were maintained in culture for 10 passages (2–3 months) to remove any traces of Sendai viral particles and ensure that the cells remain stable during a prolonged culturing period. Over 90% of iPSCs in the derived lines expressed high levels of the pluripotency markers NANOG and OCT4, as determined using flow cytometry. Karyotyping analysis showed a normal karyotype (46, XX). iPS cell lines tested negative for mycoplasma contamination using MycoAlert Plus kit (Lonza). iPS cell clones that passed all of the quality-control checks were frozen down and used for downstream experiments. hiPS cells were maintained on irradiated CF1 mouse embryonic fibroblasts (MEFs; Thermo Fisher Scientific; A34181) in embryonic stem (ES) cell medium supplemented with 10 ng ml[−1] basic fibroblast growth factor (bFGF, Peprotech; 100-18B). The medium was changed every other day. Passaging was performed every 7 days at 1/4–1/6 dilution ratio depending on colony size. During passaging, iPS cells were detached as clusters by incubation for 13 min at 37 °C with collagenase type IV (250 UI ml[−1] final concentration) (Thermo Fisher Scientific; 17104019) and were pelleted at room temperature by centrifugation at 150*g*.

**hiPSC-Macs.** iPS cell clusters were resuspended in ES cell medium supplemented with 10 ng ml[−1] bFGF (Peprotech; 100-18B) and plated onto NUNC plates containing 12,500 to 16,000 MEFs per cm[2]. hiPSC-Macs were obtained using a previously published protocol[79], modified as follows. At day 0 of differentiation, expanded hiPS cells were detached as described above and transferred (from a 150 mm plate, to four wells) for cultivation in six-well low-adhesion plates in ES cell medium supplemented with 10 μM ROCK inhibitor (Sigma-Aldrich; Y0503). The plates were kept on an orbital shaker at 100 rpm for 6 days to allow for the spontaneous formation of embryoid bodies with haematopoietic potential. At day 6 of differentiation, 200–500 μm cystic embryoid bodies were picked under a dissecting microscope and transferred onto adherent tissue culture plates (around 2.5 embryoid bodies per cm[2]) for cultivation in HD medium. At day 18 of differentiation, macrophages produced by embryoid bodies were collected from suspension and cultivated on tissue culture plates at a density of around 10,000 cells per cm[2] in MC medium for 6 days in before use for downstream experiments. All cells were cultured at 37 °C, 5% $CO_2$, in standard tissue culture incubators.

### Human primary cells

**Human NK cells.** Human NK cells (IQ Biosciences, IQB-Hu1-NK5) were cultured in NK MACS medium (Miltenyi Biotec) supplemented with 10% heat-inactivated pooled human AB serum (Sigma-Aldrich), 100 U ml[−1] penicillin and 100 μg ml[−1] streptomycin (Invitrogen) and 20 ng ml[−1] hIL-2 incubated at 37 °C in 5% $CO_2$.

**Human CD8 T cells.** Human CD8 T cells (IQ Biosciences, IQB-Hu1-CD8T10) were cultured with RPMI supplemented with 10% heat-inactivated pooled human AB serum, 100 U ml[−1] penicillin and 100 μg ml[−1] streptomycin and 20 ng ml[−1] hIL-2 incubated at 37 °C in 5% $CO_2$.

### Methodology

**In vivo treatment with DT, CSF1R inhibitor, phosphatidylserine blockade, SIRPA blocking antibodies, and NK and CD8 T cell blocking antibodies.** For DT-mediated depletion of KCs, *Clec4f*[cre]*Rosa26*[LSL-DTR] mice and *Rosa26*[LSL-DTR] mice were intraperitoneally injected with 100 ng DT (D0564-1MG, Sigma-Aldrich) as a single dose or weekly injections[25], as indicated in the figure legends for the corresponding experiments. The efficiency of KC depletion was determined in Extended Data Fig. 1j.

For CSF1R inhibitor treatment with PLX5622[31], C57BL/6J mice were fed ad libitum with PLX5622-impregnated chow (1,200 mg per kg, provided by Plexxicon) or control chow 2 weeks before injection of tumour cells.

For the SIRPA blockage assay, *Clec4f*[cre]*Id3*[f/f] mice and *Id3*[f/f] littermates were intraperitoneally injected with control IgG (HRPN, BioXcell) or 250 µg anti-SIRPA (P84, BioXcell) 2 days before and every 2 days after tumour cell injection.

For phosphatidylserine blockade[80], C57BL/6J mice were injected i.v. with PBS or 1 µg MFG-E8(D89E)[81] (gift from S. Nagata), 6 h before injection of tumour cells.

For NK cell and CD8 T cell depletion, 8–12-week-old *Clec4f*[cre]*Id3*[f/f] mice and *Id3*[f/f] littermates were intraperitoneally injected with control IgG (HRPN, BioXcell), or 200 µg anti-NK1.1 (BE0036, BioXcell) and 200 µg anti-CD8 (BP0061, BioXcell) 1 day before tumour injection and every 4 days afterwards.

**Metastasis-initiating cell assays. In vitro oncosphere formation.** Sphere-formation assays[82] were performed by sorting 1,000 CD9[+]CD133[+] KPC-1 cells per well and 1,000 CD9[−]CD133[−] KPC-1 cells per well and plating them in ultra-low-attachment 96-well plates (Corning) in DMEM/F-12 medium supplemented with B-27 serum (1:50, Invitrogen), 20 ng ml[−1] bFGF (R&D Systems) and 50 U ml[−1] penicillin–streptomycin for a total of 7 days. Images were acquired using the Leica DM IL inverted phase-contrast microscope with Leica application SuiteX software and quantified using ImageJ.

**In vivo metastasis assay.** In vivo metastasis assays were performed by sorting $5 \times 10^4$ or $2 \times 10^5$ CD47[bright]CD9[+]CD133[+] tumour cells and CD47[low]CD9[low]CD133[low] tumour cells from KPC-1-GFP tumour cells, followed by intraportal injection into C57BL/6J mice for 1 week. Metastatic potential was determined by bioluminescence imaging analysis (see the 'Short-term liver metastasis model (intraportal injection of tumour cell lines)' section below).

**Transduction of mouse and human tumour cell lines.** To generate luci-tdT-, luci-GFP- and mtdT-expressing tumour cell lines, KPC-1, KPC-2, PAN02, MC38, B16F10 and LLC1 cells were seeded at a density of $5 \times 10^5$ cells per well in six-well culture plates. Lentiviral supernatant carrying the *luciferase-gfp* gene (plasmid pFUGW-FerH-ffLuc2-eGFP, Addgene, 71393) or the *luciferase-tdT* gene (plasmid pUltra-Chili-Luc, Addgene, 48688) and 10 µg ml[−1] polybrene (Millipore-Sigma) were added to the tumour cell culture medium for 12 h. The medium was then replaced and, 72 h later, GFP[+] or tdT[+] tumour cells were FACS-sorted three times before in vivo injection in mice.

To generate mtdT-expressing tumour cells, KPC-1 and PANC-1 cells were seeded at a density of $5 \times 10^5$ cells per well in a six-well culture plate. The lentiviral supernatant carrying retroviruses expressing the *mtdT* gene (plasmid pQC membrane tdTomato IX, Addgene, 37351) and 10 µg ml[−1] polybrene (Millipore-Sigma) were added to the tumour cell culture medium for 12 h. The medium was then replaced and, 72 h later, tdT[+] tumour cells were FACS-sorted three times before use in experiments.

To generate *Cd47*-KO KPC tumour cells, sgRNAs targeting *Cd47* sequence, sgCD47 5′-CCTTGCATCGTCCGTAATG-3′[83] were cloned into pSpCas9(BB)-2A-Puro (PX459) V2.0 (Addgene, 62988) according to the Addgene cloning protocol. To establish *Cd47*-KO KPC cell line, electroporation of pSpCas9-sgCD47 plasmid into $1 \times 10^6$ KPC-1-luci-tdT cells was performed according to the Neon Transfection System protocol. Cells were FACS-sorted based on the loss of CD47 staining with anti-mouse CD47-AF647 antibodies (1:200, BioLegend) for three rounds to get pure populations of *Cd47*-KO cells.

**Transduction of macrophages.** For lentiviral transduction of BMDMs, BMDMs were seeded at a density of $1 \times 10^6$ cells per well of six-well plate after 5 days culture of bone marrow cells. Lentiviral supernatant carrying control, mouse *Id3* gene, mouse shMYC, shHES1, shE2A, shELK1, scramble (Santa Cruz) and 10 µg ml[−1] polybrene (Millipore-Sigma) were added to BMDM culture medium for 12 h. The medium was then replaced and, 48 h later, transduced cells were

selected with 2 µg ml[−1] puromycin for 4 days. The transduced BMDMs were analysed by FACS, RT–qPCR, time-lapse imaging or in vivo rescue experiments.

For lentiviral transduction of KCs, KCs were seeded at a density of $8 \times 10^5$ cells per well of six-well plates in the presence of 20 ng ml[−1] M-CSF and transduced with lentiviral supernatant carrying control, mouse shE2A, shELK1, VPX supernatant (pSIV3-VPX plasmids, a gift by M. Ménager) and 10 µg ml[−1] polybrene for 12 h. The medium was then replaced and, 48 h later, transduced cells were selected with 1 µg ml[−1] puromycin for 3 days. The transduced KCs were analysed by FACS.

For Lentiviral transduction of hiPSC-Macs, hiPSC-Macs were seeded at a density of $5 \times 10^5$ cells per well of a six-well plate in the presence of 20 ng ml[−1] M-CSF and transduced with lentiviral supernatant carrying control, human *ID3* gene, VPX supernatant and 10 µg ml[−1] polybrene for 12 h. The medium was then replaced and, 48 h later, transduced cells were selected with 1 µg ml[−1] puromycin for 3–4 days. The transduced hiPSC-Macs were analysed by RT–qPCR and time-lapse imaging.

**Flow cytometry, cell sorting and cell counting. Blood and BM cell preparation.** Mouse blood cells were obtained as follows. Mice were anaesthetized by intraperitoneal injection of ketamine/xylazine/acepromazine anaesthesia cocktail. Blood were collected using the cardiac puncture approach. In brief, mice were placed on their back and the needle of 1 ml syringes was inserted (pretreated with 1 ml 100 mM EDTA buffer) under the rib cage. The plunger was gently pulled to collect blood. Red blood cells were lysed using red blood cell lysis buffer (eBioscience). Mouse bone marrow cells were obtained as follows. The femur, tibia and iliac bones from mice were flushed with PBS, and red blood cells were lysed using red blood cell lysis buffer.

**Tissue cell suspension preparation.** Mice were perfused with 10 ml PBS under terminal anaesthesia, tissue samples were minced into small pieces and incubated in digestion buffer containing 1× PBS, collagenase D (1 mg ml[−1], Sigma-Aldrich), dispase (2.4 mg ml[−1], Thermo Fisher Scientific), DNase (0.2 mg ml[−1], Sigma-Aldrich) and 3% heat-inactivated fetal bovine serum (FBS, Invitrogen) for 30 min at 37 °C. Cells suspensions were dissociated and passed through a 100 µm cell strainer (BD) and resuspended in 50 µl of blocking buffer containing 1× PBS, 0.5% BSA, 2 mM EDTA, anti-mouse CD16/32 (1:100), 5% normal rat, 5% normal mouse and 5% normal rabbit serum (Jackson ImmunoResearch) for 15 min at 4 °C. The samples were stained with the indicated antibodies (a list of which is provided in Supplementary Data 5; 1:200) for 30 min at 4 °C. Flow cytometry was performed using the BD Biosciences LSR Fortessa flow cytometer with Diva software. All data were analysed using FlowJo v.10.6 (Tree Star).

**Staining and gating strategies.** Mouse liver macrophage/myeloid cell panels were as follows: Pop1 macrophages, Hoechst[−]CD45[+]CD3[−]CD19[−]Nkp46[−]Ly6G[−]F4/80[+]TIM4[+]; Pop2 macrophages, Hoechst[−]CD45[+]CD3[−]CD19[−]NKP46[−]Ly6G[−]F4/80[+]TIM4[−]MHCII[+]; Pop3 myeloid cells, Hoechst[−]CD45[+]CD3[−]CD19[−]NKP46[−]Ly6G[−]F4/80[−]TIM4[−]MHCII[−]. KC subsets: KC subset 1, Hoechst[−]CD45[+]CD3[−]CD19[−]NKP46[−]Ly6G[−]F4/80[+]TIM4[+]CD206[+]; KC subset 2, Hoechst[−]CD45[+]CD3[−]CD19[−]NKP46[−]Ly6G[−]F4/80[+]TIM4[+]CD206[high]. Other mouse macrophage panels were as follows: kidney macrophages, Hoechst[−]CD45[+]CD3[−]CD19[−]NKP46[−]CD11b[low]F4/80[bright]. Brain macrophages, Hoechst[−]CD45[+]CD3[−]CD19[−]NKP46[−]CD11b[+]F4/80[+]. Lung alveolar macrophages, Hoechst[−]CD45[+]CD11b[−]CD11c[+]CD64[+]SIGLECF[+]. Lung interstitial macrophages, Hoechst[−]CD45[+]CD11b[+]Ly6G[−]CD64[+]. Skin macrophages, Hoechst[−]CD45[+]CD3[−]CD19[−]NKP46[−]CD11b[+]F4/80[+]. Spleen RPM, Hoechst[−]CD45[+]CD3[−]CD19[−]NKP46[−]CD11b[low]F4/80[+]. Pancreas macrophages, Hoechst[−]CD45[+]CD3[−]CD19[−]NKP46[−]F4/80[+]. Mouse myeloid cells panels were as follows: cDC1, Hoechst[−]CD45[+]F4/80[−]Ly6G[−]CD11b[−]CD11c[+]MHCII[+]. cDC2, Hoechst[−]CD45[+]F4/80[−]Ly6G[−]CD11b[+]CD11c[+]MHCII[+]. Ly6C[+] monocytes (blood), Hoechst[−]CD3[−]CD19[−]NKP46[−]CD11b[+]CD115[+]Ly6C[+]. Ly6C[+] monocytes (spleen, liver), Hoechst[−]CD45[+]CD3[−]CD19[−]NKP46[−]F4/80[−]Ly6G[−]CD11b[+]Ly6C[high]. Ly6G[+] granulocytes

(blood), Hoechst⁻CD3⁻CD19⁻NKP46⁻Ly6G⁺. Ly6G⁺ granulocytes (spleen, liver), Hoechst⁻CD45⁺CD3⁻CD19⁻F4/80⁻Nkp46⁻F4/80⁻Ly6G⁺. Mouse lymphocyte panels were as follows: NKT cells (liver), Hoechst⁻CD45⁺F4/80⁻TCRβ⁺CD1dTetramers⁺; γδT cells (liver), Hoechst⁻CD45⁺F4/80⁻CD3⁺TCRβ⁻TCRγδ⁺; CD3⁺ T cells (spleen, liver), Hoechst⁻CD45⁺F4/80⁻CD3⁺. CD8⁺ T cells (spleen, liver), Hoechst⁻CD45⁺F4/80⁻CD3⁺CD8⁺; CD4⁺ T cells (spleen, liver), Hoechst⁻CD45⁺F4/80⁻CD3⁺CD4⁺; CD19⁺ cells (spleen, liver), Hoechst⁻CD45⁺F4/80⁻CD3⁻CD19⁺; NKP46⁺ cells (spleen, liver), Hoechst⁻CD45⁺F4/80⁻CD3⁻NKP46⁺. CD3⁺ T cells (blood), Hoechst⁻Ly6G⁻NKP46⁻CD19⁻CD3⁺. CD8⁺ T cells (blood), Hoechst⁻Ly6G⁻Nkp46⁻CD19⁻CD3⁺CD8⁺; CD4⁺ T cells (blood), Hoechst⁻Ly6G⁻NKP46⁻CD19⁻CD3⁺CD4⁺; CD19⁺ cells (blood), Hoechst⁻Ly6G⁻NKP46⁻CD19⁺; NKP46⁺ cells (blood), Hoechst⁻Ly6G⁻CD19⁻CD3⁻Nkp46⁺. Mouse bone marrow panels were as follows: long-term HSCs (LT-HSCs), Hoechst⁻CD3⁻CD19⁻NKP46⁻Ly6G⁻Kit⁺SCA1⁺CD150⁺CD48⁻; short-term HSCs (ST-HSCs), Hoechst⁻CD3⁻CD19⁻NKP46⁻Ly6G⁻Kit⁺SCA1⁺CD150⁻CD48⁻; multipotent progenitors (MPPs), Hoechst⁻CD3⁻CD19⁻NKP46⁻Ly6G⁻Kit⁺SCA1⁺CD150⁻CD48⁺. Human lymphocyte intracellular staining panels were as follows: CD8⁺ T cells, Hoechst⁻CD8⁺IFNγ⁺TNF⁺. CD56⁺ NK cells, Hoechst⁻CD56⁺IFNγ⁺. See Supplementary Figs. 1–3 and Supplementary Tables 1 and 2.

**Cell counting.** Cells number was assessed using a cell counter (GUAVA easyCyte HT).

**Cell sorting.** Cell sorting was performed using the Aria III BD cell sorter. Single live cells were gated on DAPI⁻ and using forward scatter width (FSC-W) and FSC-A to exclude doublets.

**Cytokine intracellular staining.** For IFNγ and TNF intracellular staining in CD8⁺ and CD8⁻ T cells, mouse liver cell suspension, and human CD8⁺ T cells were treated with a cocktail of phorbol 12-myristate 13-acetate (PMA), ionomycin, brefeldin A and monensin (Thermo Fisher Scientific) for 4–12 h. Staining for IFNγ and TNF was performed using the eBioscience Transcription Factor Staining kit (Thermo Fisher Scientific) according to the manufacturer's instructions. IFNγ and TNF production were analysed by flow cytometry.

IFNγ intracellular staining of mouse splenic NK cells and human CD56⁺ NK cells was performed using the eBioscience Transcription Factor Staining kit according to the manufacturer's instructions. IFNγ production was analysed using flow cytometry.

**Lineage tracing of bone-marrow-derived cells using genetic labelling.** Bone-marrow-derived cells are labelled in *Cxcr4*^gfp/+^and *Cx3cr1*^gfp/+^ mice, and by a single injection of 4-OH TAM (37.5 mg per kg body weight) supplemented with progesterone (18.75 mg per kg body weight) in 6-week-old *Cxcr4*^creERT2^*Rosa26*^LSL-tdT^mice (ref. 65) (Extended Data Fig. 2k,o). *Cxcr4*^gfp/+^ and *Cx3cr1*^gfp/+^ mice, and *Cxcr4*^creERT2^*Rosa26*^LSL-tdT^ mice 2 weeks after 4-OH TAM injection, were injected with 1 × 10⁶ KPC-1 cells through the portal vein and euthanized 2 weeks later for the analysis of tdT⁺ cells, YFP⁺ cells or GFP⁺ cells among liver macrophage subsets.

**Parabiosis. Generation of CD45.2 *Id3*^+/+^/CD45.1 and CD45.2 *Id3*^−/−^/CD45.1 parabionts.** Female 6–8-week-old CD45.2 *Id3*^+/+^ and CD45.2 *Id3*^−/−^ host parabionts were generated with age- and weight-matched female CD45.1 mice through parabiosis surgery described previously[84]. Mice were maintained on a trimethoprim/sulfamethoxazole diet after surgery to minimize infection. After 8 weeks, parabiotic mice were separated and perfused with phosphate-buffered saline (PBS). Partner-derived CD45.1⁺ cells in TIM4⁺ KCs were determined by flow cytometry analysis.

**Generation of CD45.2/CD45.1 parabionts.** Female 6–8-week-old congenic CD45.1 and CD45.2 mice were connected through parabiosis surgery. After 8 weeks, CD45.2 parabionts were injected with 1 × 10⁶ KPC-1-luci-tdT cells through the portal vein to induce liver metastasis. Liver samples were collected 2 weeks after tumour injection. Flow cytometry and immunofluorescence imaging were performed to analyse liver macrophage exchange ratios.

**RT–qPCR.** Mouse KCs and BMDMs and human iPSC-macrophages were lysed directly on tissue culture plates and RNA was extracted using the quick-RNA Microprep kit (Zymo research; R1050) in accordance with manufacturer's instructions. cDNA preparation was performed using Quantitect Reverse transcription kit (Qiagen; 205313) according to the manufacturer's protocol. RT–qPCR was performed on the Quant Studio 6 Flex System with 10 ng cDNA per reaction using probes (Supplementary Data 5) and TaqMan Fast Advance Mastermix (Thermo Fisher Scientific; 4444557), or PowerUp SYBR Green Master Mix (Thermo Fisher Scientific; A25742), according to the manufacturer's instructions. Expression values for each tested gene relative to a *GAPDH* endogenous control were calculated using the $\Delta\Delta C_t$ method according to the formula: $2^{-(C_{t\text{test gene}} - C_{t\text{GAPDH}})}$.

**Cytology.** Cytospin preparation was performed using Cytospin 3 (Thermo Fisher Scientific, Shandon) and Cytofunnels (Thermo Fisher Scientific, BMP-CYTO-DB25) by centrifuging sorted cells onto Super-frost slides (Thermo Fisher Scientific, 12-550-15) at 800 rpm for 10 min (medium acceleration). The slides were air-dried for at least 30 min and fixed for 10 min in 100% methanol (Thermo Fisher Scientific, A412SK-4). Methanol-fixed cells were stained in 50% May–Grünwald solution (Sigma-Aldrich, MG500-500mL) for 5 min, 5% Giemsa (Sigma-Aldrich, 48900-500mL-F) for 15 min and washed with Sorensons buffered distilled water (pH 6.8) three times for 2 min after each staining. Slides were mounted with Entellan New (Millipore, 1079610100) after air-drying, representative pictures were taken using an Axio Lab. A1 microscope (Zeiss) under a N-Achroplan ×100/01.25 objective.

**Bioluminescence imaging.** Depending on the experiment, bioluminescence imaging was conducted either on isolated organs (ex vivo) in long-term orthotopic pancreatic tumour experiments, or in anaesthetized mice (in vivo) in short-term models (below) on the In ViVo Imaging System spectrum (Perkin Elmer). Quantification of bioluminescence images was performed using LivingImage (v.2.60.1; Perkin Elmer). Photoradiance was measured as photons s⁻¹ cm⁻² sr⁻¹.

**Immunofluorescence and whole-mount imaging. Immunofluorescence imaging of mouse liver.** Mice were perfused with 10 ml PBS, liver samples were dissected and fixed overnight at 4 °C with PLP fixative in phosphate buffer[84]. After the PBS wash, livers were dehydrated in 30% sucrose in PBS and embedded in OCT. Cryoblocks were cut at a thickness of 16 μm and blocked with PBS containing 5% normal goat serum (Jackson ImmunoResearch), 1% BSA and 0.3% Triton X-100 (Sigma-Aldrich) for 1 h at room temperature. The samples were incubated with anti-mouse-F4/80-eF450/AF647/AF488/eF570 (1:200, BM8, eBioscience), anti-mouse TIM4-AF647/PE (1:200, RMT4-54, BioLegend), anti-mouse CD45.1-AF488 (1:200, A20, BioLegend), chicken-anti-GFP (1:500, A10262, Invitrogen, recognize YFP), rabbit-anti-RFP (1:200, 600-401-379, Rockland), goat-anti mouse CLEC4F (1:200, AF2784, R&D Systems), anti-mouse CCL3 (1:200, 50-7532-82, Thermo Fisher Scientific), anti-mouse CCL4 (1:200, AF-451-NA, Thermo Fisher Scientific), anti-mouse CCL5 (1:200, 701030, Thermo Fisher Scientific), anti-mouse IL-12P70 (1:200, MM121B, Thermo Fisher Scientific), anti-mouse IL-15 (1:200, AF447-SP, Thermo Fisher Scientific), anti-mouse IL-18 (1:200, PA5-79481, Thermo Fisher Scientific) antibodies for 2 h at room temperature. Secondary antibody staining using anti-chicken-Alexa Fluor 488 (1:500; a11039, Thermo Fisher Scientific), anti-rabbit-Alexa Fluor 555 (1:500, A32794, Thermo Fisher Scientific), anti-rabbit-Alexa Fluor 647 (1:500, A32795, Thermo Fisher Scientific), anti-goat-Alexa Fluor 555 (1:500, a32816, Thermo Fisher Scientific), anti-goat-Alexa Fluor 647 (1:500, a21447, Thermo Fisher Scientific), anti-goat-Alexa Fluor 488 (1:500, a32814, Thermo Fisher Scientific), anti-sheep-Alexa Fluor 568 (1:500, A21099, Thermo Fisher Scientific), Streptavidin-Alexa Fluor 647 (1:500, 405237, BioLegend) were performed for 1 h at room temperature. Nuclei were

counterstained with DAPI (Invitrogen). The sections were mounted with Fluoromount-G (eBiosciences). Images were acquired on a Zeiss LSM880 confocal microscope using an oil-immersion ×40/1.4 NA objective.

**Immunofluorescence imaging of the metastatic liver of human patients with PDAC.** Metastatic liver samples of human patients with PDAC were embedded in OCT. Cryoblocks were cut at a thickness of 16 μm, fixed with 4% paraformaldehyde (PFA) for 30 min, blocked with PBS containing 5% normal goat serum (Jackson ImmunoResearch), 1% BSA and 0.3% Triton X-100 (Sigma-Aldrich) for 1 h.

The samples were incubated with anti-human CD14-AF488 (1:200, BD), sheep-anti-human CK19 (1:200, AF3506, R&D Systems), rabbit-anti-human TIM4 (1:200, PA5-53346, Thermo Fisher Scientific), goat-anti-human IL-12 (1:200, AF-219-NA, R&D Systems), goat-anti-human IL-18 (1:200, AF2548, R&D Systems), mouse-anti-human IL-15 (1:200, MAB2471, R&D Systems), goat-anti-human CCL3 (1:200, AF-270-NA, R&D Systems), goat-anti-human CCL4 (1:200, AF-271-NA, R&D Systems) and goat-anti-human CCL5 (1:200, AF-278-NA, R&D Systems) antibodies for 2 h at room temperature. The samples were then incubated with anti-rabbit-Alexa Fluor 647 (1:500, A32795, Thermo Fisher Scientific), anti-rabbit-Alexa Fluor 555 (1:500, A32794, Thermo Fisher Scientific), anti-sheep-Alexa Fluor 568 (1:500, A21099, Thermo Fisher Scientific), anti-goat-Alexa Fluor 647 (1:500, a21447, Thermo Fisher Scientific) and anti-mouse-Alexa Fluor 555 (1:500, A-31570, Thermo Fisher Scientific) for 1 h. Nuclei were stained with DAPI for 10 min. Sections were mounted with Fluoromount-G (eBiosciences). Images were acquired on a Zeiss LSM880 confocal microscope using an oil-immersion ×40/1.4 NA objective.

**Whole-mount immunofluorescence imaging of mouse liver.** Liver pieces were fixed in 4% PFA diluted in PBS for 30 min at room temperature with agitation. The samples were permeabilized with 1× PBS containing 0.3% Triton X-100, 4% BSA for 1 h at room temperature and incubated with an anti-F4/80-eF450 (1:100, eBioscience), anti-TIM4-AF647 (1:100, BioLegend) antibody mix for 2 h at room temperature. Data were acquired using the LSM880 Zeiss microscope. Imaris v.9.3.1 (Bitplane) was used to analyse the acquired images.

**Quantification of chemokines/cytokines and tumour materials relative MFI in KCs.** Quantification of tdT relative MFI in tdT+ KC lysosomes. Immunofluorescence staining for F4/80, TIM4, LAMP1+ and tdT was performed on frozen liver sections from C57BL/6J mice 2 weeks after intraportal injection of $1 \times 10^6$ KPC-1-tdT cells. Images were acquired on the Zeiss LSM880 confocal microscope. Imaris v.9.3.1 (Bitplane) was used to reconstruct the 3D surface of LAMP1+ lysosome or 97.5 μm² non-LAMP1− region in tdT+ KCs. The tdT relative mean fluorescence intensity (MFI) in LAMP1+ lysosomes was determined by normalizing to a non-LAMP1− region MFI of 1.

**Quantification of chemokine/cytokine relative MFI in mouse KCs.** Immunofluorescence staining for CCL3, CCL4, CCL5, IL-12P70, IL-15, IL-18, GFP, F4/80 and TIM4 was performed on frozen liver sections from C57BL/6J mice, *Clec4f$^{cre}$Id3$^{f/f}$* mice and *Id3$^{f/f}$* littermates 2 weeks after intraportal injection of $1 \times 10^6$ KPC-1-GFP cells. Images were acquired on the Zeiss LSM880 confocal microscope. Imaris v.9.3.1 (Bitplane) was used to reconstruct the 3D surface of F4/80+TIM4+ KCs. Chemokine/cytokine relative MFI in F4/80+TIM4+ KCs was determined by normalizing to an MFI of 1 in the tumour-free region (distance from tumour > 50 μm) in C57BL/6J mice or *Id3$^{f/f}$* mice.

Quantification of chemokine/cytokine and CK19+ tumour material relative MFI in human KCs. Immunofluorescence staining for CCL3, CCL4, CCL5, IL-12, IL-15, IL-18, CK19, CD14 and TIM4 on frozen liver sections from human patients with PDAC metastatic liver. Images were acquired on the Zeiss LSM880 confocal microscope. Imaris v.9.3.1 (Bitplane) was used to reconstruct the 3D surface of CD14+TIM4+ KCs. The chemokine/cytokine and CK19+ tumour material relative MFI in CD14+TIM4+ KCs was determined by normalizing to a background MFI of 1.

**Tumour growth, metastasis and rescue models. Endogenous KPC model.** KPC mice heterozygous for *p48$^{cre}$*, *Trp53$^{LSL-R172H}$* and *Kras$^{LSL-G12D}$* alleles[71] were generated by crossing *p48$^{cre}$* (ref. 67), *Trp53$^{LSL-R172H}$* (ref. 68) and *Kras$^{LSL-G12D}$* (ref. 69) mice under SPF conditions. KPC mice were monitored on a regular basis to check for symptoms of abdominal distension; moribund animals were euthanized by $CO_2$ asphyxiation according to IACUC guidelines. Mice were euthanized at 6 months and livers were collected and fixed as described above. Quantification of CK19+ tumour material, chemokine and cytokine relative MFI was performed as follows. Frozen liver sections from KPC mice and control (*Kras$^{LSL-G12D}$Trp53$^{LSL-R172H}$*) mice were subjected to immunofluorescence staining for CCL3, CCL4, CCL5, IL-12P70, IL-15, IL-18, CK19, F4/80 and TIM4. Images were acquired on the Zeiss LSM880 confocal microscope. Imaris v.9.3.1 (Bitplane) was used to reconstruct the 3D surface of F4/80+TIM4+ KCs. CK19+ tumour material, chemokine and cytokine relative MFI in F4/80+TIM4+ KCs was determined by normalization to *Kras$^{LSL-G12D}$Trp53$^{LSL-R172H}$* mice MFI of 1.

**Long-term orthotopic pancreatic tumour model.** Orthotopic injection of pancreatic cell lines into the pancreas was performed according to a published protocol[85]. Mice were anaesthetized under isoflurane gas, sterile sharp scissors were used to cut a single incision off the abdominal skin and muscle above the pancreas, the pancreas was gently positioned to allow slow injection into the pancreas of $2 \times 10^5$ KPC-2-luci-tdT or $2 \times 10^5$ KPC-2-luci-GFP cells per mouse, resuspended in 50 μl of PBS and Matrigel (354234, corning) at 2/1 ratio using 31 G insulin syringes (BD). The pancreas was gently placed back into the abdominal cavity. The muscle layer was closed using sterile absorbable vicryl suture (J463G, Ethicf on). Skin edges were closed with sterile 9 mm wound clips (Braintree Scientific). Then, 8 weeks after pancreatic orthotopic injection, the mice received retro-orbital injection of 1 mg D-luciferin (Goldbio Technology) in 100 μl sterile water, and the liver, spleen, lungs and pancreas were dissected for ex vivo analysis by bioluminescence imaging. Livers were collected and fixed as described above, and the percentage of tdT+TIM4+ cells in TIM4+ KCs was analysed by whole-mount imaging as described in the 'Whole-mount immunofluorescence imaging of mouse liver' section. (Extended Data Fig. 3f).

**Conditional depletion of KCs by DT.** For conditional depletion of KCs by DT, in the indicated experiments, *Clec4f$^{cre}$Rosa26$^{LSL-DTR}$* and *Rosa26$^{LSL-DTR}$* littermates received weekly intraperitoneal injection of 100 ng DT, starting 1 week after pancreatic orthotopic injection of tumour cells.

**Short-term liver metastasis model (intraportal injection of tumour cell lines).** In total, $1 \times 10^6$ KPC-1-luci-GFP cells, $5 \times 10^5$ B16F10-luci-GFP cells, $1 \times 10^6$ LLC1-luci-GFP cells, $1 \times 10^6$ MC38-luci-GFP cells, or $1 \times 10^6$ KPC-1-luci-tdT cells, $5 \times 10^5$ B16F10-luci-tdT cells, $1 \times 10^6$ LLC1-luci-tdT cells or $1 \times 10^6$ Pan02-luci-tdT cells were resuspended in 50 μl PBS. Mice were anaesthetized with isoflurane gas and sterile sharp scissors were used to cut a single incision off the abdominal skin and muscle. While holding the median side of the incision aside with forceps, including the skin and peritoneal lining, a sterile cotton swab was used to carefully pull the large and small intestines out until the portal vein is visualized. After covering the intestines with the sterile gauze soaked in sterile PBS, a 31 G needle (BD) loaded with tumour cells was inserted into the portal vein below the liver and the full volume (50 μl) was slowly injected. The needle was then removed while simultaneously placing a sterile cotton tip applicator on the vein with pressure, for 5 min, to keep the injection site intact. The internal organs were gently placed back into the abdominal cavity. The muscle layer was closed using sterile absorbable vicryl suture (J463G, Ethicon). Skin edges were closed with sterile 9 mm wound clips (Braintree Scientific). Tumour-bearing mice were analysed as follows.

**Conditional depletion of KCs by DT.** In the indicated experiments, *Clec4f$^{cre}$Rosa26$^{LSL-DTR}$* and *Rosa26$^{LSL-DTR}$* littermates received

intraperitoneal injection of 100 ng DT before and/or after the tumour cell grafts as described above.

**Survival experiments.** In the indicated experiments, the cohort of tumour-bearing mice was examined by a veterinarian twice a week for 5 weeks. Moribund animals were determined by the veterinarian and euthanized by $CO_2$ asphyxiation according to IACUC guidelines. Comparison of survival curves was performed using log-rank (Mantel–Cox) tests (Figs. 1g, 3j and 6g).

**Liver tumour burden at 24 h.** In the indicated experiments, CD45$^-$GFP$^+$ or CD45$^-$tdT$^+$ liver tumour cell numbers in tumour-bearing mice were analysed by flow cytometry. The percentage of GFP$^+$TIM4$^+$ cells or the percentage of tdT$^+$TIM4$^+$ cells in KCs was analysed by immunofluorescence staining.

**Liver tumour burden at 2 weeks.** In the indicated experiments, the liver tumour burden at 2 weeks was assessed by in vivo bioluminescence imaging. Chemokine and cytokine production by KCs was analysed using RT–qPCR and immunofluorescence staining, and the percentage of GFP$^+$TIM4$^+$ cells or the percentage of tdT$^+$TIM4$^+$ cells in KCs was analysed by immunofluorescence staining. The numbers of immune cells were analysed using flow cytometry and immunofluorescence staining. The production of IFNγ and TNF by NK cells or CD8 T cells was analysed using flow cytometry.

**Rescue of KPC liver metastasis in *Clec4f$^{cre}$Id3$^{f/f}$* mice by BMDMs.** *Clec4f$^{cre}$Id3$^{f/f}$* mice (aged 6–12 weeks) received $1 \times 10^6$ KPC-1-luci-GFP cells by intraportal injection, followed after 1 week (day 7 after tumour injection) by intraportal injection of either $1 \times 10^6$ BMDMs expressing lenti-control or lenti-mId3 cells. The tumour burden was performed 14 days after tumour injection by bioluminescent images described above.

**Rescue of LLC liver metastasis in wild-type mice by BMDMs.** C57BL/6J mice (aged 6–8 weeks) received $1 \times 10^6$ LLC1-luci cells by intraportal injection, followed after one week (day 7 after tumour injection) by intraportal injection of $1 \times 10^6$ BMDMs expressing lenti-control, lenti-mId3 cells or not. The tumour burden was analysed 14 days after tumour injection by bioluminescence imaging as described above. Survival was analysed as described above. Comparison of survival curves was performed using log-rank (Mantel–Cox) tests.

**B16F10 melanoma subcutaneous tumours and rescue by BMDMs.** C57/BL6J mice (aged 6–12 weeks) received subcutaneous injection of $1 \times 10^6$ B16F10-luci-GFP cells, into the left and right flank, followed by intratumour injection of $5 \times 10^5$ BMDMs expressing lenti-control, lenti-mId3 cells at day 7 after tumour injection. Then, 7 days later, the tumour burden was assessed by in vivo bioluminescence imaging as described above. The numbers of immune cells and the production of IFNγ and TNF by NK cells or CD8$^+$ T cells were analysed using flow cytometry.

**Intravital imaging of liver KCs and KPC-1-mtdT tumour cells in vivo.** C57BL/6J mice (aged 6–12 weeks) were injected with $1 \times 10^6$ KPC-1-mtdT cells through intraportal injection as described above. Then, 2 weeks after tumour cell injection, the mice anaesthetized under isoflurane gas and anaesthesia was maintained through continuous inhalation of Isoflurane (0.5 l min$^{-1}$) in oxygen through a nose cone. The mice were then injected retro-orbitally with 5 µl CellEvent caspase-3/7-green reagent (Invitrogen), a four-amino-acid peptide (DEVD) caspase-3/7 cleavage reporter conjugated to a nucleic-acid-binding dye that becomes fluorescent when bound to DNA (Cas-Green) to monitor tumour cell apoptosis and death[86], and 10 µl of anti-TIM4-AF647 antibodies (BioLegend) in 50 µl PBS to label apoptotic/dead cells and KCs, respectively. Sterile eye lubricant was applied to both eyes to prevent corneal drying during the experiment. A 1.5 cm horizontal incision was cut off the skin and muscle above the liver, and the liver left lobe was extruded gently. The mouse was then inverted and positioned on

a custom-made aluminium tray stage inserted through circular 2.5 cm diameter hole, covered with a glass coverslip that was attached with silicone grease. PBS-soaked sheets of paper were prepositioned on the cover slip to surround the area of the exposed liver, then the mouse was ready for intravital imaging. During the whole imaging period, PBS was gently added every 20 min on both sides of the mouse to keep the area moist. A thermostat-controlled heated chamber keeps the whole microscope, mice, tray and microscope objectives at 32 °C to prevent hypothermia during the experiment. Imaging was performed using the Zeiss LSM880 confocal laser scanning microscope. Acquisition of CellEvent caspase-3/7-green, tdT and AF647 fluorescence signals was performed in line in a single channel. The power used for each laser line was as follows; 1%, 488 nm; 1%, 568 nm; and 5%, 647 nm—the lowest required to obtain a sufficient signal for each fluorescent probe and chosen to minimize phototoxicity. Seven consecutive stacks with an interval of 2.5 µm were captured using a Zeiss Plan-Apochromat ×20/0.75 objective, with digital zoom set to 1, every 1 min per position for up to 8 h. Time-lapse videos and 3D surface reconstructions were generated using Imaris v.9.3.1 (Bitplane).

**In vitro mouse coculture assays. Engulfment assay: ex vivo 3D co-culture and time-lapse imaging of KCs and KPC-1-mtdT tumour cells.** F4/80$^+$TIM4$^+$ KCs were sorted from freshly isolated liver of C57BL/6J or *Id3$^{-/-}$* mice using digestion buffer and the antibody panel described in the 'Flow cytometry, cell sorting and cell counting' section. In total, $1 \times 10^4$ KCs were mixed with $2 \times 10^3$ KPC-1-mtdT cells (5:1 ratio), embedded in growth-factor-reduced Matrigel (356231, Corning) and cultured overnight in a 24-well µ-plate (Ibidi) with DMEM medium in the presence of 20 ng ml$^{-1}$ M-CSF and, when indicated, with 1 µg D89E, 60 mm latrunculin A, 50 µg ml$^{-1}$ anti-SIRPA (P84, BioXcell) or 20 µg ml$^{-1}$ anti-dectin-1 (R1-8g7, InvivoGen). Before imaging, the co-cultures were stained with 2 µM CellEvent caspase-3/7-green reagent (Invitrogen) and anti-F4/80-AF647 antibodies (1:200, BioLegend) for 30 min. Imaging was performed using the Zeiss LSM880 confocal laser-scanning microscope equipped with an imaging chamber maintained at 37 °C, 5% $CO_2$, 20% $O_2$ and 90% relative humidity. Five consecutive stacks at an interval of 2.5 µm were captured using the Zeiss LD C-Apochromat ×40/1.1 water-immersion objective ($x = 212.55$ µm, $y = 212.55$ µm, $z = 15$ µm) every 5 min per position for 20 h. The data were analysed using Imaris v.9.3.1 (Bitplane). For each sample, the percentage of engulfing KCs was determined by averaging the percentage of F4/80$^+$ cells engulfing live tumour cells (CellEvent, cleavage caspase-3/7$^-$ tumour cells) from at least three simultaneously imaged fields of view. Time to engulfment values were determined as the time from stable interaction between macrophages and tumour cells to tumour cell engulfment. Time to Cas cleavage was determined as the time from stable interaction between macrophages and tumour cells to the detection of caspase-3/7 cleavage. The total *n* numbers of macrophages tracked per each sample are indicated.

**Engulfment assay: ex vivo 3D co-culture and time-lapse imaging of BMDMs and KPC-1-mtdT tumour cells.** For BMDM/KPC-1-mtdT tumour cell time-lapse imaging, $1 \times 10^4$ BMDMs expressing lenti-control or lenti-mouse-*Id3* were mixed with $2 \times 10^3$ KPC-1-mtdT cells (5:1 ratio), embedded in growth-factor-reduced Matrigel (356231, Corning) and cultured overnight in a 24-well µ-plate (Ibidi) with DMEM medium in the presence of 20 ng ml$^{-1}$ M-CSF and, when indicated, with 50 µg ml$^{-1}$ anti-SIRPA (P84, BioXcell). before imaging, the co-cultures were stained with 2 µM CellEvent caspase-3/7-green reagent (Invitrogen), anti-F4/80-AF647 antibodies (1:200, BioLegend) for 30 min. Imaging was performed using the Zeiss LSM880 confocal laser-scanning microscope and analysed as described in the 'Engulfment assay: ex vivo 3D co-culture and time-lapse imaging of KCs and KPC-1-mtdT tumour cells' section.

**Production of chemokines and cytokines by KCs in a coculture assay with tumour cells.** In total, $3 \times 10^5$ KCs from *Id3$^{f/f}$* or *Clec4f$^{cre}$Id3$^{f/f}$* mice

were seeded in a 12-well plate with $1.5 \times 10^5$ KPC tumour cells or not. Then, 48 h later, the supernatants from the coculture were collected for the following study. The production of chemokines and cytokines by KCs was analysed using RT−qPCR described as above.

**Role of supernatants in lymphoid cell activation. Mouse NK cell/ supernatant coculture assay.** Mouse splenic NK cells from C57BL/6J mice were seeded in a 96-well round-bottom plate at $3 \times 10^4$ cells per well in 100 μl NK culture medium (RPMI supplemented with 10% FBS, 100 U ml$^{-1}$ penicillin, 100 μg ml$^{-1}$ streptomycin and 20 ng ml$^{-1}$ mIL-2) in the presence of 100 μl of the above-mentioned supernatant. Then, 3 days later, IFNγ production by NK cells was analysed using flow cytometry.

**In vitro human coculture assays. Engulfment assay: ex vivo 3D co-culture and time-lapse imaging of hiPSC-Macs and PANC-1-mtdT tumour cells.** For hiPSC-Mac/PANC-1-mtdT tumour cell time-lapse imaging, a total of $1 \times 10^4$ hiPSC-Macs expressing lenti-control or lenti-human-*ID3* were mixed with $2 \times 10^3$ PANC-1-mtdT cells (5:1 ratio), embedded in growth-factor-reduced Matrigel (356231, Corning) and cultured overnight in a 24-well μ-plate (Ibidi) with RPMI1640 medium in the presence of 100 ng ml$^{-1}$ M-CSF. Before imaging, the co-cultures were stained with 2 μM CellEvent caspase-3/7-green reagent (Invitrogen), anti-CD14-AF647 antibodies (1:200, BioLegend) for 30 min. Imaging was performed using the Zeiss LSM880 confocal laser-scanning microscope and analysed as described in the 'Engulfment assay: ex vivo 3D co-culture and time-lapse imaging of KCs and KPC-1-mtdT tumour cells' section.

**Production of chemokines, cytokines by human macrophage in coculture assay with tumour cells.** In total, $10^5$ hiPSC-Macs expressing lenti-control or lenti-human-*Id3* were seeded in a 12-well plate, and treated or not with $5 \times 10^4$ PANC-1 tumour cells. Then, 48 h later, the coculture supernatants were collected for the following study. The production of chemokines and cytokines by hiPSC-Macs was analysed using RT−qPCR described as above.

**Role of supernatants in lymphoid cell activation.** For the human NK cell/supernatant coculture assay, human NK cells were seeded in a 96-well round-bottom plate at $10^4$ cells per well in 100 μl NK culture medium (see above) in the presence of 100 μl of the above-mentioned supernatant. Then, 3 days later, IFNγ production were analysed by flow cytometry. For the human CD8 T cell/supernatant coculture assay, human CD8$^+$ T cells, stained with CFSE, were seeded in a 96-well round-bottom plate at $10^4$ cells per well in 100 μl CD8 culture medium (see above) in the presence of 100 μl of the above-mentioned supernatant, then PBS washed anti-hCD3/hCD28 activation beads were added to the medium. Then, 3 days later, the samples were treated with a cocktail of PMA, ionomycin, brefeldin A and monensin for 6 h. CFSE proliferation, TNF and IFNγ production by CD8 T cells were analysed using flow cytometry.

**Identification of candidate functional E2A and ELK1 motifs.** The position weight matrix (PWM) of E2A/TCF3 and ELK1 motifs was downloaded from the JASPAR database with motif IDs MA0522.1 and MA0028.2, respectively[87]. To find motif matches, we first computed a motif score or PWM score for all of the putative regulatory elements of KCs at the *Sirpa* locus and filtered with a minimum PWM score cut-off that passed a false-positive rate of <0.2%. We then computed the DeepLIFT scores based on a deep learning model (see below) at every regulatory element and overlaid these scores with motif matches for E2A and ELK1 to predict functional motifs. Our final set of functional motifs all have a PWM score exceeding the score cut-off and at least three positions within top 20% based on DeepLIFT scores.

**Training and interpretation of the deep learning model.** The deep learning model was trained and interpreted as described previously[88].

In brief, we adapted a strategy of AgentBind[89] and fine-tuned a pre-trained DeepSEA model[90] using all active enhancers in KCs on the basis of previously published ATAC−seq and H3K27ac ChIP−seq data under Gene Expression Omnibus (GEO) GSE128338 (ref. 91). The AgentBind model consists of (1) pretraining convolutional neural networks, which infer important sequence context features and learn combinations and orientations of these features that are predictive of binding, using ChIP−seq and DNase-I-sequencing profiles collected from ENCODE18 and the Epigenomics Roadmap Project20 across dozens of cell types; and (2) fine-tuning an individual model for each transcription factor to identify bound versus unbound sequences, described previously[38]. The DeepSEA model (deep learning−based sequence analyzer) is a fully sequence-based algorithmic framework for non-coding-variant effect prediction, described previously[39]. The software used for this methodology was as follows: Python (v.3), Keras (v.2.3.1), tensorflow (v.2.1.0), scikit-learn (v.0.21.3), deeplift (v.0.6.10.0) and biopython (v.1.76). Training data were prepared as follows. Positive data labelled as 1 were 300 bp sequences of ATAC−seq peaks associated with strong levels of H3K27ac. We first obtained the processed data file from GEO GSE128338, which includes the reproducible ATAC−seq peaks merged from KCs of both healthy and NASH-diet mice and their tag counts of H3K27ac ChIP−seq in the expanded 2,000 bp regions[91]. We removed sex chromosomes and filtered the peaks with a minimum cut-off of 32 tags of H3K27ac ChIP−seq. The positive sequences were balanced with the same number of 300 bp negative sequences, which were GC-content-matched random genomic regions selected from the mm10 genome and were labelled as 0. During the training, we left out sequences on chromosome 8 for cross-validation and those on chromosome 9 for testing. The final model had an area under the receiver operating characteristic curve (auROC) equal to 0.828 on the testing data. We next used DeepLIFT[92] to generate importance scores with single-nucleotide resolution using uniform nucleotide backgrounds. For each input sequence, we generated two sets of scores, one for the original sequence and the other for its reverse complement. The final scores were the absolute maximum at each aligned position. We defined predicted functional nucleotides by the top 20% (that is, top 60) positions within each input 300 bp sequence.

**Experimental analysis of candidates E2A and ELK1 binding motifs using CUT&RUN.** CUT&RUN was performed using the Epicypher (14-1048) kit according to the manufacturer's protocol with modifications. A total of $2 \times 10^5$ TIM4$^+$ KCs was sorted from the liver and resuspended in 1 ml nucleus isolation buffer (0.5 mM Tris, pH 8.0, 0.5 mM EDTA, 5 mM magnesium chloride, 0.1 M sucrose, 0.05% Triton X-100, 1× EDTA-free protease inhibitor (11836170001, Sigma-Aldrich)) and incubated on ice for 10 min. Nuclei were resuspended in 100 μl wash buffer, DNA was purified using the QIAamp DNA Micro Kit, using 5 μl of the sample as the input. The rest of the sample was mixed with 10 μl concanavalin A beads and rotated at room temperature for 30 min. The supernatant was removed by placing beads and nuclei mixture on a magnetic stand. Nuclei were resuspended in 50 μl antibody buffer mixed with 3 μl rabbit-anti-E2A (gift from the K. Murre laboratory, made by D. Wiest), or 4 μl rabbit-anti-Elk1 (Cell Signaling Technology, 9182 S) and incubated overnight at 4 °C. The next morning, nuclei were washed in wash buffer twice, resuspended in 50 μl cell permeabilization buffer containing 2.5 μl pAG-MNase and incubated for 10 min at room temperature. After two washes, 1 μl of chromatin digest additive was added, and the samples were incubated at 4 °C for 2 h with rotation. After addition of 33 μl of stop buffer, the samples were incubated for 10 min at 37 °C. The tubes were then placed onto a magnetic stand and the supernatant containing enriched DNA was transferred to 1.5 ml tubes. DNA was purified using the Chip DNA Clean&Concentrator kit (Zymo research). CUT&RUN-enriched DNA and input DNA were analysed by qPCR on the QuantStudio (TM) 6 Flex System (Applied Biosystems)

with the Power SYBR Green PCR Master Mix (Thermo Fisher Scientific, A25742) and calculated as the percentage of input.

**Bulk RNA-seq analysis.** In total, 80,000 KCs per sample were FACS-sorted into 1.5 ml Eppendorf tubes with 800 µl TRIzol LS Reagent (Thermo Fisher Scientific, 15596018) or in 1.5 ml Eppendorf tubes precoated with 10% BSA. RNA samples were submitted to the Integrated Genomics Operation (IGO) at MSKCC for quality and quantity analysis, library preparation and sequencing. In brief, phase separation in cells lysed in TRIzol Reagent was induced with chloroform. RNA was precipitated with isopropanol and linear acrylamide and washed with 75% ethanol. The samples were resuspended in RNase-free water. After RiboGreen quantification and quality control using the Agilent BioAnalyzer, 2 ng total RNA with RNA integrity numbers ranging from 9.4 to 10 underwent amplification using the SMART-Seq v4 Ultra Low Input RNA Kit (Clonetech, 63488), with 12 cycles of amplification. Subsequently, 10 ng of amplified cDNA was used to prepare libraries with the KAPA Hyper Prep Kit (Kapa Biosystems KK8504) using 8 cycles of PCR. The samples were barcoded and run on the HiSeq 4000 system in a paired-end 100 bp run, using the HiSeq 3000/4000 SBS Kit (Illumina). For RNA-seq data processing and analysis, sequenced reads from the RNA-seq were aligned to the mouse reference genome GRCm39 or mm10 using STAR (v.2.7.10a)[93]. The aligned reads were quantified as gene counts using HTSeq[94] with GENCODE release M30[95]. DESeq2[96] was applied to the gene counts table to identify differentially expressed genes (DEGs). Adjusted $P$ values were determined with DEseq2 using the Benjamini–Hochberg method for multiple comparisons with two-sided tests. DEGs were ranked on the basis of their $\log_2$-transformed fold change and associated $P$ values ($P_{adj} < 0.05$). For gene set enrichment analysis, pathways enriched in the ranked DEGs were identified against the mouse Molecular Signatures Database (MSigDB)[97] pathway collection ($P_{adj} < 0.25$) using the fgsea package in R, and the most biologically informative lists are shown.

**scRNA-seq analysis.** The CRC dataset (GSE146409)[98] contained three patients with colorectal liver metastasis and a non-tumour individual. The PDAC dataset (GSE205013)[99] contained three patients with PDAC liver metastasis. The non-tumour dataset (GSE115469)[51] control contained five non-tumour individuals. scRNA-seq analysis was conducted using the Seurat package (v.4.3.0) in R studio (4.2.0). For each dataset, quality control was performed by retaining cells with nFeature_RNA > 200 but <10,000, and mitochondrial content <10%. The PDAC and the non-tumour datasets were integrated using the SCTransform workflow. First, the two Seurat objects were merged, normalized and scaled with 2,000 features. Subsequently, the SCTransform() function was applied to the merged object with the dataset source regressed out. Linear dimensionality reduction was applied to the SCT assay and the first 50 principal components. Harmony (v.0.1.1) was used to correct the dataset and samples. The clustering analysis was based on the harmonized Seurat object. The first 40 principal components were used in the RunTSNE() and FindNeighbors() functions, whereas the resolution parameter was set to 1.8 in the FindCluster() function. For the other parameters unspecified above, the default values were used in the Seurat workflow. The CRC dataset was preintegrated by the authors, so we performed the analysis using the standard Seurat pipeline. Clusters were visualized in a two-dimensional $t$-distributed stochastic neighbour embedding ($t$-SNE) and were annotated using differential expressed marker genes based on the human protein atlas (https://www.proteinatlas.org/). The expression patterns of characteristic genes were presented in the $t$-SNE plot. Expression data of characteristic genes in KC and TAM clusters were extracted and presented in violin plots using the ggplot2 package (v.3.4.1). Average expression levels in each cluster were labelled on the violin plots. Adjusted $P$ values were obtained with the Seurat FindMarkers() function using Wilcoxon tests and Bonferroni correction based on the total number of genes in the dataset.

**Statistics and reproducibility.** Analysis of bulk RNA-seq and scRNA-seq data is included in the corresponding sections. For other experiments, error bars in graphical data represent mean ± s.d. Statistical significance was determined using two-tailed Student's $t$-tests and ANOVA for normally distributed data, or Mann–Whitney $U$-tests and Kruskal–Wallis tests when data were not normally distributed based on Shapiro–Wilk test or Anderson–Darling test ($P < 0.05$) (Supplementary Tables 1 and 2). Comparison of survival curves was performed using log-rank (Mantel–Cox) tests. $P < 0.05$ was considered to be statistically significant. Statistical analyses were performed using GraphPad Prism. The $n$ values represent biological replicates unless otherwise specified in the legend. Experiments were repeated to ensure the reproducibility of the observations. No statistical methods were used to predetermine sample size. Results were obtained from at least three (Figs. 1a–e, 2a,f,g, 3h,e, 4g, 5e and 6a,h and Extended Data Figs. 1c, 2a–g, 4a,b, 5b,d,e, 6c–h, 7b–d,f, 8b and 10a,c) and two (Figs. 1h–j, 2b,d,e,h, 3a,c,d,j, 4f,h,i,l,k, 5c,g and 6g and Extended Data Figs. 1b,d,m, 2h, 3e, 5a,f, 7a,g, 8g and 10d) independent experiments.

### Reporting summary

Further information on research design is available in the Nature Portfolio Reporting Summary linked to this article.

## Data availability

Materials and additional details will be provided on request, under material transfer agreements with MSKCC. Bulk RNA-seq raw data have been deposited at the GEO (GSE234638), and analysed RDS files and the original datasets are available at *Zenodo* (https://doi.org/10.5281/zenodo.10121153). Source data are provided with this paper.

## Code availability

All codes are available at Zenodo (https://doi.org/10.5281/zenodo.10121153).

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

**Acknowledgements** We thank S. Nagata and M. Ménager for the gift of reagents; A. Rudensky, N. Cox, R. Vicario and A. Gitlin for reading the manuscript; S. Grassmann for the gift of reagents and suggestions; S. Frakogianni and S. Ma for help with bioinformatic analysis; and the members of the Geissmann laboratory for support and suggestions. We acknowledge the use of the Integrated Genomics Operation Core, funded by the NCI Cancer Center Support Grant (CCSG, P30 CA08748), Cycle for Survival, and the Marie-Josée and Henry R. Kravis Center for Molecular Oncology. This work was supported by NIH/NCI P30CA008748 to the MSKCC; NIH/NIAID 1R01AI130345, NIH/NHLBI R01HL138090 to F.G.; NCI RO1-CA234139 to R.B.; the Ludwig transatlantic network of excellence to C.K.G. and F.G.; the Ludwig institute for Cancer research basic immunology grant to F.G. and J.M.; Cycle for Survival grants to F.G.; the Alan and Sandra Gerry Metastasis and Tumour Ecosystems Center fellowship to P.-L.L.; CRI Irvington postdoctoral fellowships to Z.D.; and the Ludwig Center at MSKCC basic and translational immunology postdoctoral fellowship to Z.D. and H.Y.

**Author contributions** Z.D. and F.G. designed the study, analysed data, prepared figures and wrote the manuscript. Z.D., P.-L.L., T.L. and H.Y. performed experiments, analysed data and prepared figures. L.L., B.B. and O.E. performed and supervised bioinformatic analyses and prepared figures. Z.C.S. and C.K.G. performed deep learning analysis of macrophages enhancers. Y.Z., C.A.I.-D., J.M., R.B. and J.S. provided reagents and expertise and participated in data analysis and interpretation. All of the authors participated in writing the final manuscript.

**Competing interests** The authors declare no competing interests.

**Additional information**
**Correspondence and requests for materials** should be addressed to Frederic Geissmann.

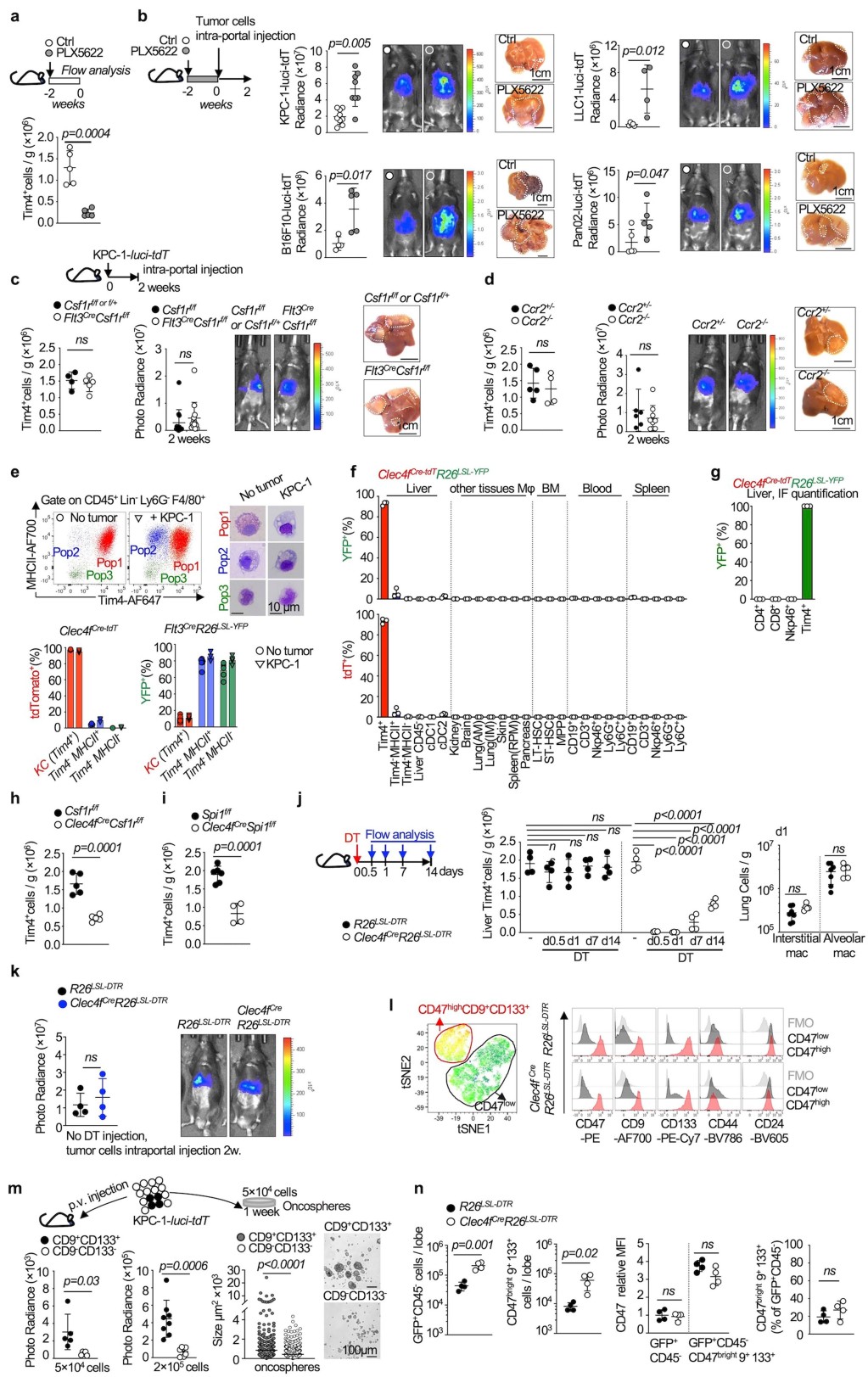

**Extended Data Fig. 1** | See next page for caption.

**Extended Data Fig. 1 | Related to Fig. 1. Targeting of Kupffer cells in tumour models. a-** Kupffer cell number (Tim4$^+$ F4/80$^+$, see methods) analysed by flow cytometry in 6 to 8 weeks-old *C57BL/6J* mice treated with the Csf1r small molecule antagonist PLX5622 food, or control food for 2 weeks, n = 5 mice per group. **b-** Photoradiance and histology analysis of liver tumour burden in 6–8 weeks-old C57bl/6j mice (n = 7) pretreated with PLX5622 Csf1r small molecule antagonist food or control food for 2 weeks, followed by intra-portal injection of 1 × 10$^6$ cells from the, KPC-1-*luciferase* (n = 9 and 8 for control and PLX respectively), LLC1-*luciferase* (n = 5 and 5) or Pan02-*luciferase* (n = 5 and 4) tumour cell lines or 3 × 10$^5$ cells from the B16F10-*luciferase* line (n = 5 and 5), Results are obtained from 2-3 independent experiments per cell lines. **c,d-** Flow cytometry analysis of Tim4$^+$ KCs numbers, photoradiance analysis of tumour burden, and representative liver micrographs from 6-8 weeks old *Flt3$^{Cre}$Csf1r$^{f/f}$* mice (n = 4, 13), *Csf1r$^{f/f}$* littermates (n = 5, 13), or *CCR2$^{-/-}$* mice (n = 5, 6) and *Ccr2$^{+/-}$* littermates (n = 4, 9). two weeks after intra-portal injection of 1 × 10$^6$ KPC-1-*luciferase* cells. Results are obtained from 3 independent experiments. **e-** flow cytometry and cytospin giemsa stain analysis of Kupffer cells (pop1: F4/80$^+$Tim4$^+$), and other myeloid cells (pop2: F4/80$^+$Tim4$^-$MHCII$^+$ and pop3: F4/80$^+$Tim4$^-$MHCII$^-$) from the liver of *C57BL/6J* mice 2 weeks after intra-portal injection of 1 × 10$^6$ KPC-1-luci-tdT cells. The bar plot represents the % of cells from each population that are labelled by tdT in the liver of *Clec4f$^{Cre-tdT}$* mice (n = 3/group) and by YFP in the liver of *Flt3$^{Cre}$R26$^{LSL-YFP}$* mice (no tumour n = 6, KPC-1 n = 4) 2 weeks after intra-portal injection of 1 × 10$^6$ KPC-1 cells or in the absence of tumour injection. **f-** Genetic labelling efficiency in 8 weeks old *Clec4f$^{Cre-tdT}$R26$^{LSL-YFP}$* mice, %YFP$^+$ and % tdT$^+$ cells are measured by flow cytometry in liver myeloid cells: Tim4$^+$ KCs, F4/80$^+$Tim4$^-$MHCII$^+$, F4/80$^+$Tim4$^-$MHCII$^-$, cDC1, cDC2), liver CD45$^-$ cells, tissue macrophages: kidney macrophages, brain macrophages, lung alveolar macrophages (AM), lung interstitial macrophages (IM), skin macrophages, splenic red pulp macrophages (RPM), bone marrow long term HSCs (LT-HSC), short term HSCs (ST-HSC), multipotent progenitor MPP, blood and spleen CD19$^+$ B cells, Ly6G$^+$ granulocytes, Ly6C$^+$ monocytes, CD3$^+$ T cells (see methods). n = 3 mice per group. **g-** Percentage of YFP$^+$ cells among CD4$^+$ T cells, CD8$^+$ T cells, Nkp46$^+$ NK cells, Tim4$^+$ KCs on liver cryosection from 8 weeks old Clec4f$^{Cre-tdT}$R26$^{LSL-YFP}$ mice. n = 3 mice per group. **h,i-** Flow cytometry analysis of Tim4$^+$ KCs numbers in 6-8 weeks-old *Clec4f$^{Cre}$Csf1r$^{f/f}$* mice (n = 5) or *Csf1r$^{f/f}$* littermates (n = 5) (**h**), *Clec4f$^{Cre}$Spi1$^{f/f}$* mice(n = 4) or *Spi1$^{f/f}$* littermates(n = 6) (**i**). **j-** Six to 8 weeks-old *Clec4f$^{Cre}$R26$^{LSL-DTR}$* mice and *R26$^{LSL-DTR}$* mice are injected with DT, liver Tim4$^+$KCs numbers (n = 4/group), lung interstitial macrophages and alveolar macrophages numbers (n = 7,5 respectively) are quantified by flow cytometry at indicated time point after DT injection. **k-** Photoradiance analysis of tumour burden in 6-8 weeks old *Clec4f$^{Cre}$R26$^{LSL-DTR}$* mice and *R26$^{LSL-DTR}$* littermates 2 weeks after intra-portal injection of 1 × 10$^6$ KPC1-*luciferase* cells, n = 4 mice per group. **l-** Representative tSNE and histogram analysis of expression of the markers CD47$^{bright}$, CD9, and CD133, among GFP$^+$ CD45$^-$ tumour cells in *Clec4f$^{Cre-tdT}$R26$^{LSL-DTR}$* (n = 4) and *Clec4f$^{Cre-tdT}$* littermates (n = 4), treated with DT and that received intra-portal injection of 1 × 10$^6$ KPC-1 cells 2 weeks before analysis (see methods). Results from 2 independent experiments. **m-** analysis of metastatic potential of CD47$^{bright}$ CD9$^+$CD133$^+$ tumour cells and CD47$^{low}$ CD9$^{low}$ CD133$^{low}$ tumour cells in vivo by bioluminescent analysis, two weeks after intra-portal injection of 5 × 10$^4$ cells (n = 5/group) or 2 × 10$^5$ KPC-1-luci-tdT cells (n = 8,7 respectively) (Left, circles represent individual mice), and in vitro clonogenic potential in oncosphere culture (n = 561,563 respectively) (right, circles represent individual oncospheres, see Methods). **n-** Plots indicate the number of GFP$^+$CD45$^-$ cells and of GFP$^+$CD45$^-$CD47$^{bright}$CD9$^+$CD133$^+$ cells per liver lobe, and the MFI of CD47 in GFP$^+$CD45$^-$ cells and GFP$^+$CD45$^-$CD47$^{bright}$CD9$^+$CD133$^+$ cells, in mice from (l). Statistics: One-way ANOVA (**i**). unpaired two-tailed t test (**a,b,c,d,g,h,i,j,k,m,n**). Mann-Whitney test(two-tailed) (**m**). Dots represent individual mice (**a,b,c,d,e,f,h,i,j,k,m,n**). mean ± sd. ns, not significant.

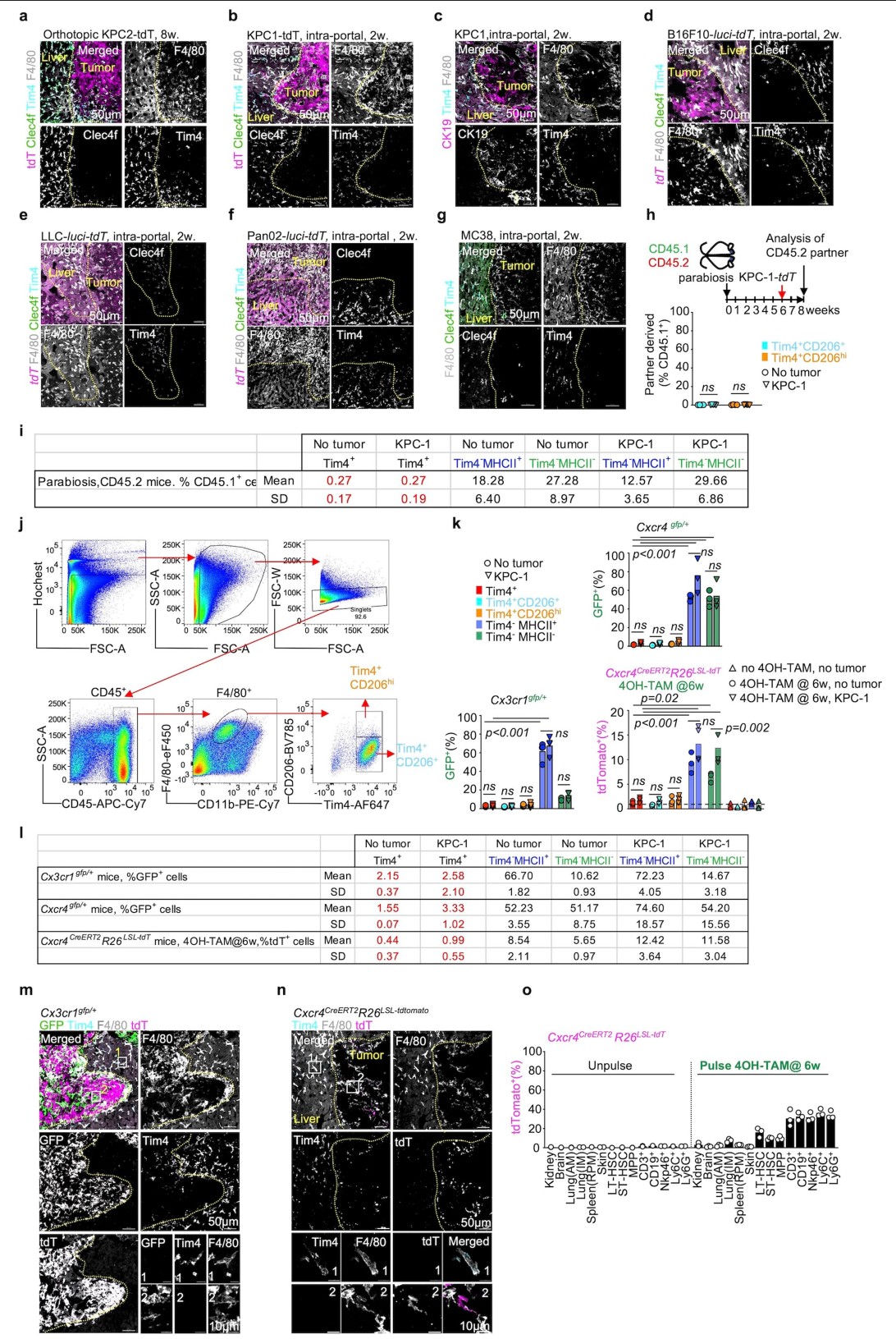

**Extended Data Fig. 2** | See next page for caption.

**Extended Data Fig. 2 | Related to Fig. 2. Anatomical location and turnover of liver macrophages in metastatic liver. a-g-** Representative immunofluorescence staining for Clec4f, F4/80, Tim4, tdT, CK19 on frozen liver sections from wt C57BL/6j mice 8 weeks after pancreatic orthotopic injection of $2 \times 10^5$ KPC-2-*tdT* cells. **b-g-** idem, from wt C57BL/6j mice 2 weeks after intra-portal injection of $1 \times 10^6$ KPC-1-*tdT* cells, $1 \times 10^6$ KPC-1 cells, $5 \times 10^5$ B16F10-*luci-tdT* cells, $1 \times 10^6$ LLC1-*luci-tdT* cells, $1 \times 10^6$ Pan02-*luci-tdT* cells, or $1 \times 10^6$ MC38 cells. n = 3 mice per group. **h-** Flow cytometry analysis of the % of CD45.1$^+$ cells among Tim4$^+$CD206$^+$ KCs and Tim4$^+$CD206$^{hi}$ KCs in the liver of CD45.1/CD45.2 parabiotic pairs 2 weeks after intra-portal injection of $10^6$ KPC-1-tdT in the CD45.2 partner (n = 5), or not injected as control (n = 4). **i-** table represents the percentage (mean and sd) of CD45.1$^+$ partner-derived cells among Tim4$^+$, Tim4$^+$CD206$^+$, and Tim4$^+$CD206$^{hi}$ KC, and TIM4$^-$ TAMs in tumour free and tumour bearing livers from CD45.2 parabionts in (h). **j-** Gating strategy for separation of Tim4$^+$CD206$^+$ and Tim4$^+$CD206$^{hi}$ liver KC by flow cytometry. **k-** Bar-plots show the percentage of Tim4$^+$, Tim4$^+$CD206$^+$, Tim4$^+$CD206$^{hi}$ KC, and TIM4$^-$ TAMs labelled with GFP in *Cxcr4$^{gfp/+}$* mice (n = 3/group) and in Cx*3cr1$^{gfp/+}$* mice (n = 3/group), and labelled with tdT in *Cxcr4$^{CreERT2}$;R26$^{LSL-tdTomato}$* mice pulsed with 4OH-TAM at 6 week-old, that have received intra-portal injection of $1 \times 10^6$ KPC-1-*luci-tdT* cells 2 weeks before analysis (n = 4), or not injected with tumour cells (n = 3/group), not injected with 4OH-TAM (n = 3/group) as control. **l-** Table represents the percentage (mean and sd) of Tim4$^+$ KC, Tim4$^+$ CD206$^+$ KC, Tim4$^+$CD206$^{hi}$ KC, and TIM4$^-$ TAMs in the liver of tumour free and tumour bearing *Cxcr4$^{gfp/+}$* mice, *Cx3cr1$^{gfp/+}$* mice, and *Cxcr4$^{CreERT2}$;R26$^{LSL-tdTomato}$* mice pulsed at 6 weeks with 4OH-TAM. **m-** Immunofluorescence staining for F4/80, Tim4, GFP and tdTomato on frozen liver section from *Cx3cr1$^{gfp/+}$ tumour* bearing mice in (**k**). **n-** immunofluorescence staining for F4/80, Tim4 and tdTomato on frozen liver section from *Cxcr4$^{CreERT2}$ R26$^{LSL-tdTomato}$* tumour bearing mice in (**k**). **o-** *Cxcr4$^{CreERT2}$R26$^{LSL-tdTomato}$* mice are pulsed with 4OH-TAM (n = 3) or PBS (n = 2) at 6 weeks and analysed 2 weeks later. Bar graphs represent the % of tdT$^+$ cells determined by flow cytometry among the indicated cell types. Statistics: One-way ANOVA (**h,k**). Dots represent individual mice. mean ± sd. ns, not significant.

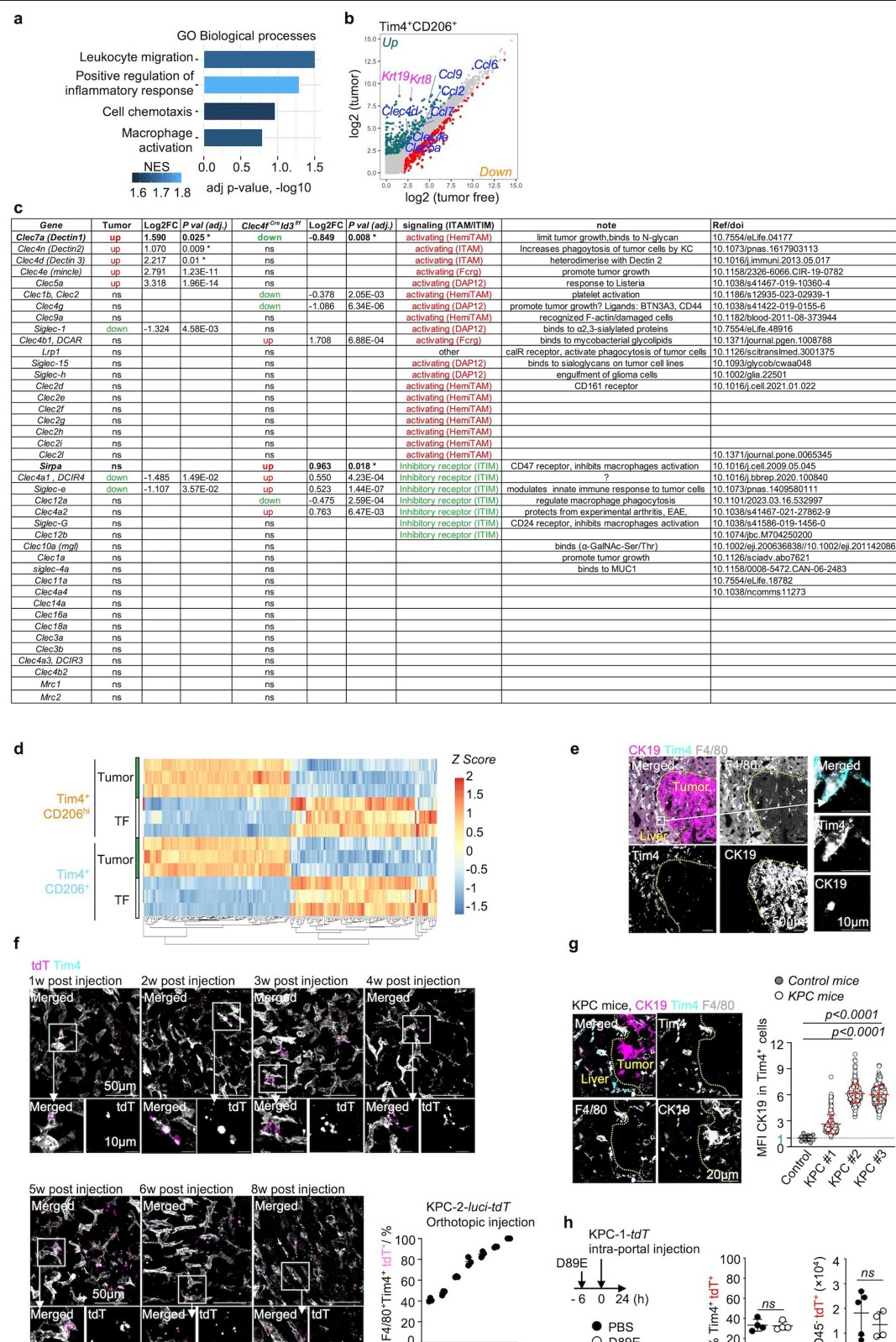

**Extended Data Fig. 3** | See next page for caption.

**Extended Data Fig. 3 | Related to Fig. 2. Kupffer cells engulf tumour cells.**
**a-** Selected pathways (see Methods) upregulated in RNA-seq analysis of KCs from C57BL/6j mice 2 weeks after intra-portal injection of $1 \times 10^6$ KPC-1 cells, in comparison to no injection, n = 3 mice per group, (Supplementary data T 2).
**b-** Scatterplot of differentially expressed genes (adj. p-values are obtained using Benjamini and Hochberg method for multiple testing and considered significant when adj. p < 0.05) in KCs from tumour-bearing mice, from the RNA-seq analysis in (a). Selected genes are indicated in blue (lectins and chemokines) and in pink (cytokeratins, from engulfed tumour cells). **c-** Expression of activating and inhibitory receptors by KCs from RNAseq data (Supplementary data 2 and 3) and RTqPCR data (*). Second to 4th column indicates if gene expression is up or down in metastatic liver, fold changes, and adj. p value. Fifth to $7^{th}$ column '$Clec4f^{Cre}Id3^{f/f}$' indicates gene which expression is up or down regulated in $Clec4f^{Cre}Id3^{f/f}$ mice, fold changes, and adj. p value. Column 8 indicates the presence of ITIM or ITAM motifs. **d-** Heatmap represents the top 100 up and down DEG genes in Tim4$^+$CD206$^+$ KC cells and Tim4$^+$CD206$^{hi}$ KC respectively, in RNA-seq analysis from (a). **e-** Representative immunofluorescence staining for CK19, F4/80, Tim4 on frozen liver section from *C57BL/6J* mice received $1 \times 10^6$ KPC-1 cells through intra-portal injection for 2 weeks. n = 3 mice per group. **f-** Representative whole mount immunofluorescence imaging and quantification of tdT$^+$% cells in Tim4$^+$ KCs from metastatic liver, from *C57BL/6J* mice which received $2 \times 10^5$ KPC-2-*luci-tdT* cells through pancreatic orthotopic injection and are analysed weekly for 2 months. n = 3 mice per group.
**g-** Representative immunofluorescence staining for CK19, Tim4, F4/80, quantification of CK19 relative MFI in Tim4$^+$ KCs on liver samples from KPC mice ($p48^{Cre} Kras^{LSL-G12D}p53^{LSL-R172H}$) and control mice ($Kras^{LSL-G12D}p53^{LSL-R172H}$). Dots represent individual KCs from 3 mice per group. **h-** Analysis of livers from mice receiving phosphatidylserine blockade D89E or PBS 6 h before intra-portal injection of $1 \times 10^6$ KPC-1-*tdT* cells and analysed after 24 h. *Left:* Quantification by Immunofluorescence of Tim4$^+$ KCs stained with tdT, n = 4/group; *right:* quantification of CD45$^-$ GFP$^+$ tumour cells by flow cytometry (right). PBS n = 4, D89E n = 4. Statistics: Kruskal-Wallis test (**g**), unpaired two-tailed t test (**h**), Dots represent individual mice (**f,h**). mean ± sd. ns, not significant.

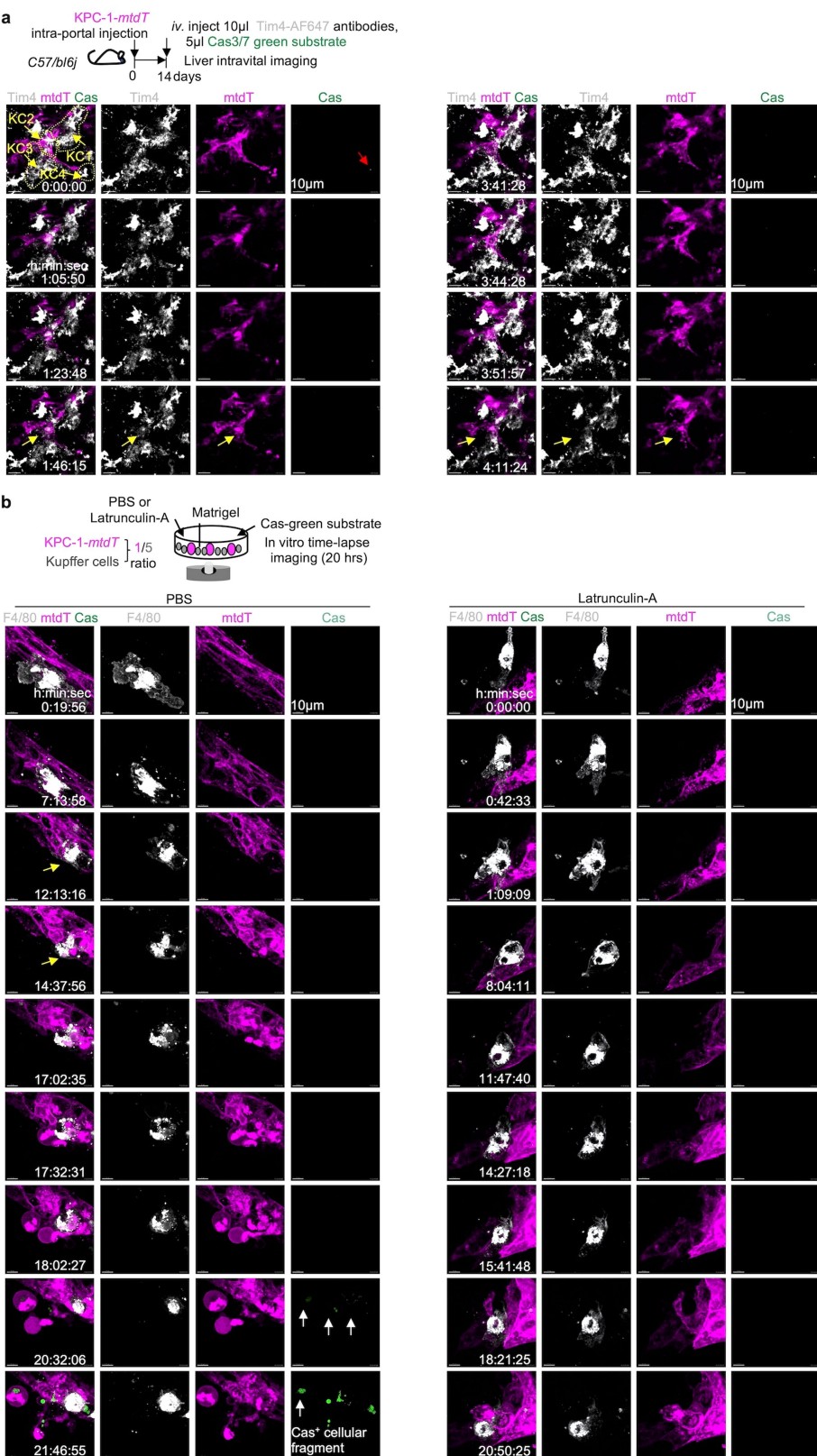

**Extended Data Fig. 4 | Related to Fig. 2. Kupffer cells engulf live tumour cells in vivo and in vitro. a**- Schematic of time lapse intravital imaging of in vivo uptake of KPC-1-*memtdT* by KCs in liver from *C57BL/6j* mice 2 weeks after intra-portal injection of $1 \times 10^6$ KPC-1-*memtdT* cells in the presence of *iv.* injection of Tim4-AF647 antibodies and CellEvent Caspase-3/7 green reagent. Representative whole mount immunofluorescence imaging and time-lapse images are shown. Sequential images show KC1, KC2, KC3, KC4 engulfing tumour cells (yellow arrows), red arrow shows Cas3/7 green signalling.

See Movie 1. **b**-Representative time-lapse images of in vitro uptake of KPC-1-*memtdT* by wide type KCs in Matrigel in the presence of F4/80-AF647 antibodies, and CellEvent Caspase-3/7 green reagent, and in the presence of Latrunculin-A or PBS control. In the PBS group, sequential images show a KC engulfing Cas3/7 green⁻ tumour cells after 5 hrs of contact, between 12 and 14 hrs of culture (yellow arrows), and Cas3/7 green⁺ activation in cellular fragments ~6 hrs later (white arrows).

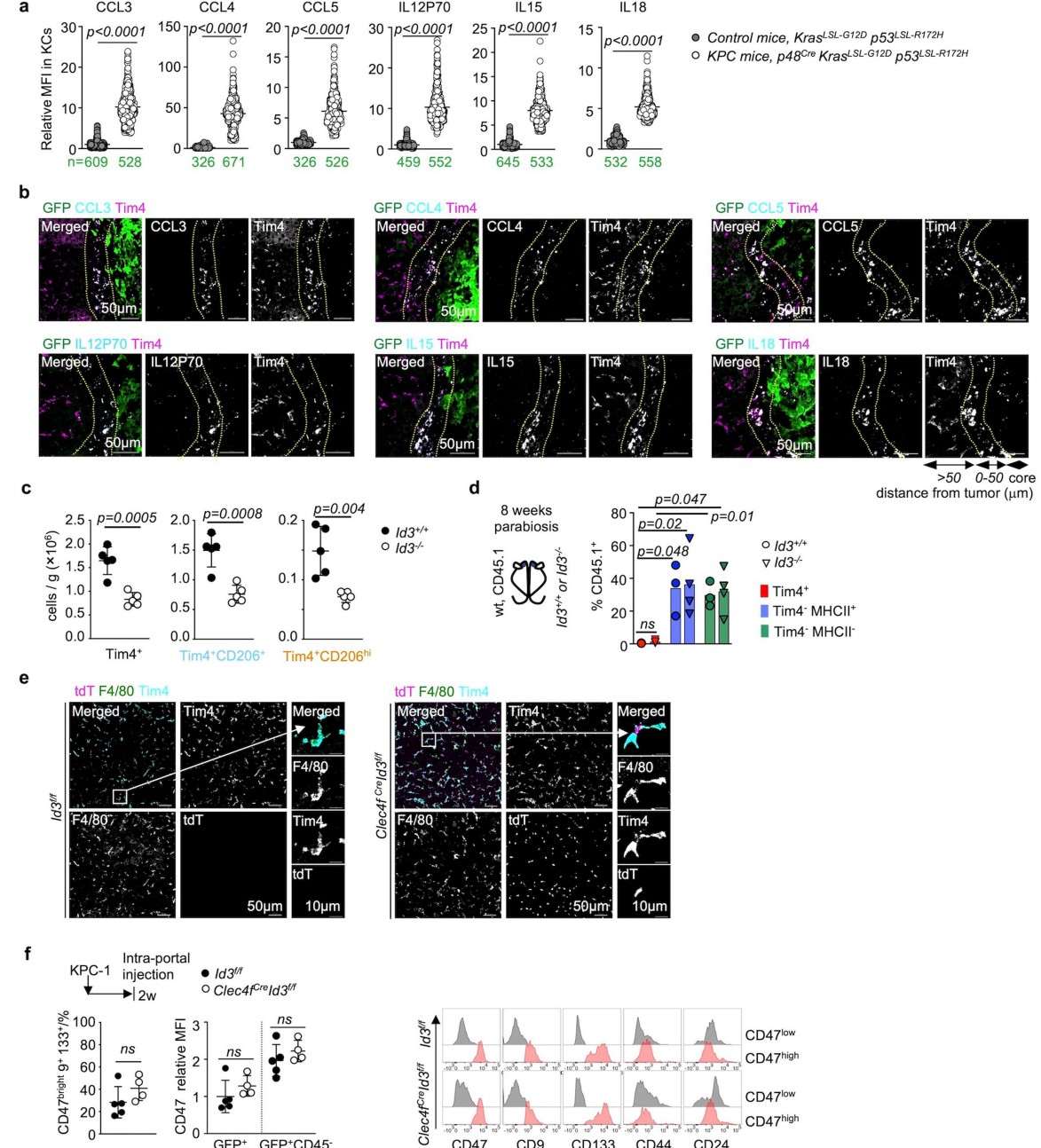

**Extended Data Fig. 5 | Related to Figs. 2 & 3. Analysis of *Id3*-deficient mouse models. a-** Quantification of cytokines/chemokines expression by Tim4⁺ KCs by immunofluorescence on liver tissue sections from KPC (*p48^Cre Kras^LSL-G12D p53^LSL-R172H*) mice and control (*Kras^LSL-G12D p53^LSL-R172H*) mice. n numbers are indicated in green. Dots represent MFI of individual KCs. **b-** Representative micrographs of immunofluorescence staining for cytokines/chemokines on frozen liver sections from wt mice 2 weeks after intra-portal injection of 1 × 10⁶ KPC-1-*gfp* cells. n = 3 mice per group. **c-** Number of Tim4⁺ cells, Tim4⁺ CD206⁺ cells and Tim4⁺CD206^hi cells per gram of liver tissue by flow cytometry in *Id3^−/−* mice, and *Id3^+/+* littermates, n = 5 mice per group. **d-** Flow cytometry quantification Percentage of partner-derived (CD45.1⁺) wt Tim4⁺ KCs and Tim4⁻ myeloid cells in the liver of CD45.2 *Id3^−/−* mice (n = 4), or CD45.2 *Id3^+/+* mice (n = 3) parabiosed

with wt CD45.1 mice for 8 weeks. **e-** Representative micrograph of Tim4, F4/80, and tdTomato expression by immunofluorescence in liver frozen section from 8 weeks old *Id3^f/f* mice and *Clec4f^Cre Id3^f/f* mice. n = 3 mice per group. **f-** (left) Percentage of CD47^brightCD9⁺CD133⁺ cells among GFP⁺CD45⁻ tumour cells, and relative expression of CD47 among total GFP⁺CD45⁻ tumour cells and GFP⁺CD45⁻ CD47^brightCD9⁺CD133⁺ tumour cells in the liver of *Id3^f/f* mice(n = 5) and *Clec4f^Cre Id3^f/f* mice(n = 4) 2 weeks after intra-portal injection of 1 × 10⁶ KPC-1-*luci-gfp* cells. (right) representative histograms of CD47, CD9, CD133, CD44, and CD24 expression tumour cells from (left). Statistics: unpaired two-tailed t test (**c,f**). Mann-Whitney test(two-tailed) (**a**), One-way ANOVA (**d**). Dots represent individual mice (**c,d,f**), mean ± sd. ns, not significant.

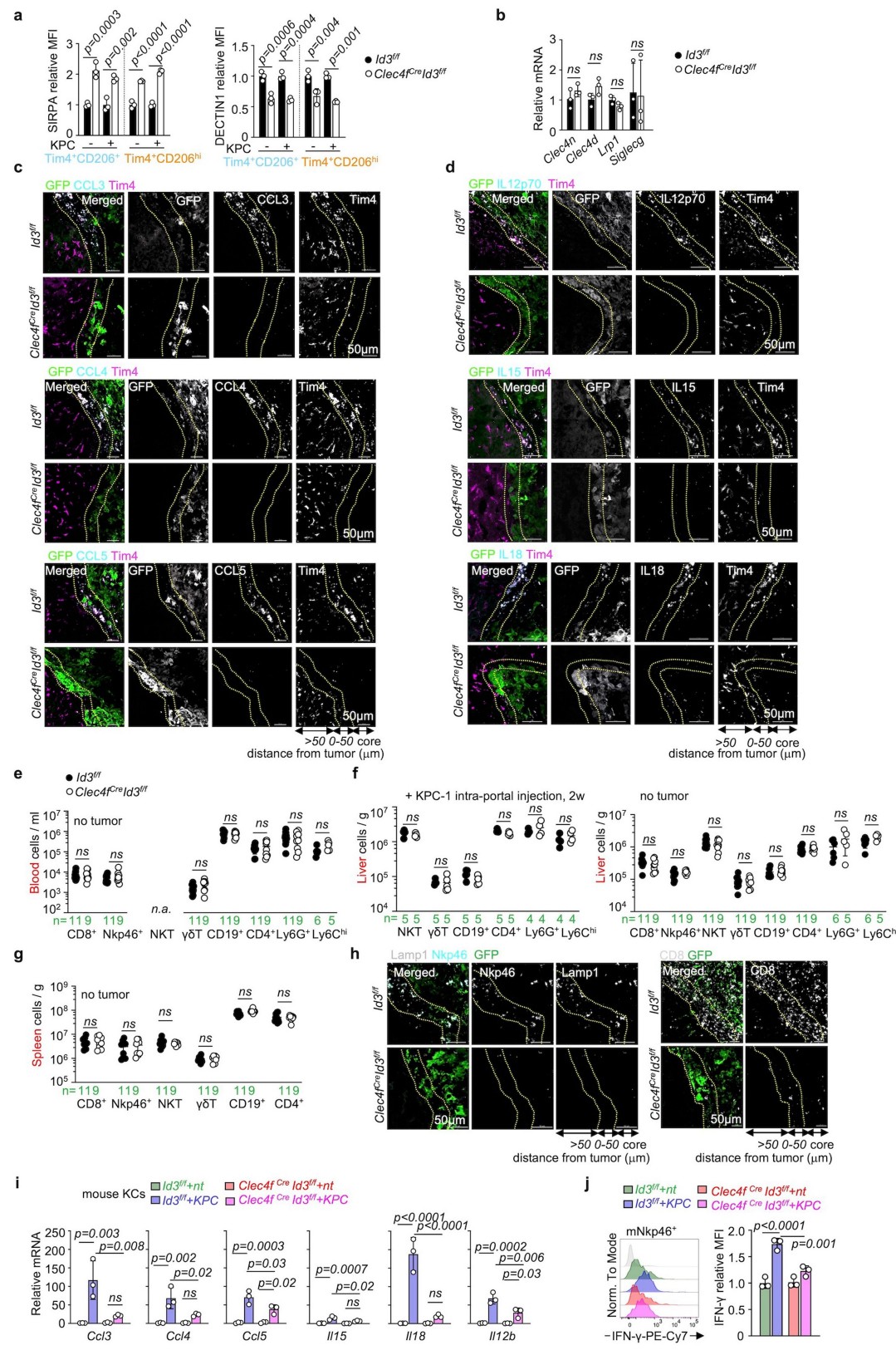

**Extended Data Fig. 6** | See next page for caption.

**Extended Data Fig. 6 | Related to Fig. 4. Id3-dependent cytokines and chemokines expression and lymphoid cells recruitment and activation in response to tumour. a-** SIRPA and Dectin-1 expression by flow cytometry of the main (Tim4⁺CD206⁺) and minor (Tim4⁺CD206^hi) KCs subsets from the liver of *Clec4f^Cre Id3^f/f* mice and *Id3^f/f* littermates, 2 weeks after intra-portal injection of $1 \times 10^6$ KPC-1-*luci-gfp* cells (KPC+), or from control mice without tumour (KPC−). n = 3 mice per group. **b-** Expression of Clec4n, Clec4d, Lrp1, and Siglec-G by RT-qPCR by KCs from the liver of *Clec4f^Cre Id3^f/f* mice and *Id3^f/f* mice, n = 3 mice per group. **c,d-** Representative immunofluorescence micrograph of chemokines (**c**) and cytokines (**d**) expression by KCs of in the liver of *Clec4f^Cre Id3^f/f* mice and *Id3^f/f* littermates 2 weeks after intra-portal injection of $1 \times 10^6$ KPC-1-*gfp* cells. n = 3 mice per group. **e-g-** Number of CD8⁺ T cells, Nkp46⁺ NK cells, $\gamma\delta$ T cells, CD19⁺ B cells, CD4⁺ T cells, Ly6G⁺ granulocytes, Ly6C^hi monocytes determined by flow cytometry in blood (**e**), and liver (**f**) and spleen (**g**) from 6-8 weeks old *Id3^f/f* mice and *Clec4f^Cre Id3^f/f* mice (f,g), or from mice 2 weeks after intra-portal injection of $1 \times 10^6$ KPC-1-*gfp* cells (f). n numbers are indicated in green. **h-** Representative immunofluorescence micrograph for Lamp1, Nkp46, CD8, and GFP expression in liver sections from mice in (**c**). **i-** Expression of chemokines and cytokines by RT-qPCR by KCs from *Clec4f^Cre Id3^f/f* mice and *Id3^f/f* littermates, cocultured with or without KPC cells for 48 h. n = 3 per group. **j-** IFN-$\gamma$ production by flow cytometry by mouse splenic NK cells cultured for 3 days with supernatant from coculture in (h), n = 3 per group. Statistics: unpaired two-tailed t test (**b,e,f,g**), One-way ANOVA (**a,i,j**). Dots represent individual mice (**a,b,e,f,g**). mean ± sd. ns, not significant.

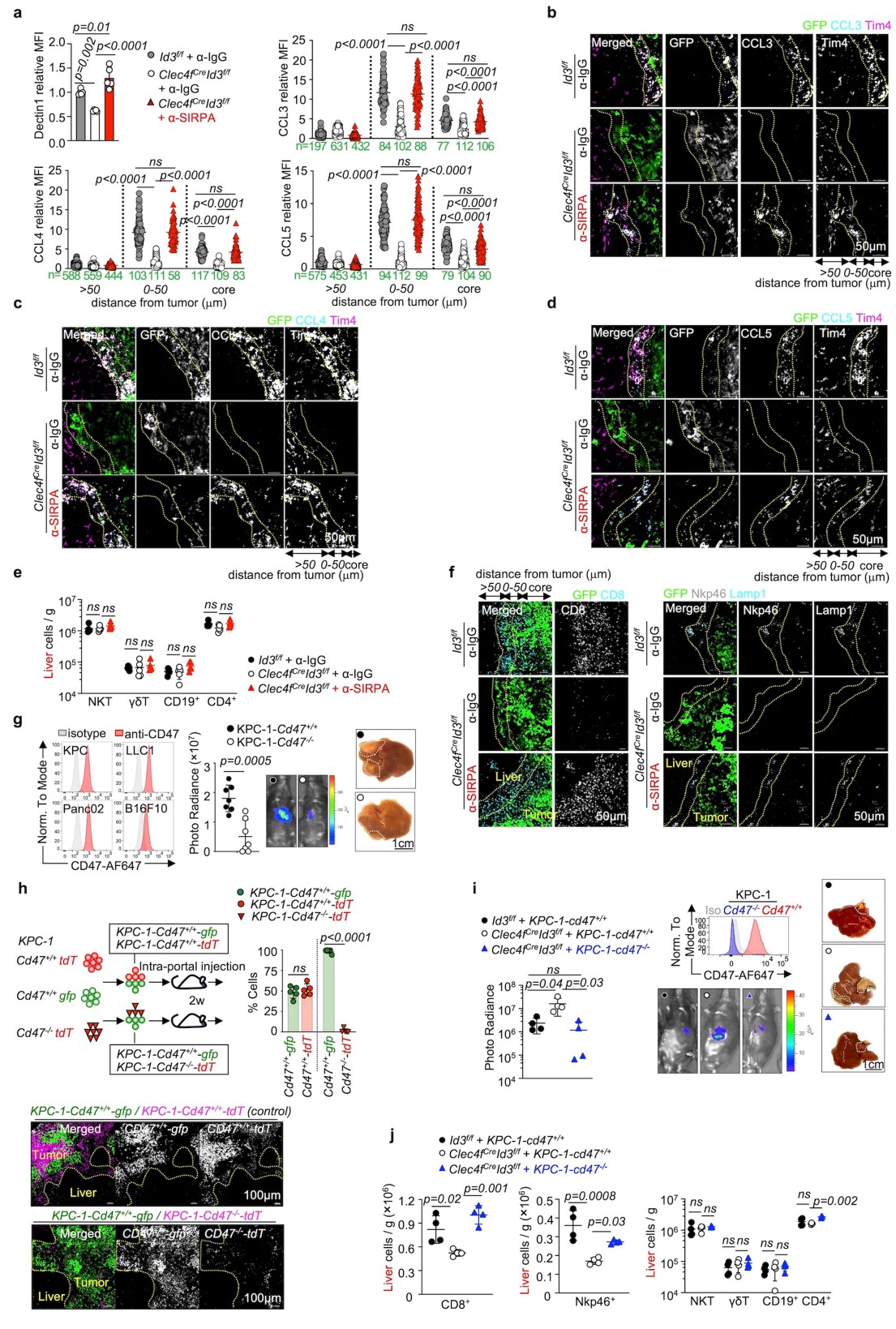

**Extended Data Fig. 7** | See next page for caption.

**Extended Data Fig. 7 | Related to Fig. 5. SIRPA blockade and CD47 deficiency rescue *Id3* deficient mice. a-** Dectin1 expression by flow cytometry (dots represent individual mice, n = 5/group), and CCL3, CCL4, and CCL5 expression by immunofluorescence (dots represent individual KCs) by Tim4[+] KCs in liver from 8–12 weeks-old mice *Clec4f^{Cre} Id3^{f/f}* mice and *Id3^{f/f}* littermates, two weeks after intra-portal injection of $1 \times 10^6$ KPC-1-*luciferase* cells, treated with anti-SIRPA antibodies or IgG control antibodies. n numbers are indicated in green. **b-d-** Representative immunofluorescence micrograph of CCL3, CCL4, CCL5, by Tim4[+] KCs at the boundary of GFP[+] tumours in liver sections from mice in (**a**). **e-** Number of NKT cells, $\gamma\delta$ T cells, CD19[+] B cells, CD4[+] T cells per gram of tissue by flow cytometry analysis in mice from (**a**). n = 5 mice per group. **f-** Representative immunofluorescence micrograph of CD8, Lamp1, and Nkp46-expressing cells at the boundary of GFP+ tumours in liver sections from mice in (**a**). **g-** (left) CD47 expression by flow cytometry of tumour cell lines. (right) bioluminescence imaging and micrograph of liver tumour burden in *C57BL/6J* mice 2 weeks after intra-portal injection of $1 \times 10^6$ KPC-1-*Cd47^{+/+}-luci-tdT* cells or $1 \times 10^6$ KPC-1-*Cd47^{-/-}-luci-tdT* cells, n = 7 mice per group from 2 independent experiments. **h-** Competitive in vivo proliferation assay between CD47[−/−] deficient and *Cd47^{+/+}* control KPC1 tumour cells in liver metastasis. $7 \times 10^5$ *KPC-1-Cd47^{+/+}-gfp* tumour cells mixed with $7 \times 10^5$ KPC-1-*Cd47^{+/+}-tdT* tumour cells or $7 \times 10^5$ KPC-1-*Cd47^{-/-}-tdT* tumour cells in a 1:1 ratio are injected into c *C57BL/6J* mice through intra-portal injection. Percentage of fluorescent cells expression in the liver and representative immunofluorescence of GFP and tdT performance after 2 weeks are shown. n = 5 mice per group. **i-** Photoradiance analysis of liver tumour burden in 8-12 weeks-old *Clec4f^{Cre} Id3^{f/f}* mice and *Id3^{f/f} littermates* two weeks after intra-portal injection of $1 \times 10^6$ KPC-1-*luciferase-cd47^{+/+}* cells or KPC-1-*luciferase-cd47^{-/-}* cells, n = 4 mice per group. **j-** Numbers of CD8[+] T cells, Nkp46[+] NK cells, NKT cells, $\gamma\delta$ T cells, CD19[+] B cells, CD4[+] T cells per gram of tissue in the liver from mice in (**i**). n = 4 mice per group. Statistics: One-way ANOVA (**a,e,i,j**), Kruskal-Wallis test (**a**), unpaired two-tailed t-test (**g,h**). Dots represent individual mice (**a,e,g,h,i,j**). mean ± sd. ns, not significant.

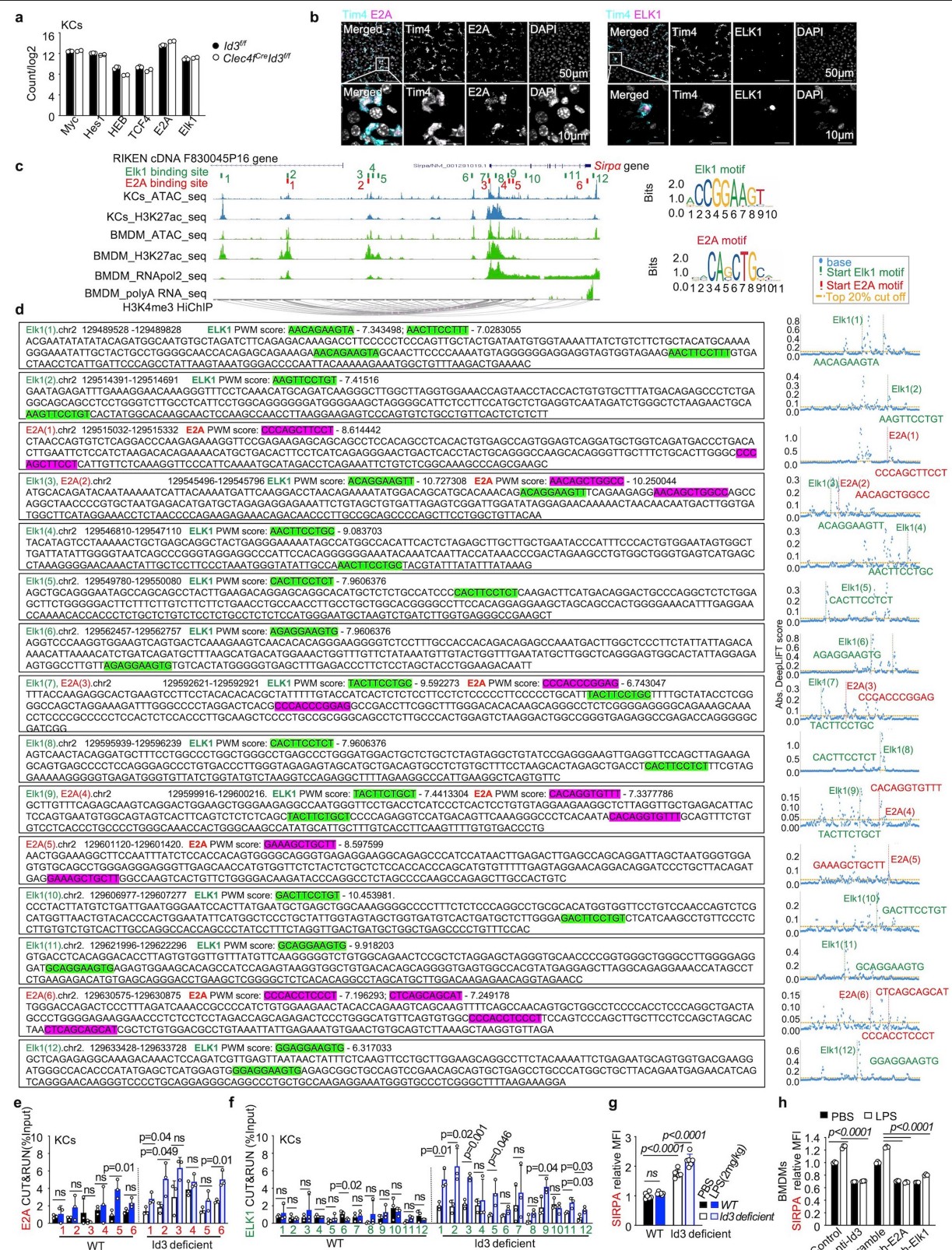

**Extended Data Fig. 8** | See next page for caption.

**Extended Data Fig. 8 | Related to Fig. 6.** *Id3* regulates *Sirpα* gene expression via *Elk1* and *E2A*. **a-** Expression of Myc, Hes1, HEB, TCF4, E2A, and Elk1 by KCs from *Clec4f^Cre^Id3^f/f^* mice (n = 2) and *Id3^f/f^* littermates (n = 3) by RNAseq, circle represent individual mice. **b-** Expression of DAPI and E2A or ELK1 by Tim4+ KC, by immunofluorescence in liver sections from 8 weeks old *C57BL/6 J* mice, n = 3 mice per group. **c-** Diagram represents candidate binding sites for E2A and Elk1 at the *Sirpα* locus (see methods). (**d**) PWM score and DeepLIFT analysis of candidate Elk1 and E2A binding sites at *Sirpα* locus in mouse KCs and BMDM. **e,f-** Binding of E2A and ELK1 to *Sirpα* promoter/enhancer in KCs from *Id3* deficient mice and control littermates 12 h after i.p. injected of 2 mg/kg LPS or PBS by CUT&RUN analysis. n = 3 mice per group. **g-** SIRPA expression by flow cytometry by KCs from *Id3* deficient mice and control littermates 12 h after i.p. injected of 2 mg/kg LPS or PBS. n = 6 mice per group. **h-** SIRPA expression by flow cytometry by BMDM expressing Scramble, or *E2a* sh-RNA, *Elk1* Sh-RNA, lenti-m*Id3* and treated with LPS (50 ng/ml for 6 hrs), or PBS. n = 3 per group. Statistics: One-way ANOVA (**g,h**), unpaired two-tailed t-test (**e,f**). Dots represent individual mice (**a,g**). mean ± sd. ns, not significant.

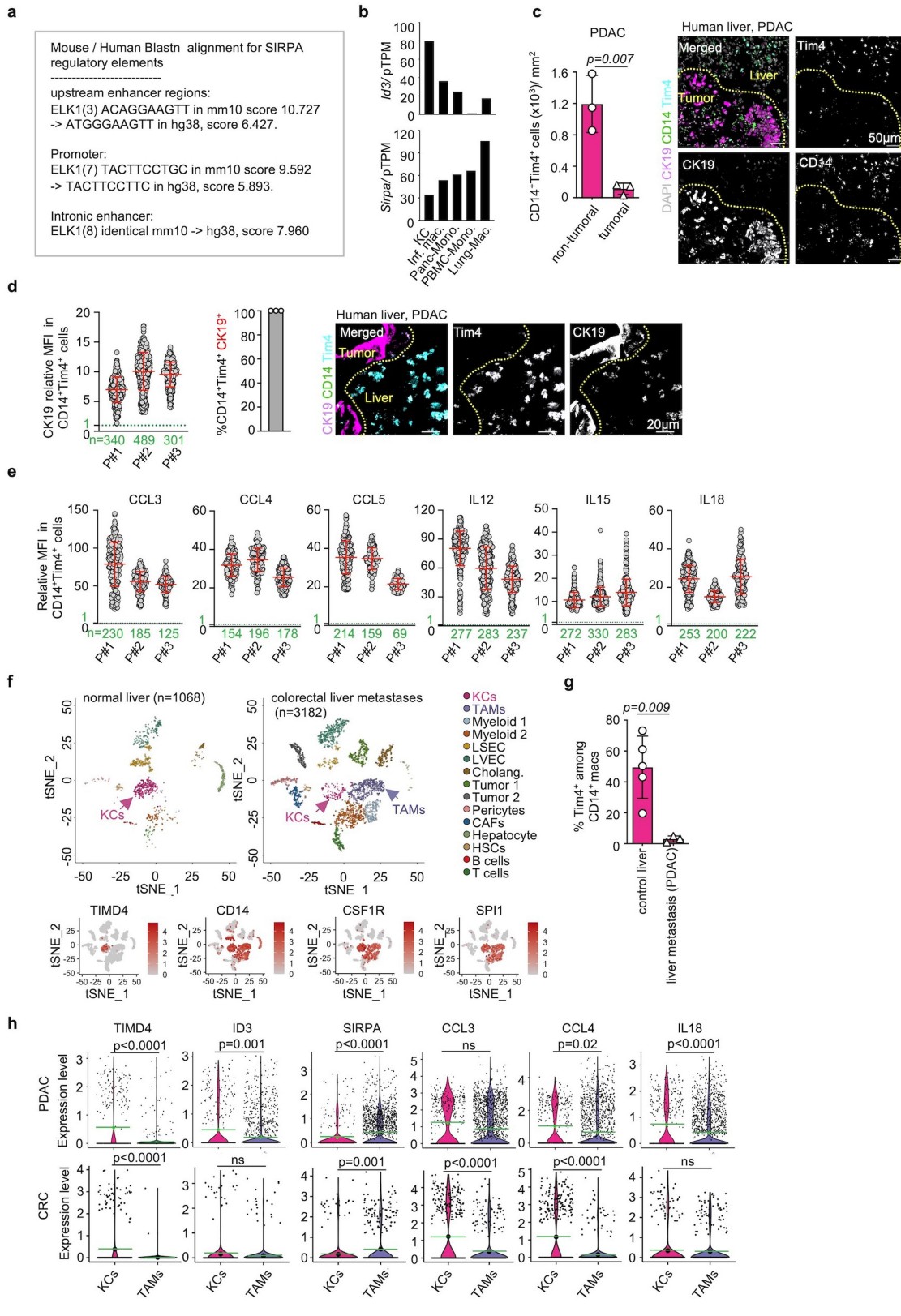

**Extended Data Fig. 9** | See next page for caption.

**Extended Data Fig. 9 | Related to Fig. 6. human KC: anatomical location, engulfment of tumour cells, and gene expression. a-** Mouse / Human Blastn alignment for Sirpa regulatory elements. **b-** Expression of ID3 and SIRPA gene expression in human cells by scRNAseq, from https://www.proteinatlas.org/. Inf. mac: Liver inflammatory macrophages, mono: monocytes. **c-** Numbers of TIM4$^+$ CD14$^+$ KC/mm$^2$ in tumoral and peri-tumoral areas from metastatic liver samples from PDAC patients, n = 3, mean ± sd, unpaired two-tailed t-test. A representative immunofluorescence micrograph of CK19, TIM4, and CD14 expression is also shown (right, bar = 50 μM). **d-** Expression of CK19 by TIM4$^+$ KCs in metastatic liver samples from PDAC patients, each dot represents a KC (left). n numbers are indicated in green. The bar-plot (center, n = 3 patients) indicate the % of KC stained for CK19. A representative high power immunofluorescence micrograph of CK19, TIM4, and CD14 expression is also shown (right, bar = 20 μM). mean ± sd. **e-** Expression of CCL3, CCL4, CCL5, IL12, IL15, IL18 by TIM4$^+$CD14$^+$KCs from patients in (**c,d**). Dots represent relative Mean Fluorescence Intensity, of individual KCs, mean ± sd, normalized to background MFI of 1 (green dotted line). n numbers are indicated in green. **f-** ScRNAseq datasets from control liver (1068 cells from one patient, GSE146409) and metastatic liver from colorectal carcinoma (CRC) patients (3182 cells from 3 patients, GSE146409). tSNE in top panels represent clustering of liver cells by cell type (KCs: Kupffer cells, TAM: tumour associated macrophages, HSCs: hepatic stellate cells, LSEC: liver sinusoidal endothelial cells, Cholang: Cholangiocytes LVEC: liver vascular endothelial cells, CAFs: cancer-associated fibroblasts). tSNE in bottom panel represent *TIMD4, CD14, CSF1R, SPI1* expression in cell clusters. **g-** % of TIM4$^+$ cells among CD14$^+$ macrophage clusters (TIM4$^+$CD14$^+$ KCs and TIM4$^-$ CD14$^+$ TREM2$^+$ tumour associated macrophages), from the datasets above plus PDAC liver metastasis liver samples (19843 cells, from 3 patients, GSE205013, n = 3) and control liver samples (8439 cells, from 5 patients, GSE115469, n = 5). mean ± sd, Statistics: unpaired two-tailed t-test. **h-** Expression of *ID3, TIMD4, SIRPA, CCL3, CCL4, and IL18* by ScRNAseq from the CRC and PDAC liver metastasis liver samples above (PDAC: 19843 cells, from 3 patients, GSE205013, CRC: 3182 cells from 3 patients, GSE146409) by the TIM4$^+$CD14$^+$ (KCs) and TIM4$^-$CD14$^+$TREM2$^+$ (tumour associated macrophages) clusters. Green line indicates the mean, p-values are obtained using Wilcoxon test(two-tailed). ns, not significant.

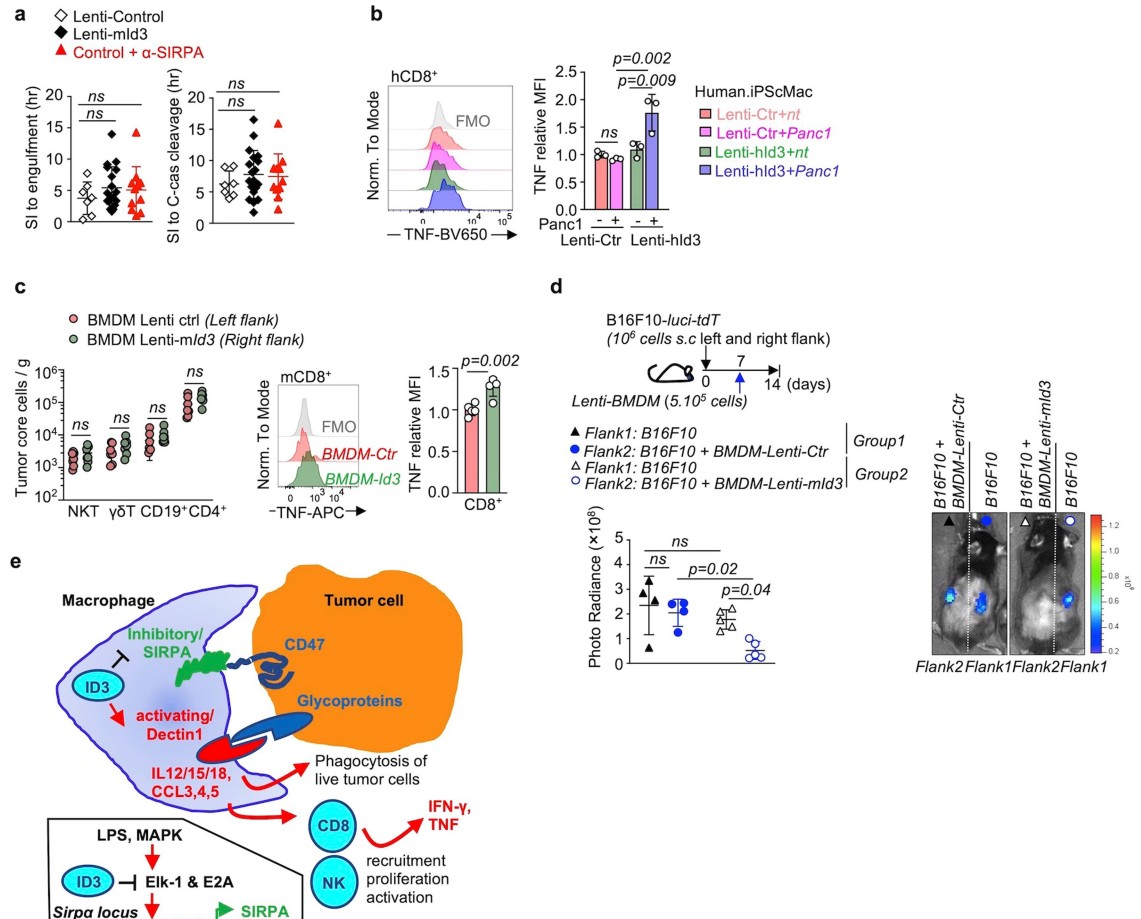

**Extended Data Fig. 10 | Related to Fig. 6. Id3-expressing macrophages control tumour growth. a** - Engulfment of KPC-1-*memtdT* cells (Casp.3/7 not cleaved) by BMDM expressing a control lentivirus (n = 7) in the presence or absence of α-SIRPA blocking antibody (n = 11), or an ID3 lentivirus (n = 18). Left: time from stable interaction between macrophages and tumour cells to tumour cells engulfment. Right: time from stable interaction between macrophages and tumour cells and the detection of Casp.3/7 cleavage, dots represent individual macrophages. **b** - Production of TNF by Human CD8⁺ T cells stimulated with anti CD3/CD28 activation beads, with supernatant from hiPSC-mac expressing lenti-control or lenti-h*Id3*, and cocultured with Panc1 cells for 48 h or not for 3 days. Human CD8⁺ T cells are treated with cocktail of PMA, ionomycin, brefeldin A and monensin for 6 h, and TNF production measured by flow cytometry. n = 3 per group. **c** - (left) Numbers of NKT cells, γδT cells, CD19⁺ B cells, and CD4⁺ T cells in the tumours from 8–12 weeks-old

*C57BL/6 J* mice two weeks after subcutaneous injection of 1 × 10⁶ B16F10-*luci-tdT* cells into left and right flank, followed by intra-tumour injection of 5 × 10⁵ BMDM expressing lenti-control (left flank) or lenti-m*Id3* (right flank) at day 7 post tumour injection. n = 20 mice per group from 3 experiments. (right) production of TNF by CD8⁺ T cells determined by flow cytometry. n = 5 mice/ group. **d** - Photoradiance analysis of liver tumour burden in 8–12 weeks-old C57BL/6j mice two weeks after subcutaneous injection of 1 × 10⁶ B16F10-*luci-tdT* cells into left and right flank, followed by intra-tumour injection of 5 × 10⁵ BMDM expressing lenti-control (n = 4), lenti-m*Id3* cells (n = 5) or not at day7 post tumour injection. Statistics: One-way ANOVA (**a,b,d**). unpaired two-tailed t-test (**c**). Dots represent individual mice (**c, d**). mean ± sd. ns, not significant. **e** - Schematic shows a hypothesis for the mechanisms that underly Id3-dependent anti-tumour activity of macrophages.

# Reporting Summary

## Statistics

For all statistical analyses, confirm that the following items are present in the figure legend, table legend, main text, or Methods section.

| n/a | Confirmed | |
|---|---|---|
| ☐ | ☒ | The exact sample size (*n*) for each experimental group/condition, given as a discrete number and unit of measurement |
| ☐ | ☒ | A statement on whether measurements were taken from distinct samples or whether the same sample was measured repeatedly |
| ☐ | ☒ | The statistical test(s) used AND whether they are one- or two-sided *Only common tests should be described solely by name; describe more complex techniques in the Methods section.* |
| ☒ | ☐ | A description of all covariates tested |
| ☐ | ☒ | A description of any assumptions or corrections, such as tests of normality and adjustment for multiple comparisons |
| ☐ | ☒ | A full description of the statistical parameters including central tendency (e.g. means) or other basic estimates (e.g. regression coefficient) AND variation (e.g. standard deviation) or associated estimates of uncertainty (e.g. confidence intervals) |
| ☐ | ☒ | For null hypothesis testing, the test statistic (e.g. *F*, *t*, *r*) with confidence intervals, effect sizes, degrees of freedom and *P* value noted *Give P values as exact values whenever suitable.* |
| ☒ | ☐ | For Bayesian analysis, information on the choice of priors and Markov chain Monte Carlo settings |
| ☒ | ☐ | For hierarchical and complex designs, identification of the appropriate level for tests and full reporting of outcomes |
| ☒ | ☐ | Estimates of effect sizes (e.g. Cohen's *d*, Pearson's *r*), indicating how they were calculated |

*Our web collection on statistics for biologists contains articles on many of the points above.*

## Software and code

Policy information about availability of computer code

| Data collection | IVIS Spectrum for Bioluminescence imaging.<br>Leica DM IL inverted phase contrast microscope for Oncosphere imaging.<br>LSR Fortessa for Flow cytometry data.<br>GUAVA easyCyte HT for cell numbers.<br>Aria III BD cell sorter for cell sorting data.<br>Quant Studio 6 Flex System for Q_PCR data.<br>Axio Lab.A1 microscope for Cytospin data.<br>Zeiss LSM880 confocal microscope for Immunofluorescence ,wholemount imaging, time lapse imaging.<br>HiSeq 4000 for RNAseq data. |
|---|---|
| Data analysis | In vitro Oncosphere formation: ImageJ1.51.<br>Flow cytometry, cell sorting: FlowJo 10.6 .<br>Cell counting: GUAVA easyCyte HT.<br>Bioluminescence imaging: LivingImage 2.60.1.<br>Immunofluresence imaging, time lapse imaging: Imaris9.3.1.<br>Single cell RNAseq:Seurat" package (4.3.0) ,R studio (4.2.0).<br>BulkRNAseq:STAR v2.7.10a.DESeq2 1.34.0<br>Identification of candidate functional E2A and ELK1 motifs:Python 3, Keras 2.3.1, tensorflow 2.1.0, scikit-learn 0.21.3, deeplift 0.6.10.0, biopython 1.76. |

For manuscripts utilizing custom algorithms or software that are central to the research but not yet described in published literature, software must be made available to editors and reviewers. We strongly encourage code deposition in a community repository (e.g. GitHub). See the Nature Portfolio guidelines for submitting code & software for further information.

## Data

Policy information about availability of data

All manuscripts must include a data availability statement. This statement should provide the following information, where applicable:

- Accession codes, unique identifiers, or web links for publicly available datasets
- A description of any restrictions on data availability
- For clinical datasets or third party data, please ensure that the statement adheres to our policy

Public data used in this study, Single-cell RNA-seq analysis: The CRC dataset:GSE146409, The PDAC dataset : GSE205013, the nontumor control dataset : GSE115469. Identification of candidate functional E2A and ELK1 motifs: GSE128338.mouse reference genome GRCm39.human protein atlas (https://www.proteinatlas.org/). All the codes, analyzed RDS files, and the original data sets are available via the following link (https://doi.org/10.5281/zenodo.10121153).Bulk RNAseq raw data are deposited to the GEO repository with GEO accession number: GSE234638

## Research involving human participants, their data, or biological material

Policy information about studies with human participants or human data. See also policy information about sex, gender (identity/presentation), and sexual orientation and race, ethnicity and racism.

| | |
|---|---|
| Reporting on sex and gender | sex was not considered in this study |
| Reporting on race, ethnicity, or other socially relevant groupings | The study did not collect any information about race, ethnicity information |
| Population characteristics | Patient1: Age62/M,cTxNxM1 (Stage IV). Treatment: Taxoprexin (4 cycles); PD; Gemcitabine (1 cycle). Mutation: Kras, p.G12D; TP53, p.T155P. Patient2: Age57/M, cT3NxM1 (Stage IV).  Treatment: Taxoprexin (1 cycle); PD; Gemcitabine (4 cycles). Mutation:  Kras, p.Q61H; TP53: p.L257P. Patient3: Age60/M, no treatment. Mutation: Kras, p.G12D; TP53, p.L344P. |
| Recruitment | Autopsy from diagnosed PDAC patients. |
| Ethics oversight | Human tissues were obtained with patient-informed consent and used under approval by the Institutional Review Boards from Memorial Sloan Kettering Cancer Center(IRB protocols #15-021). |

Note that full information on the approval of the study protocol must also be provided in the manuscript.

# Field-specific reporting

Please select the one below that is the best fit for your research. If you are not sure, read the appropriate sections before making your selection.

☒ Life sciences ☐ Behavioural & social sciences ☐ Ecological, evolutionary & environmental sciences

For a reference copy of the document with all sections, see nature.com/documents/nr-reporting-summary-flat.pdf

# Life sciences study design

All studies must disclose on these points even when the disclosure is negative.

| | |
|---|---|
| Sample size | No statistical methods were used to predetermine sample size, Sample size for each experiment is indicated in the figures and figure legends. Sample sizes were estimated based on preliminary experiments, with an effort to achieve a minimum of n=3, mostly n=20 mice per treatment group, which proved sufficient to determine reproducible results. |
| Data exclusions | No data was excluded from the analysis presented in this study. |
| Replication | All experiments were repeated multiple times with similar results. Replicates were stated in the Figure Legends and methods/Statistical analysis.Key Observations from RNA-seq studies have been validated with Flow cytometry  and RT_qPCR. |
| Randomization | For in vivo experiments, mice were randomly assigned to different groups. |
| Blinding | For in-vivo study, blinding was performed during tumor cells injection and  tumor burden measurements, For Bulk RNAseq, time lapse imaging,  blinding was performed from sample collection to data analysis. For other experiments, investigators were not blind to group allocation as this information was essential for experiment conducting. |

# Reporting for specific materials, systems and methods

We require information from authors about some types of materials, experimental systems and methods used in many studies. Here, indicate whether each material, system or method listed is relevant to your study. If you are not sure if a list item applies to your research, read the appropriate section before selecting a response.

## Materials & experimental systems

| n/a | Involved in the study |
|-----|----------------------|
| ☐ | ☒ Antibodies |
| ☐ | ☒ Eukaryotic cell lines |
| ☒ | ☐ Palaeontology and archaeology |
| ☐ | ☒ Animals and other organisms |
| ☒ | ☐ Clinical data |
| ☒ | ☐ Dual use research of concern |
| ☒ | ☐ Plants |

## Methods

| n/a | Involved in the study |
|-----|----------------------|
| ☐ | ☒ ChIP-seq |
| ☐ | ☒ Flow cytometry |
| ☒ | ☐ MRI-based neuroimaging |

## Antibodies

Antibodies used

Antibodies are listed in Supplementary table 5.The following antibodies were used in immunofluorescence assay.
Primary antibodies:
anti-mouse-F4/80-eF450/AF647/AF488/eF570(1:200, BM8, eBioscience),
anti-mouse Tim4-AF647/PE(1:200, RMT4-54,Biolegend),
anti-mouse CD45.1-AF488(1:200, A20, Biolegend),
Chicken-anti-GFP (1:500, A10262,Invitrogen, recognize YFP),
Rabbit-anti-RFP (1:200, 600-401-379, Rockland),
Goat-anti mouse Clec4f (1:200, AF2784,R&D system),
anti-mouse CCL3(1:200, 50-7532-82,Thermo Fisher Scientific),
anti-mouse CCL4(1:200, AF-451-NA,Thermo Fisher Scientific),
anti-mouse CCL5(1:200, 701030,Thermo Fisher Scientific),
anti-mouse IL12p70(1:200, MM121B,Thermo Fisher Scientific),
anti-mouse IL15(1:200, AF447-SP, Thermo Fisher Scientific),
anti-mouse IL18(1:200, PA5-79481,Thermo Fisher Scientific) antibodies,
anti-human CD14-AF488(1:200, 561706, BD),
sheep-anti-human CK19(1:200, AF3506, R&D system),
rabbit-anti-human Tim4(1:200, PA5-53346, Thermo Fisher Scientific),
 Goat-anti-human IL12(1:200, AF-219-NA, R&D system),
Goat-anti-human IL18(1:200, AF2548, R&D system),
Mouse-anti-human IL15(1:200, MAB2471, R&D system),
Goat-anti-human CCL3(1:200, AF-270-NA, R&D system),
Goat-anti-human CCL4(1:200, AF-271-NA, R&D system),
Goat-anti-human CCL5(1:200, AF-278-NA, R&D system).
Secondary antibodies:
 anti-chicken Alexa Fluor 488 (1:500; a11039,Thermo Fisher Scientific),
anti-rabbit Alexa Fluor 555 (1:500, A32794 ,Thermo Fisher Scientific),
anti-rabbit Alexa Fluor 647(1:500, A32795, Thermo Fisher Scientific),
anti-goat Alexa Fluor 555 (1:500,a32816, Thermo Fisher Scientific),
anti-goat Alexa Fluor 647(1:500,a21447, Thermo Fisher Scientific),
anti-goat Alexa Fluor 488 (1:500,a32814,Thermo Fisher Scientific),
anti-sheep Alexa Fluor 568(1:500, A21099, Thermo Fisher Scientific),
Streptavidin Alexa Fluor 647 (1:500,405237, BioLegend).
anti-sheep Alexa Fluor 568(1:500, A21099, Thermo Fisher Scientific),
anti-mouse Alexa Fluor 555(1:500, A-31570, Thermo Fisher Scientific).
The following antibodies were used in Flow cytometry assay:
anti-mouse-CD45-APC-eF780(1:200, 30-F11,ebioscience),
anti-mouse-CD45.1-AF488/PE (1:200, A20,Biolegend),
anti-mouse-CD45.2-APC-cy7(1:200, 104,Biolegend),
anti-mouse-Tim4-AF647/PE(1:200, RMT4-54,Biolegend),
anti-mouse-CD11b-PE-Cy7(1:200, M1/70,BD Biosciences),
anti-mouse-F4/80-eF450/BV605(1:200, BM8,ebioscience),
anti-mouse-Gr1-BV711(1:200, RB6-8C5,BD Biosciences),
anti-mouse-I-A/I-E-AF700(1:200, M5/114.15.2,Biolegend),
anti-mouse-CD11c-BV605(1:200, N418,Biolegend),
anti-mouse-Ly6C-BV510/BV786/AF488(1:200, HK1.4,Biolegend),
anti-mouse-CD117 (c-KIT)-BV605(1:200, 2B8,BD Biosciences)
anti-mouse-CD64-BV711/APC(1:200, X54-5/7.1,Biolegend)
anti-mouse-CD115-BV605/APC(1:200, AFS98,Biolegend)
anti-mouse-CD19-BV711 (1:200, 1D3,Biolegend)
anti-mouse-CD19-BV711 (1:200, 1D3/CD19,Biolegend)
anti-mouse-CD3-BV711/BV421(1:200, 145-2C11,Biolegend)
anti-mouse-CD3-APC-eF780 (1:200,145-2C11,ebioscience)
anti-mouse-CD335-AF647/BV711/percp-cy5.5 (1:200,29A1.4,Biolegend)
anti-mouse-CD48-APC(1:200,HM48-1,Biolegend)
anti-mouse-CD150 (SLAM)-PE-Cy7(1:200,TC15-12F12.2,Biolegend)

anti-mouse-Siglec F-AF647/BV711(1:200,E50-2440,BD Biosciences)
anti-mouse-Ly6G-BV421/PE/Dazzle594(1:200,1A8,Biolegend)
anti-mouse-Sca1-BV421(1:200,D7,BD Biosciences)
anti-mouse-CD64-BV711(1:200,X54-5/7.1,Biolegend)
anti-mouse-CD16/32 (1:100, 93,Biolegend)
anti-mouse-CD47-FITC/PE/AF647(1:200,miap301,Biolegend)
anti-mouse-CD133-PE-Cy7(1:200,315-2C11,Biolegend)
anti-mouse-CD44-BV785(1:200,IM7,Biolegend)
anti-mouse-CD9-AF700(1:200,KMC 8,ebioscience)
anti-mouse-CD24-BV605(1:200,M1/69,Biolegend)
anti-mouse-Epcam-BV711(1:200,G8.8,Biolegend)
anti-mouse-Sirpa-AF488/PE/PE-Cy7 (1:200,P84,Biolegend)
anti-mouse-CD206-BV785(1:200,C068C2,Biolegend)
anti-mouse-Clec7a-PE/APC(1:200,RH1,Biolegend)
anti-mouse-CD1d PBS-57 tetrame-AF647(1:200,NIH Tetramer Core Facility)
anti-mouse-TCRβ-PE-Cy7(1:200,H57-597,BD Biosciences)
anti-mouse-TCRγδ-BV786(1:200,GL3,BD Biosciences)
anti-mouse-CD4-BV421(1:200,RM4-5,Biolegend)
anti-mouse-CD8-BV605/AF647(1:200,53-6.7,Biolegend)
anti-mouse-IFNg-PE-Cy7(1:200,XMG1.2,Biolegend)
anti-mouse-TNFa-AF647(1:200,MP6-XT22,Biolegend)
anti-human-CD8-PE(1:200,SK1,Biolegend )
anti-human-CD56-PE(1:200,HCD56,Biolegend)
anti-human-IFNg-APC(1:200,B27,BD Biosciences)
anti-human-TNFa-BV650(1:200,MAb11,Biolegend)

**Validation**

all antibodies used are commercially available and catalog numbers/clones are stated in Supplementary table 5, detailed as follow:
anti-mouse-CD45(30-F11,ebioscience) https://www.thermofisher.com/antibody/product/CD45-Antibody-clone-30-F11-Monoclonal/47-0451-82?
gclid=CjwKCAiA0syqBhBxEiwAeNx9N0Wh3nljQ1xtqQcJSwawK0O6o3sfnM6ldsEKIcKH4aP4Jn5iVIWxEBoC1ZUQAvD_BwE&ef_id=CjwKCAiA0syqBhBxEiwAeNx9N0Wh3nljQ1xtqQcJSwawK0O6o3sfnM6ldsEKIcKH4aP4Jn5iVIWxEBoC1ZUQAvD_BwE:G:s&s_kwcid=AL!3652!3!278870232429!!!g!!!1454324556!63404918784&cid=bid_pca_frg_r01_co_cp1359_pjt0000_bid00000_0se_gaw_dy_pur_con
anti-mouse-CD45.1(A20,Biolegend) https://www.biolegend.com/en-us/products/alexa-fluor-488-anti-mouse-cd45-1-antibody-3103
anti-mouse-CD45.2(104,Biolegend) https://www.biolegend.com/en-us/products/apc-cyanine7-anti-mouse-cd45-2-antibody-3906
anti-mouse-Tim4(RMT4-54,Biolegend) https://www.biolegend.com/en-us/products/alexa-fluor-647-anti-mouse-tim-4-antibody-5243
anti-mouse-CD11b(M1/70,BD Biosciences) https://www.bdbiosciences.com/en-us/products/reagents/flow-cytometry-reagents/research-reagents/single-color-antibodies-ruo/pe-cy-7-rat-anti-cd11b.552850
anti-mouse-F4/80(BM8,ebioscience) https://www.thermofisher.com/antibody/product/F4-80-Antibody-clone-BM8-Monoclonal/48-4801-82
anti-mouse-Gr1(RB6-8C5,BD Biosciences) https://www.bdbiosciences.com/en-us/products/reagents/flow-cytometry-reagents/research-reagents/single-color-antibodies-ruo/bv711-rat-anti-mouse-ly-6g-and-ly-6c.740658
anti-mouse-I-A/I-E(M5/114.15.2,Biolegend) https://www.biolegend.com/en-us/products/alexa-fluor-700-anti-mouse-i-a-i-e-antibody-3413
anti-mouse-CD11c(N418,Biolegend) https://www.biolegend.com/en-us/products/brilliant-violet-605-anti-mouse-cd11c-antibody-7865
anti-mouse-Ly6C(HK1.4,Biolegend) https://www.biolegend.com/en-us/products/brilliant-violet-785-anti-mouse-ly-6c-antibody-11982
anti-mouse-CD117 (c-KIT)(2B8,BD Biosciences) https://www.biolegend.com/en-us/products/brilliant-violet-605-anti-mouse-cd117-c-kit-antibody-16969
anti-mouse-CD64(X54-5/7.1,Biolegend) https://www.biolegend.com/en-us/products/brilliant-violet-711-anti-mouse-cd64-fcgammari-antibody-9920
anti-mouse-CD115(AFS98,Biolegend) https://www.biolegend.com/en-us/products/brilliant-violet-605-anti-mouse-cd115-csf-1r-antibody-9013
anti-mouse-CD19(1D3,Biolegend) https://www.bdbiosciences.com/en-us/products/reagents/flow-cytometry-reagents/research-reagents/single-color-antibodies-ruo/bv711-rat-anti-mouse-cd19.563157
anti-mouse-CD19(1D3/CD19,Biolegend) https://www.biolegend.com/en-us/products/alexa-fluor-700-anti-mouse-cd19-antibody-22036
anti-mouse-CD3(145-2C11,Biolegend) https://www.biolegend.com/en-us/products/brilliant-violet-711-anti-mouse-cd3epsilon-antibody-11975
anti-mouse-CD3(145-2C11,ebioscience) https://www.thermofisher.com/antibody/product/CD3e-Antibody-clone-145-2C11-Monoclonal/47-0031-82
anti-mouse-CD335 (NKp46,Biolegend) https://www.bdbiosciences.com/en-us/products/reagents/flow-cytometry-reagents/research-reagents/single-color-antibodies-ruo/bv711-rat-anti-mouse-cd335-nkp46.740822
anti-mouse-CD48(29A1.4,Biolegend) https://www.biolegend.com/en-us/products/apc-anti-mouse-cd48-antibody-3622
anti-mouse-CD150 (SLAM)(TC15-12F12.2,Biolegend) https://www.biolegend.com/en-us/products/pe-cyanine7-anti-mouse-cd150-slam-antibody-3056
anti-mouse-Siglec F(E50-2440,BD Biosciences) https://www.bdbiosciences.com/en-us/products/reagents/flow-cytometry-reagents/research-reagents/single-color-antibodies-ruo/alexa-fluor-647-rat-anti-mouse-siglec-f.562680
anti-mouse-Ly6G(1A8,Biolegend) https://www.biolegend.com/en-us/products/pe-dazzle-594-anti-mouse-ly-6g-antibody-12246
anti-mouse-Sca1(D7,BD Biosciences) https://www.bdbiosciences.com/en-us/products/reagents/flow-cytometry-reagents/research-reagents/single-color-antibodies-ruo/bv421-rat-anti-mouse-ly-6a-e.562729
anti-mouse-CD64(X54-5/7.1,Biolegend) https://www.biolegend.com/en-us/products/brilliant-violet-711-anti-mouse-cd64-fcgammari-antibody-9920
anti-mouse-CD16/32 (93,Biolegend) https://www.biolegend.com/en-us/products/purified-anti-mouse-cd16-32-antibody-190
anti-mouse-CD47(miap301,Biolegend) https://www.biolegend.com/en-us/products/pe-anti-mouse-cd47-antibody-4926
anti-mouse-CD133(315-2C11,Biolegend) https://www.biolegend.com/en-us/products/pe-cyanine7-anti-mouse-cd133-antibody-10193

anti-mouse-CD44(IM7,Biolegend) https://www.biolegend.com/en-us/products/brilliant-violet-785-anti-mouse-human-cd44-antibody-7959

anti-mouse-CD9(KMC 8,ebioscience) https://www.thermofisher.com/antibody/product/CD9-Antibody-clone-eBioKMC8-KMC8-Monoclonal/56-0091-82

anti-mouse-CD24(M1/69,Biolegend) https://www.biolegend.com/en-us/products/brilliant-violet-605-anti-mouse-cd24-antibody-9691

anti-mouse-Epcam(G8.8,Biolegend) https://www.biolegend.com/en-us/products/brilliant-violet-711-anti-mouse-cd326-ep-cam-antibody-13763

anti-mouse-Sirpa(P84,Biolegend) https://www.biolegend.com/en-us/products/alexa-fluor-488-anti-mouse-cd172a-sirpalpha-antibody-14089

anti-mouse-CD206(C068C2,Biolegend) https://www.biolegend.com/en-us/products/brilliant-violet-785-anti-mouse-cd206-mmr-antibody-12013

anti-mouse-Clec7a(RH1,Biolegend) https://www.biolegend.com/en-us/products/pe-anti-mouse-cd369-dectin-1-clec7a-antibody-8102

anti-mouse-TCRβ(H57-597,BD Biosciences) https://www.bdbiosciences.com/en-us/products/reagents/flow-cytometry-reagents/research-reagents/single-color-antibodies-ruo/pe-cy-7-hamster-anti-mouse-tcr-chain.560729

anti-mouse-TCRγδ(GL3,BD Biosciences) https://www.bdbiosciences.com/en-us/products/reagents/flow-cytometry-reagents/research-reagents/single-color-antibodies-ruo/bv786-hamster-anti-mouse-t-cell-receptor.744117

anti-mouse-CD4(RM4-5,Biolegend) https://www.biolegend.com/en-us/products/brilliant-violet-421-anti-mouse-cd4-antibody-7349

anti-mouse-CD8(53-6.7,Biolegend) https://www.biolegend.com/en-us/products/brilliant-violet-605-anti-mouse-cd8a-antibody-7636

anti-mouse-IFNg(XMG1.2,Biolegend) https://www.biolegend.com/en-us/products/pe-cyanine7-anti-mouse-ifn-gamma-antibody-5865

anti-mouse-TNFa(MP6-XT22,Biolegend) https://www.biolegend.com/en-us/products/alexa-fluor-647-anti-mouse-tnf-alpha-antibody-2724

anti-human-CD8(SK1,Biolegend ) https://www.biolegend.com/en-us/products/pe-anti-human-cd8-antibody-6247

anti-human-CD56(HCD56,Biolegend) https://www.biolegend.com/en-us/products/pe-anti-human-cd56-ncam-antibody-3796

anti-human-IFNg(B27,BD Biosciences) https://www.bdbiosciences.com/en-us/products/reagents/flow-cytometry-reagents/research-reagents/single-color-antibodies-ruo/apc-mouse-anti-human-ifn.562017

anti-human-TNFa(MAb11,Biolegend) https://www.biolegend.com/en-us/products/brilliant-violet-650-anti-human-tnf-alpha-antibody-7680

Rabbit-anti-mouse-Lamb1(ab24170,Abcam) https://www.abcam.com/products/primary-antibodies/lamp1-antibody-lysosome-marker-ab24170.html

Rat-anti-mouse-Lamp1(1D4B,Biolegend) https://www.biolegend.com/en-us/products/alexa-fluor-647-anti-mouse-cd107a-lamp-1-antibody-3589

Goat-anti-mouse-Clec4f(AF2784,R&D system) https://www.rndsystems.com/products/mouse-clec4f-clecsf13-antibody_af2784?gclid=CjwKCAiA0syqBhBxEiwAeNx9NzuVAyZtHcQ2t2mAX-HxQZON-V4an6sh-7454lXYNRAJ5N6veglKKxoCHHsQAvD_BwE&gclsrc=aw.ds

Rabbit-anti-mouse-Elk1(9182S,Cell signaling technology) https://www.cellsignal.com/products/primary-antibodies/elk-1-antibody/9182

Rat-anti-mouse-CCL3(50-7532-82,Thermo Fisher Scientific) https://www.thermofisher.com/antibody/product/CCL3-MIP-1-alpha-Antibody-clone-DNT3CC-Monoclonal/50-7532-82

Goat-anti-mouse-CCL4(AF-451-NA,R&D system) https://www.rndsystems.com/products/mouse-ccl4-mip-1beta-antibody_af-451-na?gclid=CjwKCAiA0syqBhBxEiwAeNx9N83iPzTpnzHXEO1KGjN-zOqjiuMOP4ewN1ty2NgN0GAz7s691FOeLxoC8TAQAvD_BwE&gclsrc=aw.ds

Rabbit-anti-mouse-CCL5(701030,Thermo Fisher Scientific) https://www.thermofisher.com/antibody/product/CCL5-RANTES-Antibody-clone-25H14L17-Recombinant-Monoclonal/701030

Rat-anti-mouse-IL12p70(MM121B,Thermo Fisher Scientific) https://www.thermofisher.com/antibody/product/IL-12-p70-Antibody-clone-C17-8-Monoclonal/MM121B

Goat-anti-mouse-IL15(AF447-SP,R&D system) https://www.rndsystems.com/products/mouse-il-15-antibody_af447

Rabbit-anti-mouse-IL18(PA5-79481) https://www.thermofisher.com/antibody/product/IL-18-Antibody-Polyclonal/PA5-79481

Mouse-anti-human CD14(M5E2,BD Biosciences) https://www.bdbiosciences.com/en-us/products/reagents/flow-cytometry-reagents/research-reagents/single-color-antibodies-ruo/alexa-fluor-488-mouse-anti-human-cd14.561706

Rabbit-anti-human TIMD4(PA5-53346,Thermo Fisher Scientific) https://www.thermofisher.com/antibody/product/TIMD4-Antibody-Polyclonal/PA5-53346

Sheep-anti-human CK19(AF3506,R&D system) https://www.rndsystems.com/products/human-cytokeratin-19-antibody_af3506

Mouse-anti-human CK19(ab7754,Abcam) https://www.abcam.com/products?keywords=Cytokeratin+19&gad=1&gclid=CjwKCAiA0syqBhBxEiwAeNx9N3BdLDzd2ZIGgf3XWLlxor2FvGnraTGiBLFpLZzOdH6LyL-Ky5C8URoCPIQQAvD_BwE&gclsrc=aw.ds

Goat-anti-human IL12(AF-219-NA,R&D system) https://www.rndsystems.com/products/human-il-12-antibody_af-219-na

Goat-anti-human IL18(AF2548,R&D system) https://www.rndsystems.com/products/human-rhesus-macaque-il-18-il-1f4-antibody_af2548

Mouse-anti-human IL15(MAB2471,R&D system) https://www.rndsystems.com/products/human-il-15-antibody-34559_mab2471

Goat-anti-human CCL3(AF-270-NA,R&D system) https://www.rndsystems.com/products/human-ccl3-mip-1alpha-antibody_af-270-na

Goat-anti-human CCL4(AF-271-NA,R&D system) https://www.rndsystems.com/products/human-ccl4-mip-1beta-antibody_af-271-na

Goat-anti-human CCL5(AF-278-NA,R&D system) https://www.rndsystems.com/products/human-ccl5-rantes-antibody_af-278-na

Goat anti-rabbit IgG(a11034,Thermo Fisher Scientific) https://www.thermofisher.com/antibody/product/Goat-anti-Rabbit-IgG-H-L-Highly-Cross-Adsorbed-Secondary-Antibody-Polyclonal/A-11034

Goat anti-rabbit IgG(a32795,Thermo Fisher Scientific) https://www.thermofisher.com/antibody/product/Donkey-anti-Rabbit-IgG-H-L-Highly-Cross-Adsorbed-Secondary-Antibody-Polyclonal/A32795

Goat anti-rabbit IgG(a21430,Thermo Fisher Scientific) https://www.thermofisher.com/antibody/product/Goat-anti-Rabbit-IgG-H-L-Cross-Adsorbed-Secondary-Antibody-Polyclonal/A-21430

Donkey anti-goat IgG(a32816,Thermo Fisher Scientific) https://www.thermofisher.com/antibody/product/Donkey-anti-Goat-IgG-H-L-Highly-Cross-Adsorbed-Secondary-Antibody-Polyclonal/A32816

Donkey anti-goat IgG(a32814,Thermo Fisher Scientific) https://www.thermofisher.com/antibody/product/Donkey-anti-Goat-IgG-H-L-Highly-Cross-Adsorbed-Secondary-Antibody-Polyclonal/A32814

Donkey anti-goat IgG(a21447,Thermo Fisher Scientific) https://www.thermofisher.com/antibody/product/Donkey-anti-Goat-IgG-H-L-Cross-Adsorbed-Secondary-Antibody-Polyclonal/A-21447

Goat-anti-Chicken IgG(a11039,Thermo Fisher Scientific) https://www.thermofisher.com/antibody/product/Goat-anti-Chicken-IgY-H-L-Secondary-Antibody-Polyclonal/A-11039
Donkey anti-Sheep IgG(A21099,Thermo Fisher Scientific) https://www.thermofisher.com/antibody/product/Donkey-anti-Sheep-IgG-H-L-Cross-Adsorbed-Secondary-Antibody-Polyclonal/A-21099
Donkey anti-Rabbit IgG (A32795,Thermo Fisher Scientific) https://www.thermofisher.com/antibody/product/Donkey-anti-Rabbit-IgG-H-L-Highly-Cross-Adsorbed-Secondary-Antibody-Polyclonal/A32795
Donkey anti-Rabbit IgG (A32794,Thermo Fisher Scientific) https://www.thermofisher.com/antibody/product/Donkey-anti-Rabbit-IgG-H-L-Highly-Cross-Adsorbed-Secondary-Antibody-Polyclonal/A32794
Donkey anti-Mouse IgG(A32766TR,Thermo Fisher Scientific) https://www.thermofisher.com/antibody/product/Donkey-anti-Mouse-IgG-H-L-Highly-Cross-Adsorbed-Secondary-Antibody-Polyclonal/A32766TR
Donkey anti-Mouse IgG(A-31570,Thermo Fisher Scientific) https://www.thermofisher.com/antibody/product/Donkey-anti-Mouse-IgG-H-L-Highly-Cross-Adsorbed-Secondary-Antibody-Polyclonal/A-31570
Streptavidin(405237,BioLegend) https://www.biolegend.com/en-us/products/alexa-fluor-647-streptavidin-9305
Chicken-anti-GFP(A10262,Thermo Fisher Scientific) https://www.thermofisher.com/antibody/product/GFP-Antibody-Polyclonal/A10262
Rabbit-anti-RFP(600-401-379,Rockland) https://www.rockland.com/categories/primary-antibodies/rfp-antibody-pre-adsorbed-600-401-379/?gad=1&gclid=CjwKCAiA0syqBhBxEiwAeNx9N5tsuMVFbCe23BzyQPUmvdS86suX3gpnUnuEkHqWfhXC07qhZyY_RRoC_fMQAvD_BwE
anti-mouse Dectin1(blockage)(R1-8g7,InvivoGen) https://www.invivogen.com/anti-mdectin1
anti-mouse-NK1.1(blockage)(BE0036,BioXcell) https://bioxcell.com/invivomab-anti-mouse-nk1-1-be0036?gad=1&gclid=CjwKCAiA0syqBhBxEiwAeNx9Nxt8Qgd8eOvuEcllXGCPClfYhuElX_wLypwqhBWJP5c7r1yn7QPJPBoCtVgQAvD_BwE
anti-mouse-CD8a(blockage)(BP0061,BioXcell) https://bioxcell.com/invivoplus-anti-mouse-cd8-alpha-bp0061
Control IgG(HRPN,BioXcell) https://bioxcell.com/invivomab-rat-igg1-isotype-control-anti-horseradish-peroxidase-be0088
Sirpa(P84,BioXcell) https://bioxcell.com/invivomab-anti-mouse-cd172a-sirp-alpha-be0322

# Eukaryotic cell lines

Policy information about cell lines and Sex and Gender in Research

| | |
|---|---|
| Cell line source(s) | Mouse cell lines: KPC-1, KPC-2,PAN02,MC38,B16F10,LLC1.Human cell lines: PANC-1. KPC-1,2 cancer cell lines derived from KPC pancreatic tumors by passaging in cell culture. |
| Authentication | Morphology , antibodies stain and PCR validate cell lines. |
| Mycoplasma contamination | No contamination |
| Commonly misidentified lines (See ICLAC register) | No commonly misidentified lines. |

# Animals and other research organisms

Policy information about studies involving animals; ARRIVE guidelines recommended for reporting animal research, and Sex and Gender in Research

| | |
|---|---|
| Laboratory animals | Materials and Methods/Mice.Rosa26LSL-YFP , Rosa26LSL-tdT , Flt3Cre , Id3f/f , Id3-/- , Cx3cr1gfp/+ , CCR2-/- , Csf1rf/f , Spi1f/f , Cxcr4 gfp/+ , Cxcr4CreERT2 , Clec4fCre-tdT , Rosa26LSL-DTR , p48Cre , p53LSL-R172H , KrasLSL-G12D,CD45.1,C57BL/6J,Mice were housed in groups of no more than 5 per cage in standard closed plastic cages containing bedding, food and water. The room is controlled at stable temperature ~21-22°C with 30-70% humidity, 12light/12dark cycle(on from 6:00-18:00), 10-15 fresh air exchanges hourly. |
| Wild animals | No wild animals were employed in the study. |
| Reporting on sex | Sex was not considered in this study. |
| Field-collected samples | No field-collected samples were used in this study. |
| Ethics oversight | Materials and Methods/Mice. Animal procedures were performed in adherence with the Institutional Review Board (IACUC 15-04-006) at Memorial Sloan Kettering Cancer Center (MSKCC). |

Note that full information on the approval of the study protocol must also be provided in the manuscript.

# Plants

| | |
|---|---|
| Seed stocks | N.A. |
| Novel plant genotypes | N.A. |
| Authentication | N.A. |

# ChIP-seq

## Data deposition

☒ Confirm that both raw and final processed data have been deposited in a public database such as GEO.

☒ Confirm that you have deposited or provided access to graph files (e.g. BED files) for the called peaks.

| | |
|---|---|
| Data access links<br>*May remain private before publication.* | CHIPseq data are shown in Extended data Fig8c,d |
| Files in database submission | Available in GEO:GSE128338. ref.PMID: 32362324. |
| Genome browser session<br>(e.g. UCSC) | UCSC |

## Methodology

| | |
|---|---|
| Replicates | Materials and Methods/Identification of candidate functional E2A and ELK1 motifs. |
| Sequencing depth | libraries were size selected 250-500 bp using gel extraction using 10% TBE acrylamide gels. Libraries were single-end sequenced using either a HiSeq 4000 or a NextSeq 500 to a depth of 10-20 million reads. |
| Antibodies | Antibodies are shown in ref.PMID: 32362324. |
| Peak calling parameters | Peak files were merged with HOMER's mergePeaks and annotated with raw tag counts with HOMER's annotatePeaks using parameters -noadj, -size given. To annotate H3K27ac signal around ATAC-seq peaks the parameter -size 2000 was used. |
| Data quality | DESeq2 was used to identify the differentially bound H3K27ac signal or chromatin accessibility with FC > 2 and p-adj < 0.05. D |
| Software | Materials and Methods/Identification of candidate functional E2A and ELK1 motifs. |

# Flow Cytometry

## Plots

Confirm that:

☒ The axis labels state the marker and fluorochrome used (e.g. CD4-FITC).

☒ The axis scales are clearly visible. Include numbers along axes only for bottom left plot of group (a 'group' is an analysis of identical markers).

☒ All plots are contour plots with outliers or pseudocolor plots.

☒ A numerical value for number of cells or percentage (with statistics) is provided.

## Methodology

| | |
|---|---|
| Sample preparation | Materials and Methods/Flow cytometry, cell sorting, cell counting. |
| Instrument | Flow cytometry was performed using a BD Biosciences LSRFortessa flow cytometer with Diva software. |
| Software | All data were analyzed using FlowJo 10.6 (Tree Star Inc.). |
| Cell population abundance | The purity of sorted samples was analyzed by flow cytometry on the same instruments used for sorting. |

| Gating strategy | Materials and Methods/Flow cytometry, cell sorting, cell counting/Gating strategy.  Extended Data Fig1e, Extended Data Fig2j, Supplementary information Figure 1, 2, 3. |

☒ Tick this box to confirm that a figure exemplifying the gating strategy is provided in the Supplementary Information.

