## [Peer Review File · Nature]

Manuscript Title: The nuclear factor ID3 endows macrophages with a potent anti-tumor activity

Reviewer Comments & Author Rebuttals

Reviewer Reports on the Initial Version:

Referees' comments:

Referee #1 (Remarks to the Author):

In this paper, Deng et al examine the role of Id3 in Kupffer cells in protecting against tumors in the liver. Using a variety of tumor models, fate mapping, intravital microscopy and molecular analysis, the authors show that Kupffer cells can play critical roles in preventing tumor growth in the liver. This may involve a number of functions including phagocytosis and killing of the cancer cells, as well as recruitment of effector lymphocytes such as CD8+ T cells and NK T cells. As well as being crucial for the early development of Kupffer cells, the work shows for the first time that Id3 is also important for the acquisition of these tumor-protective properties in mature Kupffer cells. This involves downregulation of inhibitory SIRP α expression via preventing the binding of ELK1 and E2A transcription factors to the SIRP α locus. The studies are elegant, using a number of powerful and precise experimental tools which are state of the art in macrophage biology and together they allow clear conclusions to be made. The data are novel and although their application to tumor therapy remains a longer term probability, they add significantly to our knowledge of an important cell type, as well as enhancing understanding of the liver immune niche and of macrophage specification in general. Some specific comments:

- 1) While the authors take considerable pains to ensure that bona fide Kupffer cells are being assessed, can they exclude the possibility that at least some of the functions being imprinted during tumor growth and/or after deletion of Kupffer cells in vivo reflect the presence of inflammation? In a similar vein, given the known limitations of the fate mapping tools and parabiosis system, how sure can they be that there is not some contribution of recruited monocytes to the Kupffer cell pool being analyzed?
- 2) The reason for describing "CD47bright CD9+ CD133+ phenotypic metastasis initiating cells" warrants some introduction and explanation.
- 3) The fate mapping data shown in Supplementary Figs 2h-o are very impressive and informative, but the various approaches are not described or explained anywhere in the text or legends and I suspect these may not be particularly familiar to readers who are not specialists in macrophage biology. If the authors consider it necessary to retain all this information, it needs explained in more detail.
- 4) Much of the intravital imaging is extremely elegant and convincing. However it is very difficult to see any green staining in most of the images supposed to show Cas-Green staining in Figures 2H or Supplementary S4. One or two seem to show a few 2 small dots that are apparently associated with a tumor cell and not a Kupffer cell. It was unclear to me from the text what this approach was aiming to show – is it to illustrate Kupffer cells that have taken up dying tumor cells, or is simply to label any dead tumor cell? This needs explained and the relevant images should be improved, with appropriate annotation provided.
- 5) A related issue with the intravital staining is that I could not see any pink staining for ELK1 in Supplementary Figure S7c.
- 6) No data are shown for Id3fl/fl mice treated with anti-SIRP α in Figure 5, which I believe would be important to provide as baseline controls for the effects in Clec4fcre; Id3fl/fl mice.
- 7) A further issue with the intravital staining is that I could not see any pink staining for ELK1 in Supplementary Figure S7c
- 8) The English needs attention in places. For instance, there are often issues with the use of plural

nouns/verbs, while the word "phagocyte" should be "phagocytose" when used as a verb.

Referee #2 (Remarks to the Author):

This is a well-written, (primarily mouse-based) experimental study highlighting the role of Kupffer cells (i.e. liver-resident macrophages) as a gatekeeper to limit metastatic malignancies.

1. The series of experiments is certainly elegant and convincing. An important question that remains open, to my opinion, is the translatability of the findings to human disease. Clec4f is restricted to mouse KCs, which makes it much more challenging to identify and study human KCs. However, some important questions remain fully open: Do human KCs also organize themselves around liver metastases as in the mouse models? Do they share the same pattern of high ID3, low SIRPa and high tumor cell phagocytosis? Would their presence or absence indicate prognosis / metastatic spread?

2. Regarding the features of tumor-associated macrophages in human liver, a comparative scRNA-seq (+ spatial analysis) of human liver with metastasis (with proper control conditions) and the mouse models may be very informative to substantiate the claims on the translational value of the study. The scRNA-seq studies that are quoted in this study (ref #93/94) do not reflect this situation at all.

3. Along the thought of translatability, the authors use very simple and "immunogenic" tumor models, primarily the injection of tumor cell lines. This approach is highly prone to biases, as it may over-estimate the phagocytic capacity of Kupffer cells. KCs are well known to be highly phagocytic and may be particularly active upon intraportal injection of tumor cell lines, as done in this experimental setting. An endogenous tumor model with spontaneous metastasis at a late stage (in mice) will be way more reflective of tumor biology than the models used in this study.

4. The mechanisms of "anti-tumor activity" of KCs were not fully clear to me. On the one hand, the phagocytosis of tumor cells may be involved (which is probably less relevant in the cases of established tumors?); on the other hand, the expression of lymphocyte-attracting chemokines could promote the NK- and CD8 T cell responses. What is the contribution of each mechanism? How does the chemokine expression explain the anti-tumor activity of the NK and CD8 T cells? What I mean here: is the attraction enough or is there a KC-dependent mechanism of activation (e.g. antigen presentation).

5. The authors claim that BMDMs or human iPSC-derived macrophages can be reprogrammed towards KCs by ID3 and would then exert anti-tumor activity. I found this part particularly weak, since the experimental system appears quite artificial and involves high numbers of adoptively transferred, genetically modified cells.

6. Many tissues have "resident macrophages" – I would find it very interesting and relevant, whether similar mechanisms exist, for instance, in the brain or in the lung to limit metastasis. Simply looking at single factors like SIRPa may not give the correct answer here, since KCs are quite particular in this sense (e.g. Clec4F, ID3 etc. very specific to KCs).

Referee #3 (Remarks to the Author):

Focusing on Kupffer cells (KCs), Deng et al. demonstrate the role of ID3 in the anti-tumor functions of these macrophages in mice. Using multiple elaborate experimental tools, the authors showed that ID3, whose expression is mainly restricted to KCs, prevents Sirp-a expression by

interfering with the DNA binding of two activating transcription factors (E2A and ELK1). Interestingly, ectopic expression of ID3 in monocyte-macrophages prevents Sirp-a expression and enhances the anti-tumor functions of the cells, opening up potential therapeutic implications. Overall, the authors are to be commended for the quality of their study, which should be of interest to the cancer research community. However, there are aspects of this work that need to be completed to support and clarify the findings presented.

Experimental Design:

- All experiments were performed using mice with loxP sites flanking the genes of interest as controls. What are the effects of recombinase in these mice? Additional data are needed to illustrate the effects (if any) mediated by the recombinase itself.

- Overall survival experiments are needed to define whether ID3 expression by KCs has a major inhibitory effect on tumor progression.

ID3 and phagocytosis:

- The authors set up an elegant in vitro assay to study the uptake of tumor cells by KCs. Although phagocytosis of apoptotic cancer cells is excluded, it will be useful to further visualize and monitor the phagocytosis action mediated by these cells. The use of inhibitors, the visualization of actin cytoskeleton rearrangements, or the assessment of fluorescence of tumor debris in phagosomes will provide important information about the mechanism of action being studied.

- The data suggest differences between the KCs of non-tumor-bearing mice and those of tumor-bearing mice. But what are the functional differences between these KCs? How does the presence of a tumor alter their phenotypes and functions?

- ID3 appears to simultaneously stimulate upregulation of Dectin-1 and downregulation of Sirp-a. Does Sirp-a down-regulation alone promote phagocytosis? Are other molecules (phagocytic receptors) involved and what are their roles? At least a more thorough study of Dectin-1 is needed.

ID3 and effector lymphoid response:

- Since KCs can be a major source of IL-12 in the liver (Seki et al, The Journal of Immunology, 2001, Hou et al, Cellular & Molecular Immunology, 2016, Yong et al, Scientific reports, 2017, Siwicki et al, Science Immunology, 2021), and IL-12 can promote antitumor T cell-mediated immunity, it is possible that this signaling pathway plays an important role in the antitumor functions of KCs described in this study. In this context, what is the involvement of ID3 in the secretion of IL-12 by KCs, and the stimulation of an effector lymphoid response?

- Intratumoral injection of BMDMs expressing ID3 reduced local tumor growth and was associated with a higher frequency of intratumoral CD8 T cells, compared to contralateral tumors. But it is not clear from the data presented whether this injection of BMDMs could trigger a systemic effect (and delay the growth of the contralateral tumor). Specifically, what are the systemic effects induced by this adoptive transfer compared to untreated mice? As ID3 expression by macrophages limits tumor growth and metastasis, it would be important to clarify this point to support a potential therapeutic application of the presented approach.

Minor comments:

- Fig 1.g: Could the authors show the effectiveness of depletion at different time points?

- Fig6a: There appears to be a color reversal between red and blue. E2A inhibits macrophage activation and ID3 activates it. Could the authors please check?

- The text contains a fair number of typographical errors. Could the authors please check and

correct all these errors before resubmitting the manuscript?

Author Rebuttals to Initial Comments:

Referee #1: In this paper, Deng et al examine the role of Id3 in Kupffer cells in protecting against tumors in the liver. Using a variety of tumor models, fate mapping, intravital microscopy and molecular analysis, the authors show that Kupffer cells can play critical roles in preventing tumor growth in the liver. This may involve a number of functions including phagocytosis and killing of the cancer cells, as well as recruitment of effector lymphocytes such as CD8+ T cells and NK T cells. As well as being crucial for the early development of Kupffer cells, the work shows for the first time that Id3 is also important for the acquisition of these tumor-protective properties in mature Kupffer cells. This involves downregulation of inhibitory SIRPa expression via preventing the binding of ELK1 and E2A transcription factors to the SIRPa locus. The studies are elegant, using a number of powerful and precise experimental tools which are state of the art in macrophage biology and together they allow clear conclusions to be made. The data are novel and although their application to tumor therapy remains a longer-term probability, they add significantly to our knowledge of an important cell type, as well as enhancing understanding of the liver immune niche and of macrophage specification in general. Some specific comments:

1) While the authors take considerable pains to ensure that bona fide Kupffer cells are being assessed, can they exclude the possibility that at least some of the functions being imprinted during tumor growth and/or after deletion of Kupffer cells in vivo reflect the presence of inflammation?

We used several genetic models to assess the roles of KC in tumor growth. Inducible deletion of KC after DT treatment in *Clec4f^{Cre}R26^{LSL-DTR}*, which increases tumor growth, also cause a transient recruitment of monocytes, but not of neutrophils (Sakai et al., Immunity 2019 Fig1 and S1G), which could contribute to the phenotype at least in theory. Nevertheless, we obtained similar results following genetic deletion of ID3 in KC, which does not result in a liver inflammatory phenotype detectable by histology (Mass et al., 2016) and we did not find differences in leukocytes numbers in the liver, blood, or spleen of *Clec4f^{Cre}ID3^{fl/fl}* mice in comparison to control (Revised **Ext.data Fig.6I**). We find tumor cells elicit an 'inflammatory' response by KC characterized for example by increased expression of phagocytic receptors, chemokines, and cytokines and the recruitment and activation of CD8 T Cells and NK cells (Revised **Fig.2c-n**), but not of neutrophils and monocytes (Revised **Ext.data Fig6I**). The absence of ID3 in KC, results in decreased reduction of phagocytic receptors, chemokines, and cytokine expression, (Revised **Fig.4a-d**), decreased effector lymphoid cells NK cell and CD8 T cell recruitment and production of IFN γ and TNF(**Fig.4h-l**), but did not affect granulocytes, monocytes, or CD4 T cells numbers (Revised **Ext.data Fig.6I**). Therefore, although it is impossible to exclude possible effects of KC depletion on inflammation, our results suggest that post-natal deletion of ID3 in KC regulates their inflammatory response to tumor cells, without increasing inflammation in the liver.

Of note, in the revised manuscript we extended our findings, showing that tumor cells also upregulate the production of cytokines by KC, specifically Il12, Il15, and Il18, at the mRNA and protein level, and which is associated with KC-dependent production of IFN γ and TNF by recruited NK and CD8 T cells (**Revised Fig.2k-n, Fig.4h-l, Fig.5f,j, Fig.6k, Ext.data Fig.5c, Ext.data Fig.6j.k, Ext.data Fig. 7a-d**).

In a similar vein, given the known limitations of the fate mapping tools and parabiosis system, how sure can they be that there is not some contribution of recruited monocytes to the Kupffer cell pool being analyzed?

AU- Parabiosis underestimate the contribution of blood-derived cells to a given organ because of the partial chimerism in blood that results from the procedure. Nevertheless, as now shown in the **revised Ext.data Fig.2i**, circulating cells from the parabiosis partner contributed to <0.5% of KC (as defined phenotypically by Tim4⁺ Clec4f⁺ cells located in the liver) 8 weeks after parabiosis and 2 weeks after intraportal tumor injection, in both controls and tumor bearing mice (without significant difference between controls and tumor bearing mice), to be compared with ~25% chimerism observed in CD45⁺ Tim4⁻ cells, which is a fifty time higher. Assuming 25% chimerism correspond to a total replacement of liver/tumor CD45⁺ Tim4⁻ cells from the blood, the contribution of blood cells to KC would be estimated at a maximum of ~ 1%. Independent lineage tracing approaches in tumor free and tumor bearing mice in 3 genetic models (*Cx3cr1^{gfp}* mice, *Cxcr4^{gfp}* mice, and *Cxcr4^{CreERT2}R26^{LSL-tdT}* mice pulsed with OH-TAM at 6 weeks of age), gave comparable results (**revised Ext.data Fig.2i-o**). Therefore, parabiosis, reporter, and genetic labeling experiments consistently suggest that monocytes / blood derived cells contribute to less than 4% of the KC pool in tumor free and tumor bearing livers. We have revised accordingly the manuscript (line 135-148) and **revised Ext.data Fig.2**.

2) The reason for describing “CD47^{bright} CD9⁺ CD133⁺ phenotypic metastasis initiating cells” warrants some introduction and explanation.

AU- We have added text in the result section, line 122 to 125, to explain these data. “ flow cytometry analysis showed that tumor cells were increased ~10 fold in KC-deficient liver in comparison to control, including a subset of tumor cells which coexpress CD47^{bright} and markers previously associated with metastatic potential such as CD9 and CD133^{66,67} (**Ext.data Fig.1k,l**) and endowed with metastatic potential *in vivo* and *in vitro*⁶⁸⁻⁷⁰ (**Ext.data Fig.1m**).

3) The fate mapping data shown in Supplementary Figs 2h-o are very impressive and informative, but the various approaches are not described or explained anywhere in the text or legends and I suspect these may not be particularly familiar to readers who are not specialists in macrophage biology. If the authors consider it necessary to retain all this information, it needs explained in more detail.

AU- We agree with the reviewer. Parabiosis and the 3 genetic models that help to conclude on KC location and turnover have been explained in more detail in the manuscript and revised Ext.data Fig.2 (see above). In contrast we have removed the 2 fate mapping analysis that were not needed to support our conclusions.

4) Much of the intravital imaging is extremely elegant and convincing. However it is very difficult to see any green staining in most of the images supposed to show Cas-Green staining in Figures 2H or Supplementatry S4. One or two seem to show a few 2 small dots that are apparently associated with a tumor cell and not a Kupffer cell. It was unclear to me from the text what this approach was aiming to show – is it to illustrate Kupffer cells that have taken up dying tumor cells, or is simply to label any dead tumor cell? This needs explained and the relevant images should be improved, with appropriate annotation provided.

AU- The Caspase-3/7 green probes is a four amino acid peptide (DEVD) conjugated to a nucleic acid-binding dye that become fluorescent when bound to DNA. The Cas-Green probe was included in experiments to label dead tumor cells, and test whether cells engulfed by KC were undergoing CAS cleavage before or after uptake. The results confirm that KC engulf live non-apoptotic cells. As a positive control in intravital imaging experiments, we show Green fluorescence in a neighboring tumor cell (**Ext.data Fig 5a**). As a positive control in *in vitro* experiments, we show Green fluorescence in a tumor cell previously engulfed by a KC and subsequently extruded before

undergoing CAS cleavage (**Fig.2j. Ext.data Fig.5b**), We have added the green channel alone in the **revised Ext.data Fig.5a,b**), but the sequence is best visible in the in vitro time lapse movie **Supplementary Video S2**).

5) A related issue with the intravital staining is that I could not see any pink staining for ELK1 in Supplementary Figure S7c.

AU- We have improved the balance of color channels in **revised Ext.data Fig.8c**, to improve the visualization of ELK1 staining in the 'merge' image.

6) No data are shown for Id3fl/fl mice treated with anti-SIRPa in (**Fig.5**), which I believe would be important to provide as baseline controls for the effects in Clec4fcre; Id3fl/fl mice.

AU- We have added the results for Id3^{fl/fl} mice treated with anti-SIRPa in the **revised Fig.5c**.

7) A further issue with the intravital staining is that I could not see any pink staining for ELK1 in Supplementary Figure S7c

AU- See (5)

8) The English needs attention in places. For instance, there are often issues with the use of plural nouns/verbs, while the word "phagocyte" should be "phagocytose" when used as a verb.

AU- We made every effort to correct typos in the revised manuscript.

Referee #2. This is a well-written, (primarily mouse-based) experimental study highlighting the role of Kupffer cells (i.e. liver-resident macrophages) as a gatekeeper to limit metastatic malignancies.

1. The series of experiments is certainly elegant and convincing. An important question that remains open, to my opinion, is the translatability of the findings to human disease. Clec4f is restricted to mouse KCs, which makes it much more challenging to identify and study human KCs.

However, some important questions remain fully open:

Do human KCs also organize themselves around liver metastases as in the mouse models?

Do they share the same pattern of high ID3, low SIRPa and high tumor cell phagocytosis?

Would their presence or absence indicate prognosis / metastatic spread?

AU- To address the reviewer question we have examined human tumor samples and several scRNAseq datasets. The results are summarized in the revised manuscript in a paragraph entitled '*Conserved features of Human KC in metastatic liver*' lines 336-348, and the **revised Ext.data Fig.9**. We examined liver samples from 3 patients with PDAC and found that human KCs (CD14⁺Tim4⁺ cells) also organize themselves around liver metastases (**revised Ext.data Fig.9b**), and contain CK19 material (**revised Ext.data Fig.9c**) suggesting phagocytic activity. We also analyzed 2 independent scRNAseq datasets of human PDAC (3 patients, Nat Commun. 2023 Feb 13;14(1):79) and of human CRC (3 patients and 1 control, Mol Syst Biol. 2020 Dec;16(12):e9682) metastatic liver. Results confirmed that human KC represent the most macrophages in normal liver, but a minor subset in tumoral liver (**revised Ext.data Fig.9e-g**),

consistent with their peritumoral location, and further indicated that human KCs in tumor samples share the mouse pattern of high ID3, low SIRPα in comparison to tumor infiltrating macrophages (CD14⁺TIM4⁻), and express more CCL3 and 4 and IL18 (**revised Ext.data Fig.9h**).

Although KC are mostly located around the tumor, the prognostic value of their relative abundance within tumors nodules cannot be evaluated here.

2. Regarding the features of tumor-associated macrophages in human liver, a comparative scRNA-seq (+ spatial analysis) of human liver with metastasis (with proper control conditions) and the mouse models may be very informative to substantiate the claims on the translational value of the study. The scRNA-seq studies that are quoted in this study (ref #93/94) do not reflect this situation at all.

AU- The analysis of human tumor samples and tumor scRNAseq datasets described above, and summarized in the revised manuscript in a paragraph entitled '*Conserved features of Human KC in metastatic liver*' lines 336-348, and in the **revised Ext.data Fig.9**, indicate that the comparative features of human TAMs and KC in human metastatic liver are comparable with the results mouse models in terms of anatomical location (CD14⁺ TIM4⁻ TAMs are located within the tumors while CD14⁺ TIM4⁺ KC are located outside), and ID3, SIRPα and chemokine/cytokine gene expression, as CD14⁺ TIM4⁻ TAMs express lower ID3, higher SIRPα, and less CCL3, CCL4, and IL18 than KC.

However, our study was not designed to extensively investigate the properties of mouse and human TAMs, which appears to be highly heterogeneous (for example, Laviron et al., 2022, doi.org/10.1016/j.celrep.2022.110865, & Li et al., 2022, doi.org/10.1016/j.celrep.2022.110609).

3. Along the thought of translatability, the authors use very simple and "immunogenic" tumor models, primarily the injection of tumor cell lines. This approach is highly prone to biases, as it may over-estimate the phagocytic capacity of Kupffer cells. KCs are well known to be highly phagocytic and may be particularly active upon intraportal injection of tumor cell lines, as done in this experimental setting. An endogenous tumor model with spontaneous metastasis at a late stage (in mice) will be way more reflective of tumor biology than the models used in this study.

AU- In the revised manuscript we expanded the analysis of an endogenous tumor model with spontaneous metastasis at a late stage (6 month old KPC mice⁵⁶, *p48^{Cre} Kras^{LSL-G12D} p53^{LSL-R172H}*). Our data altogether show that Kupffer cells from 6-month-old KPC mice organize themselves around liver metastases (**Revised Fig2a**), and engulf CK19⁺ material (**revised Ext.data Fig.4c**) as much as in models that rely on the injection of tumor cell lines (see **revised Fig.2F, Ext.data Fig.4b**). **In addition**, immunofluorescence analysis indicates that **KC from 6-month-old KPC mice** also produce the chemokines CCL3,4,5 and cytokines IL12, 15, 18 (**revised Ext.data Fig.5c**), which recruits and activate lymphoid effector cells (see below).

4. The mechanisms of "anti-tumor activity" of KCs were not fully clear to me. On the one hand, the phagocytosis of tumor cells may be involved (which is probably less relevant in the cases of established tumors?); on the other hand, the expression of lymphocyte-attracting chemokines could promote the NK- and CD8 T cell responses. What is the contribution of each mechanism? How does the chemokine expression explain the anti-tumor activity of the NK and CD8 T cells? What I mean here: is the attraction enough or is there a KC-dependent mechanism of activation (e.g. antigen presentation).

AU- The analysis of the mechanism of KC antitumor activity was underdeveloped in our original manuscript. In response to the reviewer request, we expanded our analysis of the role of KC in lymphoid cell activation. As noted by reviewer 3, KCs are a major source of cytokines such as IL-

12, IL15 and IL18 which activate effector T cells and NK cells, and can promote antitumor T /NK cell-mediated immunity. We show that wt KC exposed to tumor cells *in vivo* increased their expression of IL12, IL15, and IL18, in addition to chemokines (**revised Fig.2e**). We confirmed these results at the protein and anatomical level by immunofluorescence, which indicated that increased expression of IL12, IL15, and IL18 is preferentially a property of KC located at the tumor boundaries (**revised Ext.data Fig 5d**).

This upregulation of cytokines gene expression (as well as chemokines) was reduced in Id3-deficient Kupffer cells (*Clec4fCre;Id3^{fl/fl}* mice) (**revised Fig.4b,c**), also confirmed by immunofluorescence at the protein level, by the loss of cytokine and chemokine expression by KC in tumor bearing *Clec4fCre;Id3^{fl/fl}* mice (**revised Fig.4h,i, revised Ext.data Fig. 6j,k**). Loss of chemokine and cytokine expression was associated with reduced recruitment of NK cells and CD8+ T cells to the liver and tumor boundaries (**revised Fig.4j,k**), and also reduced production of INF γ and TNF by NK cells and CD8+ T cells by flow cytometry in tumor bearing *Clec4fCre;Id3^{fl/fl}* mice (**revised Fig.4l**).

In rescue experiments, treatment of *Clec4fCre;Id3^{fl/fl}* mice with anti-Sirpa antibodies was sufficient to rescue expression of IL12, IL15, and IL18 by KC (**revised Fig. 5b**), the recruitment of NK and CD8+ T-cells to the tumor (**revised Fig.5g-i**), and their production of INF γ and TNF (**revised Fig.5j**). These data suggest that in addition to chemokine expression, production of IL-12, IL15 and IL18 is a KC-dependent mechanism of activation that can explain the anti-tumor activity of the NK and CD8 T cells.

Analysis of the mouse KPC endogenous tumor model at 6 months confirmed production of chemokines and cytokines by KC (**revised Ext.data Fig.5c**). The production of chemokines (CCL3/4) and cytokines (IL18) by human KC was also detectable in the scRNAseq datasets from PDAC and CRC metastatic liver (**revised Ext.data Fig.9h**).

Regarding the respective contribution of phagocytosis and recruitment/activation of lymphoid effector cells, we show they both contribute to KC anti-tumor activity, as depletion of CD8T cells and NK cells increase tumor growth in control mice to a level intermediate between untreated wt and *Clec4f^{Cre}Id3^{fl/fl}* mice, but did not increase tumor growth in *Clec4f^{Cre}Id3^{fl/fl}* mice (**Revised Fig.4m**). This suggest that KC-mediated phagocytosis and KC-mediated promotion of lymphoid cell responses are both important and at least additive.

5. The authors claim that BMDMs or human iPSC-derived macrophages can be reprogrammed towards KCs by ID3 and would then exert anti-tumor activity. I found this part particularly weak, since the experimental system appears quite artificial and involves high numbers of adoptively transferred, genetically modified cells.

AU- The claim on our part that BMDMs or human iPSC-derived macrophages can be 'reprogrammed' towards KCs by ID3 would be an overstatement. We propose instead that ectopic expression of Id3 is sufficient to endow BMDMs or human iPSC-derived macrophages with the anti-tumor activity of wt KC. Other features of KC, e.g. expression of TIM4 or Clec4f, are not driven by ID3 expression. In the revised manuscript, we have attempted to improve the wording to avoid confusion.

We have also improved our analysis of the role of ID3 gain-of-function in macrophages in mechanistic experiments, and in additional wt mice models (**revised Fig.6**). We reported in our original manuscript that that intraportal injection of ID3-BMDM 7 days after intraportal injection of KPC cells in *Clec4f^{Cre}; Id3^{fl/fl}* mice limited the growth of liver tumors after 2 weeks (**revised Fig. 6i**). In our revised manuscript we also show that one intraportal injection of ID3-expressing BMDM

prevented the growth of liver tumors in wild-type mice after 2 weeks in the more aggressive LLC1 cells model (**Revised Fig. 6j**). Moreover, a survival analysis in this model indicated that the injection of ID3-expressing BMDM improved survival of wild-type mice. In the absence of BMDM treatment or with lenti-control BMDM, wt mice required sacrifice within 3 weeks, while ~half the mice were alive at 5 weeks after treatment with ID3-expressing BMDM (**Revised Fig. 6j**).

In the melanoma model, where 10^6 B16F10 cells are injected subcutaneously in the 2 flanks of a wild-type mice, followed after a week by one intratumoral injection of 5×10^5 Ctr BMDM in one flank and 5×10^5 ID3-BMDM in the controlateral flank, ID3-BMDM blocked tumor growth (**Fig. 6k**), and triggered accumulation of activated CD8⁺ T cells and NK cells producing IFN γ and TNF to the B16F10 tumors in the corresponding flank (**Fig. 6k, Ext.data Fig.10b,c**).

These results show that, in addition to be required for the anti-tumor activity of KC, expression of *Id3* is also sufficient to endow mouse and human macrophages with a potent anti-tumor activity, *in vitro* and *in vivo*, in mutant as well as in wild-type mice, against epithelial and melanocytic cancers, including increased phagocytic activity against tumor cells, and the ability to recruit and activate lymphoid effector cells at the tumor site.

6. Many tissues have “resident macrophages” – I would find it very interesting and relevant, whether similar mechanisms exist, for instance, in the brain or in the lung to limit metastasis. Simply looking at single factors like SIRPa may not give the correct answer here, since KCs are quite particular in this sense (e.g. Clec4F, ID3 etc. very specific to KCs).

AU- We fully agree with the reviewer. We identified *Id3* as one of the ‘lineage determining factors’ (LDFs) that control KC identity (Mass et al., 2016). It is likely that other LDFs may endow KCs and other tissue macrophages with the same or other properties. Although, we believe that the investigations suggested by the reviewer on the mechanisms that may control phagocytosis and anti-tumor responses by other resident macrophages are needed and may yield important results, we think that they are beyond the scope of our present study.

Referee #3: Focusing on Kupffer cells (KCs), Deng et al. demonstrate the role of ID3 in the anti-tumor functions of these macrophages in mice. Using multiple elaborate experimental tools, the authors showed that ID3, whose expression is mainly restricted to KCs, prevents Sirp-a expression by interfering with the DNA binding of two activating transcription factors (E2A and ELK1). Interestingly, ectopic expression of ID3 in monocyte-macrophages prevents Sirp-a expression and enhances the anti-tumor functions of the cells, opening up potential therapeutic implications. Overall, the authors are to be commended for the quality of their study, which should be of interest to the cancer research community. However, there are aspects of this work that need to be completed to support and clarify the findings presented.

Experimental Design:

- All experiments were performed using mice with loxP sites flanking the genes of interest as controls. What are the effects of recombinase in these mice? Additional data are needed to illustrate the effects (if any) mediated by the recombinase itself.

AU- *Clec4f^{Cre}* mice were previously published and are born in normal mendelian ratio. They do not present a discernable phenotype in comparison to control. Specifically in the absence of DT

injection, we did not observe differences in tumor growth between *Clec4f^{Cre}R26^{LSL-DTR}* mice and *R26^{LSL-DTR}* mice (**revised Ext.data Fig.1j**)

- Overall survival experiments are needed to define whether ID3 expression by KCs has a major inhibitory effect on tumor progression.

AU- We performed overall survival experiments in a model of liver metastasis in wt, KC-deficient mice, and mice with ID3-deficient KCs after intraportal injection of KPC cells. In this model ~half of wt mice are still alive after 5 weeks. Survival analysis indicated that the survival of KC-deficient mice (*Clec4f^{Cre}R26^{LSL-DTR}*, **revised Fig.1g**) and of *Clec4f^{Cre}Id3^{fl/fl}* (**revised Fig. 3j**) was reduced, as all mutant mice had to be sacrificed at 3 weeks (p=0.005, and 0.002 respectively, Log-rank (Mantel-Cox) test).

In addition, we also performed survival experiments in a more aggressive model of liver metastasis in wt mice, after intraportal injection of Lewis Lung Carcinoma (LLC, 1x10⁶ cells, **revised Fig.6j**). Mice received intraportal injection of PBS, or 1x10⁶ BMDM transduced with a control lentivirus, or with a lentivirus coding for ID3 (lenti-Id3) one week after tumor injection. Survival analysis indicated that all control mice had to be sacrificed at 3 weeks, while ~half of mice that had received lenti-Id3 BMDM were still alive after 5 weeks. These data suggest that ID3 expression by KCs, or by BMDM transduced with a lentivirus coding for ID3, has a strong protective effect on tumor progression.

ID3 and phagocytosis:

- The authors set up an elegant in vitro assay to study the uptake of tumor cells by KCs. Although phagocytosis of apoptotic cancer cells is excluded, it will be useful to further visualize and monitor the phagocytosis action mediated by these cells. The use of inhibitors, the visualization of actin cytoskeleton rearrangements, or the assessment of fluorescence of tumor debris in phagosomes will provide important information about the mechanism of action being studied.

AU- We assessed the intracellular localization of tdTomato in the LAMP1⁺ compartment by immunofluorescence in KC two weeks after intraportal injection of KPC-1-tdT cells. Results showed tumor material (tdTomato) was colocalized with Lamp1⁺ phagolysosomes (**revised Fig.2g**). In addition, addition of Latrunculin A, an inhibitor of actin polymerization, abolished engulfment of tumor cells by Kupffer cells *in vitro* (**revised Fig.2j**), supporting the active engulfment of cancer cells by Kupffer cells. These results support the hypothesis of phagocytosis of live tumor cells by KCs.

- The data suggest differences between the KCs of non-tumor-bearing mice and those of tumor-bearing mice. But what are the functional differences between these KCs? How does the presence of a tumor alter their phenotypes and functions?

AU- In response to the reviewer question we have expanded our analysis of the functional differences between KCs non-tumor-bearing mice and those of tumor-bearing mice.

Our results indicate that KC are activated by tumor cells. Functionally, wt KC located around tumor cells actively uptake them (**revised Fig. 2f-g**), and produce chemokines (notably CCL3, CCL4, CCL5) and cytokines (IL12, IL15, IL18, **see below**) (**revised Fig. 2k,l, Ext.data Fig 5d**) that recruit and activate effector lymphoid cells (NK cells and CD8⁺ T-cells) (**revised Fig. 2m,n**) which in turn produce IFN γ and TNF (**revised Fig. 4l**). We show in the manuscript that these tumor-driven functional differences are ID3-dependent.

Phenotypically, tumor cells regulate expression of activating and inhibitory receptors on KC, including ID3-dependent receptors, of which at least two of them, the activating receptor DECTIN1 and the inhibitory receptor SIRPA, regulate KC effector functions (phagocytosis and effector

lymphoid cells recruitment/activation). In the revised manuscript we added a list of activating and inhibitory receptors expressed by KC and which one are regulated by tumor cells and/or ID3 (**revised Ext.data Fig.3a**).

- ID3 appears to simultaneously stimulate upregulation of Dectin-1 and downregulation of Sirp-a. Does Sirp-a down-regulation alone promote phagocytosis? Are other molecules (phagocytic receptors) involved and what are their roles? At least a more thorough study of Dectin-1 is needed.

AU- As indicated by the reviewer, loss-of-function experiments in KC, and gain-of-function experiments in mouse BMDM and hiPSC-derived macrophages indicate that ID3 stimulate upregulation of Dectin-1 and downregulation of SIRPA (**revised Fig.4b-d, revised Fig.6g**).

Mechanistically the down-regulation of the activating receptor Dectin-1 appears to be due to up-regulation of the inhibitory receptor SIRPA, because SIRPA blockade results in the rescue of Dectin-1 expression by ID3-deficient KC (**revised Fig.5b, Ext.data Fig.7a**).

However, we also show in the revised manuscript that Dectin-1 blockade abrogates the phagocytosis of tumor cells by wt KC (**revised Fig.5e**), and decrease cytokine and chemokine expression (**revised Fig.5f**), suggesting that low SIRPA expression in itself is not sufficient to promote phagocytosis or cytokine/chemokine expression.

As indicated above, we have listed in **revised Ext.data Fig.3a** the activating and inhibitory receptors expressed by KC indicating which one are regulated by tumor cells and/or ID3. Dectin1 is the only activating receptor we found to be upregulated in response to tumor cells in a Id3-dependent manner. Other activating receptors, including Dectin2, Dectin3, Mincle and Clec5a are also upregulated in response to tumor cells, but are not regulated by ID3. In contrast Clec2 and Clec4g are not regulated by tumor cells in wt mice, but down regulated in the absence of ID3, and could therefore also contribute to KC activation. Among inhibitory receptor, SIRPA is not regulated by tumor in wt KC, but is up-regulated in the absence of ID3. Other inhibitory receptors, such as Clec4a1 and Siglece are down regulated by tumor cells and (modestly) up-regulated in the absence of ID3, and could therefore also contribute to the KC phenotype.

Therefore, we hypothesize that ID3 controls KC activation and anti-tumor activity at least in part by regulating SIRPA and Dectin1 expression.

ID3 and effector lymphoid response:

- Since KCs can be a major source of IL-12 in the liver (Seki et al, The Journal of Immunology, 2001, Hou et al, Cellular & Molecular Immunology, 2016, Yong et al, Scientific reports, 2017, Siwicki et al, Science Immunology, 2021), and IL-12 can promote antitumor T cell-mediated immunity, it is possible that this signaling pathway plays an important role in the antitumor functions of KCs described in this study. In this context, what is the involvement of ID3 in the secretion of IL-12 by KCs, and the stimulation of an effector lymphoid response?

AU- We thank the reviewer for their insightful comments, as the study of the mechanism of antitumor activity was underdeveloped in our original manuscript.

We found that wt KC exposed to tumor cells *in vivo* increased their expression of Il12, IL15, and IL18, in addition to chemokines (**revised Fig.2e**). We confirmed these results at the protein and anatomical level by immunofluorescence, which indicated that increased expression of Il12, IL15, and IL18 is preferentially a property of KC located at the tumor boundaries (**revised Ext.data Fig. 5d**).

Upregulation of cytokines gene expression (as well as chemokines) was reduced in Id3-deficient Kupffer cells (*Clec4fCre;Id3^{ff}* mice) (**revised Fig.4b,c**), which was also confirmed by

immunofluorescence at the protein level, by the loss of cytokine and chemokine expression by KC in tumor bearing *Clec4fCre;Id3^{fl/fl}* mice (**revised Fig.4h,i, Ext.data Fig.6j,k**). Loss of chemokine and cytokine expression was associated with reduced recruitment of NK cells and CD8+ T cells to the liver and tumor boundaries (**revised Fig.4j,k**), and also reduced production of INF γ and TNF by NK cells and CD8+ T cells by flow cytometry in tumor bearing *Clec4fCre;Id3^{fl/fl}* mice (**revised Fig.4l**).

In rescue experiments, treatment of *Clec4fCre;Id3^{fl/fl}* mice with anti-Sirpa antibodies was sufficient to rescue expression of IL12, IL15, and IL18 by KC (**revised Fig. 5b**), the recruitment of NK and CD8+ T-cells to the tumor (**revised Fig.5g-i**), and their production of INF γ and TNF (**revised Fig.5j**). Analysis the mouse KPC endogenous tumor model at 6 months confirmed production of chemokines and cytokines by KC (**revised Ext.data Fig.5c**).

These data altogether suggest that ID3 controls the production of IL-12, IL15 and IL18 which contribute to activation of NK and CD8 T cells by KC.

- Intratumoral injection of BMDMs expressing ID3 reduced local tumor growth and was associated with a higher frequency of intratumoral CD8 T cells, compared to contralateral tumors. But it is not clear from the data presented whether this injection of BMDMs could trigger a systemic effect (and delay the growth of the contralateral tumor). Specifically, what are the systemic effects induced by this adoptive transfer compared to untreated mice? As ID3 expression by macrophages limits tumor growth and metastasis, it would be important to clarify this point to support a potential therapeutic application of the presented approach.

AU- We expanded our analysis of the effects of intratumoral injection of BMDMs expressing ID3. Analysis of mice that received BMDM expressing lenti-ID3 in one flank and BMDM expressing lenti-control in the other indicated that the recruitment of T cells and NK cells, and their expression of INF γ and TNF was higher in the flank injected with BMDM lenti-ID3 (**revised Fig.6k**)

In response to the reviewer query, 2 cohort of C57BL/6j mice which had received subcutaneous injection of 1×10^6 B16F10-*luci-tdT* cells into left and right flank, were treated one week later either by intra-tumor injection of lenti-control BMDM in one flank and PBS in the other, or by lenti-*m/d3* BMDM expressing cells in one flank and PBS in the other (**revised Ext.data Fig.10c**). Photoradiance analysis indicated that intra-tumor injection 5×10^5 BMDM expressing lenti-*m/d3* cells in one flank did not affect the growth of PBS-treated tumors in the other flank, in comparison to mice treated with lenti-control BMDM in one flank and PBS in the other, suggesting the absence of a strong systemic effect in this model (**revised Ext.data Fig.10c**).

Nevertheless, in a lethal model of liver metastasis after intraportal injection of Lewis Lung Carcinoma (LLC, 1×10^6 cells), intraportal injection of BMDM lenti-Id3 after one week, all control mice had to be sacrificed at 3 weeks, while ~half of mice that had received lenti-Id3 BMDM were still alive after 5 weeks (**revised Fig.6j**). Altogether, these results suggest that intra-tumoral injection of BMDM lenti-Id3 in subcutaneous tumors has local effects, although we cannot exclude a systemic effect of intravenous injection.

Minor comments:

- Fig 1.g: Could the authors show the effectiveness of depletion at different time points?

AU- The depletion of KC after DT injection in *Clec4f^{Cre}R26^{LSL-DTR}* mice is shown in **Ext.data Fig.1i**

- Fig6a: There appears to be a color reversal between red and blue. E2A inhibits macrophage activation and ID3 activates it. Could the authors please check?

AU- we corrected the schematic in Fig.6a.

- The text contains a fair number of typographical errors. Could the authors please check and correct all these errors before resubmitting the manuscript?

AU- We made every effort to remove typos from the revised manuscript.

Reviewer Reports on the First Revision:

Referees' comments:

Referee #1 (Remarks to the Author):

I thank the authors for their responses to my original comments and for the addition of informative new data which significantly improve the manuscript. However some issues remain:

1) Many of the data shown in the Extended Data figures remain poorly described and/or annotated, not explaining what is being shown and why. These omissions make an already dense paper hard work and risk the study not making the impact it deserves on most readers. Amongst many such instances:

- it is not clear what the data shown in Ext Data Fig1 e, f are showing
- Ext. Data Fig.1m is not explained
- there is still no explanation of why the Cxc3cr1- and Cxcr4- driven models shown in Ext. Data Fig.2 are useful
- the annotating arrows and other features in Movie 1 are not explained
- the Tnfrsf11aCre-driven model used in Ext. Data Fig.6 is not explained
- the authors' reply to my comments about the use of the Cas-Green probe was very helpful, but this information is still not in the manuscript and this would be essential for a clearer understanding of the role of KC phagocytosis in tumor protection.

2) What does "cells identified as macrophages" mean when used in the annotation of Data Fig.9?

3) It remains very difficult to see green staining for the Cas-green expression in figures or movies, or for the CK19 expression in Ext. Data Fig.9c, which in fact looks identical to the TIM4 pattern shown in the same figure.

4) It is difficult to see much improvement in the English, where most of the original grammatical errors remain. As with the lack of information describing data, these issues make the paper difficult to follow in places.

Referee #2 (Remarks to the Author):

The authors have provided additional data, which certainly improve the manuscript a lot. I particularly liked the additional, more "physiological" tumor metastasis model. This provides confidence that the mechanism is not related to model-specific artifacts.

1. The translational data from human liver metastasis remain weak and superficial, but may be consistent with the findings presented in the manuscript. Whether ectopic Id3 expression would be a useful strategy in human metastasis (or how it would synergize with, e.g., anti-PD1 treatment) remains open.

2. The manuscript still has some shortcomings on the anti-tumor mechanisms by Kupffer cells (interactions with T cells? phagocytosis of tumor cells? antigen presentation? cytokines?).

3. I also understand that the authors do not want to look beyond the liver (my comment #6), but I still wonder whether the mechanisms are specific to liver metastasis (Kupffer cells) or not.

Referee #3 (Remarks to the Author):

The authors have elegantly addressed all the points raised, resulting in an improved version of the original manuscript; they are to be commended for their efforts.

Only a few minor points are discussed below:

In the revised manuscript, the authors have shown the lack of systemic tumor control by intratumoral injection of ID3 BMDM, while solidly demonstrating their ability to control the tumor locally. This dichotomy in no way diminishes the importance of the mechanism described, but needs to be clearly stated in the manuscript so that the scientific community can take it into account for future studies.

Contrary to the conventional view of the anti-inflammatory role of resident macrophages, this manuscript highlights the immunostimulatory capacity of KC, particularly through the secretion of IL12, IL15 and IL18, which promotes effector lymphoid cell responses and anti-tumor immunity. Previous work has shown the importance of IFN γ sensing by KC for IL12 secretion (DOI: 10.1126/sciimmunol.abi7083). Discussion of these results would add to the narrative while strengthening the data presented.

The representative images of in vitro uptake of KPC-1-tdTomato by Kupffer cells shown in Extended Data Figure 5 are redundant with the representative images shown in Figure 2j. Could the authors show a different field of view and representative time-lapse images from the latrunculin A condition? This would strengthen the results.

The legends of the processed figures should be reviewed and added if missing.

Referee #4 (Remarks to the Author):

per request, I am going to focus on only the statistical aspects on the comparisons. Overall, all comparisons in the paper were clearly described. couple of comments:

- 1) In the Results section, it is not clear how the controls were chosen and whether they were comparable to the treated.
- 2) Fig 2d, it is not clear what adj p-values were referred to. In Fig 2e, it is apparent that some of the p-values are borderline and would not be "significant" after adjusting for multiple comparisons (16 of them). For example, it's stretchy to say il18 was confirmed.
- 3) all comparisons were performed by either t-test or ANOVA. Both rely on the assumption of data being normally distributed. most of the data look at least symmetric, however, there are few data types that look quite skewed, including those in Fig 1a-d.
- 4) It is not clear in Methods what deep learning methodology was used and what software. The provided code was helpful, but not every reader would dig into the code to understand. So suggest adding more description in the extended data or suppl materials.

Author Rebuttals to First Revision:

Nature manuscript 2023-01-00391A. Point by point responses to Referees' comments:

Referee #1

I thank the authors for their responses to my original comments and for the addition of informative new data which significantly improve the manuscript. However some issues remain:

1) Many of the data shown in the Extended Data figures remain poorly described and/or annotated, not explaining what is being shown and why. These omissions make an already dense paper hard work and risk the study not making the impact it deserves on most readers. Amongst many such instances:

- it is not clear what the data shown in Ext Data Fig1 e, f are showing.
- Ext. Data Fig.1m is not explained
- there is still no explanation of why the *Cxc3cr1*- and *Cxcr4*- driven models shown in Ext. Data Fig.2 are useful
- the annotating arrows and other features in Movie 1 are not explained
- the *Tnfrsf11a*Cre-driven model used in Ext. Data Fig.6 is not explained
- the authors' reply to my comments about the use of the Cas-Green probe was very helpful, but this information is still not in the manuscript and this would be essential for a clearer understanding of the role of KC phagocytosis in tumor protection.

Author response:

We have rewritten for clarity the legends of Figures, Extended Data figures and movies.

Legend for Ext Data Fig1e: Shows the flow cytometry and cytopsin giemsa stain analysis of Kupffer cells (pop1: $F4/80^+Tim4^+$), and other myeloid cells (pop2 : $F4/80^+Tim4^+MHCII^+$ and pop3 : $F4/80^+Tim4^+MHCII^-$) from the liver of *C57BL/6J* mice 2 weeks after intra-portal injection of 1×10^6 KPC-1-*luci-tdT* cells. The bar plot represents the % of cells from each population that are labeled by tdT in the liver of *Clec4f^{Cre-tdT}* mice and by YFP in the liver of *Flt3^{Cre}R26^{LSL-YFP}* 2 weeks after intra-portal injection of 1×10^6 KPC-1 cells or in the absence of tumor injection. n= 3-6 mice from 2 independent experiments.

The results indicate that KC are selectively labeled by the *Clec4f* reporter, while the other liver myeloid cells are selectively labelled with the *Flt3^{Cre}* reporter, irrespectively of the presence of a tumors. Together with the data below (*Cxc3cr1*- and *Cxcr4*- driven models), these results are useful to show that KC are resident cells that do not renew from the bone marrow in the presence or absence of liver tumors.

Legend for Ext Data Fig1m: shows the analysis of metastatic potential of $CD47^{bright}CD9^+CD133^+$ tumor cells and $CD47^{low}CD9^{low}CD133^{low}$ tumor cells *in vivo* by bioluminescent analysis, two weeks after portal vein injection of 5×10^4 cells or 2×10^5 KPC-1-*luci-td* cells (Left, circles represent individual mice, n=4-8 mice per group from 2 independent experiments), and *in vitro* clonogenic potential in oncosphere culture (right, circles represent individual oncospheres, see Methods). The results support the hypothesis that the $CD47^{bright}CD9^+CD133^+$ tumor cells fraction has metastasis-initiating potential.

Cx3cr1- and Cxcr4- driven models: in the result section lines 146-150 of the revised manuscript we wrote: "In addition, genetic labeling of bone-marrow derived cells from tumor free and tumor bearing mice using 3 genetic models (*Cx3cr1^{gfp}* mice, *Cxcr4^{gfp}* mice, and *Cxcr4^{CreERT2}R26^{LSL-tdT}* mice pulsed with OH-TAM at 6 weeks of age, see Methods), confirmed that most TIM4+ cells (KCs) from both $CD206^+$ and $CD206^{bright}$ subsets are not labeled (**Ext. Data Fig.2k-o**)." In the

Revised Legend for Ext Data Fig2k-o, we wrote: **k-** Bar-plots show the percentage of Tim4⁺, Tim4⁺CD206⁺, Tim4⁺CD206^{hi} KC, and TIM4⁻ TAMs labeled with GFP in *Cxcr4^{gfp/+}* mice and in *Cx3cr1^{gfp/+}* mice, and labeled with tdT in *Cxcr4^{CreERT2};R26^{LSL-tdTomato}* mice pulsed with 4OH-TAM at 6 week-old, that have received intra-portal injection of 1×10⁶ KPC-1-*luci-tdT* cells 2 weeks before analysis or were left untreated. n=3-4 mice per group. **l-** Table represents the percentage (mean and sd) of Tim4⁺ KC, Tim4⁺ CD206⁺ KC, Tim4⁺CD206^{hi} KC, and TIM4⁻ TAMs in the liver of tumor free and tumor bearing *Cxcr4^{gfp/+}* mice, *Cx3cr1^{gfp/+}* mice, and *Cxcr4^{CreERT2};R26^{LSL-tdTomato}* mice pulsed at 6 weeks with 4OH-TAM. **m-** Immunofluorescence staining for F4/80, Tim4, GFP and tdTomato on frozen liver section from *Cx3cr1^{gfp/+}* tumor bearing mice in **(k)**. **n-** immunofluorescence staining for F4/80, Tim4 and tdTomato on frozen liver section from *Cxcr4^{CreERT2} R26^{LSL-tdTomato}* tumor bearing mice in **(k)**. **o-** *Cxcr4^{CreERT2} R26^{LSL-tdTomato}* mice are pulsed with 4OH-TAM or PBS at 6 weeks and analyzed 2 weeks later. Bar graphs represent the % of tdT⁺ cells determined by flow cytometry among the indicated cells types. n=2-3 mice per group. Statistics: One-way ANOVA **(h,k)**. Dots represent individual mice. mean ± sd. ns, not significant.

Supplementary Video S1. Engulfment of tumor cells by Kupffer cells *in vivo*.

The legend of movie 1 has been corrected as follows:

“Liver intravital imaging of *in vivo* uptake of KPC-1-mtdT cells by Kupffer cells in liver from C57BL/6j mice 2 weeks after intra-portal injection of 1×10⁶ KPC-1-memtdT cells in the presence of iv. injection of Tim4-AF647 antibodies and CellEvent™ Caspase-3/7 green reagent. See methods, scale and time scales are embedded in the movie file. This video show examples of three KCs engulfing Cas-green-negative tumor cells *in vivo*. Blue arrow (left) follows a KC from 0h0min to 2h13min16s. Blue arrow (right) follows a KC from 1h8min40s to 2h3min1s. Cell debris are extruded from KC from 1h45min07s to 1h49min14s. Orange arrow follows a third KC from 3h18min46s to 3h54min15s.”

The results from the Tnfrsf11aCre-driven model reproduced findings from the ID3-deficient model, and was therefore not essential, and we removed these data for the benefit of simplicity.

Cas green probe: We modified the result section (when the Cas green probe is introduced, line 177-178) as follows: “Intravital microscopy *in vivo* in the liver, using a caspase3/7 cleavage reporter (Cas-Green) to monitor tumor cell apoptosis and death (Lakhani, S. A, Flavell R et al., Science 2006 doi:10.1126/science.1115035), documented the engulfment of live KPC tumor cells by KC.

We also modified the Methods section (**Intravital imaging of liver Kupffer cells and KPC-1-MemtdT tumor cells *in vivo***) as follows: “...Mice were then injected retro-orbitally with 5μL CellEvent™ Caspase-3/7 Green reagent (Invitrogen), a four amino acid peptide (DEVD) conjugated to a nucleic acid binding dye that become fluorescent when bound to DNA...”

2) What does "cells identified as macrophages" mean when used in the annotation of Data Fig.9?

Author response:

In Extended Data Fig.9e we omitted to describe "cells identified as macrophages". They correspond to cells expressing CD14 and at least one of the other macrophage markers TIM4 or TREM2.

In the revised Extended Data Fig.9 g, we have relabeled the X axis (% Tim4⁺ among CD14+ macs), and we have indicated in the Legends that MAC are defined broadly as cells expressing CD14 and at least one of the other macrophage markers TIM4 or TREM2.

3) It remains very difficult to see green staining for the Cas-green expression in figures or movies, or for the CK19 expression in Ext. Data Fig.9c, which in fact looks identical to the TIM4 pattern shown in the same figure.

Author response:

Figure 2g, and Ext. Data Fig.S4a and b were corrected and Cas-green expression, which becomes detectable at late stage of the phagocytic process, is now clearly visible (revised Fig.2G is shown here).

CK19 expression in Ext. Data Fig.9c: We apologize for the confusion, we have replaced this panel with the correct one, in the revised Ext. Data Fig.9d. and the following legend: “d- Expression of CK19 by Tim4+ KCs in metastatic liver samples from PDAC patients (n=3), each dot represents a KC (left). The bar-plot (center) indicate the % of KC stained for CK19. A representative high power immunofluorescence micrograph of CK19, Tim4, and CD14 expression is also shown (right, bar= 20µM).

Revised Extended data Fig.9d

4) It is difficult to see much improvement in the English, where most of the original grammatical errors remain. As with the lack of information describing data, these issues make the paper difficult to follow in places.

Author response: We attempted to remove grammatical incorrections to the best of our abilities.

Referee #2

The authors have provided additional data, which certainly improve the manuscript a lot. I particularly liked the additional, more "physiological" tumor metastasis model. This provides confidence that the mechanism is not related to model-specific artifacts.

1. The translational data from human liver metastasis remain weak and superficial, but may be consistent with the findings presented in the manuscript. Whether ectopic Id3 expression would be a useful strategy in human metastasis (or how it would synergize with, e.g., anti-PD1 treatment) remains open.

Author response: We appreciate the reviewer's interest in our study and their opinion on the strengths and weaknesses of our revised manuscript. We have performed additional experiments to strengthen the human/ translational interest of the manuscript.

- in-situ analysis of human chemokines and cytokines production by KC in PDAC metastatic liver samples (revised **Extended data Fig.9e**, shown below). These experiments showed that peritumoral KC express CCL3/4/5 and IL 12/15/18 in PDAC metastatic liver samples.

Revised Extended Data Fig.9e

- We also found in the macrophage/tumor cell co-culture model that ectopic expression of ID3 in human iPSC-Macs, in addition to downregulate SIRPA and to upregulate Dectin1 (Fig.6b), also increases the production of CCL3, CCL4, CCL5, IL12, IL15 and IL18 by macs in response to tumor cells in vitro (**Revised Fig.6c**).

In addition, we show that supernatant from the ID3 expressing human macrophages incubated with tumor cells was sufficient to trigger proliferation of human CD8 T cells (**Revised Fig.6d**, shown below) and their production of IFN γ (**Revised Fig.6e**) and TNF (Extended data Fig.10b), and increased the production of IFN γ by human NK cells (**Revised Fig.6e**).

Revised Fig.6c

Revised Fig.6 d,e

These additional data support the idea that ectopic Id3 expression may be a useful strategy in human metastasis, however we feel that investigations on how this would synergize with other treatments, e.g., anti-PD1 treatment is beyond the scope of our manuscript.

2. The manuscript still has some shortcomings on the anti-tumor mechanisms by Kupffer cells (interactions with T cells? phagocytosis of tumor cells? antigen presentation? cytokines?).

Author response: As shown above, the revised manuscript shows that, in addition to endow human iPSC-Macs with the ability to phagocytose tumor cells (revised Fig. 6b), ectopic expression of ID3 in human iPSC-Macs also increased their production of the chemokines CCL3, CCL4, CCL5, and the cytokines IL12, IL15 and IL18 in response to tumor cells *in vitro* (Fig. 6c). Furthermore, supernatants from cocultures of lenti-ID3 iPSC-Macs and tumor cells were sufficient to trigger the proliferation of CD8 T cells (Fig. 6d), the production of IFN γ by CD8 T cells and NK cells (Fig. 6e), and the production of TNF by CD8 T cells (Ext. Data Fig.10b) while the supernatant of lenti-control iPSC-Macs/tumor cell cocultures had little or no effect (Fig. 6d,e, Ext. Data Fig.10b).

We also confirmed in mice expression of Id3 by KC was necessary to trigger the production of CCL3, CCL4, IL15 and IL18 by KC incubated with tumor cells *in vitro*, and increased their production of CCL5 and IL12 in response to tumor cells *in vitro* (Extended Data Fig 6i). Furthermore, we found that the supernatant from the wt KC/tumor cell coculture increased the production of IFN γ by NK cells *in vitro*, in comparison to Id3-deficient KCs or wt KC alone (Extended Data Fig 6j).

Revised Extended Data Fig.6 i,j

We have revised the discussion section to better address the anti-tumor mechanisms orchestrated by Kupffer cells, including the interactions with T cells, phagocytosis of tumor cells, antigen presentation, and cytokines, and added a schematic (Ext. Data Fig.10e):

Revised Extended Data Fig.10

Revised discussion, lines 409-415: ID3 controls the activatory/inhibitory receptor balance which in turn controls Kupffer cell activation by tumor cells. Activated Kupffer cells phagocytose live tumor cells, recruit NK cell and CD8 T lymphoid effector cells to the tumor and the peri-tumoral niche via the production of chemokines, and activate these lymphoid effector cells to proliferate and produce IFN γ and TNF, at least in part via the production of the cytokines IL12, 15 and 18. ID3 allows

activation of Kupffer cells at least in part by repressing transactivation of the inhibitory receptor SIRPA by bHLH and the MAP-Kinase pathway (**Ext. Data Fig.10e**).

Lines 423-434: Our results indicate that activation of the lymphoid effector response by ID3-expressing macrophages is at least in part mediated by supernatant from macrophage/tumor cell co-culture, thus represents a non-cognate, or innate, activation of lymphoid cells. Nevertheless, because ID3 expression is critical for the engulfment of tumor cells by Kupffer cells, one would expect a defect in cross presentation of tumor antigens by ID3 deficient macrophages, and we cannot exclude that expression of ID3 in macrophages may regulate the presentation of cognate antigen by KC to T cells. It is noteworthy however that KC have been consistently shown to be poor cognate antigen presenting cells^{1,2}. Although the anti-tumor activity of Kupffer cells in a liver metastasis model results in smaller lung and (less consistently) spleen metastasis, this could be attributed to a debulking effect as our results also suggest that the anti-tumor response mediated by ID3-expressing macrophages is local, since ID3-expressing macrophages injected in a sub-cutaneous tumor do not prevent the growth of a contra-lateral tumor.

3. I also understand that the authors do not want to look beyond the liver (my comment #6), but I still wonder whether the mechanisms are specific to liver metastasis (Kupffer cells) or not.

Author response: The role of Id3 may be limited to mouse and human KC as they express high levels of ID3 in contrast to other macrophages (Figure 3b, and ³). However, we demonstrate that this anti-tumor activity is transferable to other mouse and human macrophages, which otherwise display low phagocytic activity against tumor cells such as mouse BMDM and human iPSC-derived macrophages, via enforced expression of ID3.

Referee #3

The authors have elegantly addressed all the points raised, resulting in an improved version of the original manuscript; they are to be commended for their efforts.

Only a few minor points are discussed below:

In the revised manuscript, the authors have shown the lack of systemic tumor control by intratumoral injection of ID3 BMDM, while solidly demonstrating their ability to control the tumor locally. This dichotomy in no way diminishes the importance of the mechanism described, but needs to be clearly stated in the manuscript so that the scientific community can take it into account for future studies.

Author response: We agree with the reviewer, and the dichotomy is stated in the revised results and discussion sections.

Contrary to the conventional view of the anti-inflammatory role of resident macrophages, this manuscript highlights the immunostimulatory capacity of KC, particularly through the secretion of IL12, IL15 and IL18, which promotes effector lymphoid cell responses and anti-tumor immunity. Previous work has shown the importance of IFN γ sensing by KC for IL12 secretion (DOI: 10.1126/sciimmunol.abi7083). Discussion of these results would add to the narrative while strengthening the data presented.

Author response: We agree with the reviewer, and the very nice work by Siwicki, Pittet and collaborators ⁴ is cited in the revised discussion: lines 456-459: “Furthermore, it was shown that IFN γ stimulates production of IL-12 by KC in tumor bearing mice ⁴, suggesting that a feed-forward loop between ID3-expressing macrophages and effector lymphoid cells may contribute to the anti-tumor effect driven by macrophages.”

The results shown in our revised manuscript show that ID3-expressing tissue-resident Kupffer cells and ID3-expressing human macrophages triggers or increase the production of IL12/15/18 by macrophages in response to tumor cells *in vitro* and *in vivo*, and that the supernatant from macrophage/KC co-cultures promote IFN γ production by lymphoid cells in mouse and human, in an ID3 dependent manner. Together with the results of Siwicki and Pittet, these data are compatible with a feed-forward loop between macrophages and effector lymphoid cells that may contribute to the ID3-dependent anti-tumor effect of macrophages.

The representative images of *in vitro* uptake of KPC-1-tdTomato by Kupffer cells shown in Extended Data Figure 5 are redundant with the representative images shown in Figure 2j. Could the authors show a different field of view and representative time-lapse images from the latrunculin A condition? This would strengthen the results.

Author response: We have added a different field of view and representative time-lapse images from the latrunculin A condition in *in vitro* uptake experiment in Extended Data Figure 5.

The legends of the processed figures should be reviewed and added if missing.

Author response: We have reviewed and improved the figure legends to the best of our ability.

Referee #4

per request, I am going to focus on only the statistical aspects on the comparisons. Overall, all comparisons in the paper were clearly described. couple of comments:

1) In the Results section, it is not clear how the controls were chosen and whether they were comparable to the treated.

Author response: In all mice experiments we used 6-8 old C57BL/6 male mice, or mice from the indicated mutant strains and age-matched littermate controls, that received comparable treatment or control treatment. These controls are specified in the legends corresponding to each panel. We have also incorporated these information in the results section to the best of our ability.

2) Fig 2d, it is not clear what adj p-values were referred to.

Author response:

Fig 2c,d: These results were moved to Extended Data Figure 3(a,b). In the bar plot adj. p-values, obtained using the fgsea package in R (see Methods) refers to the x axis. The scatter plots represents significant DEG obtained by DEseq2 using Benjamini and Hochberg method for multiple testing and considered significant when adj. $p < 0.05$.

In Fig 2e, it is apparent that some of the p-values are borderline and would not be "significant" after adjusting for multiple comparisons (16 of them). For example, it's stretchy to say il18 was confirmed.

Fig 2e: (Revised Fig. 2c) Comparison was made by t-test because qPCR was performed separately for each target gene. We agree with the reviewer that the p-value for il18 transcript level in KC from control liver and tumoral liver is borderline ($p = 0.048$). We modified the text of the result section accordingly.

This does not affect the conclusions of our study because we confirmed by immunofluorescence that KC located near the tumor border express IL18 (Revised Figure 2h, $p < 0.0001$). Moreover, we show that that induction of IL18mRNA in KC is ID3 dependent (Revised Figure 4c, $p = 0.008$), and the ID3-dependent induction of IL18 protein in KC is also confirmed (Revised Figure 4h $p < 0.001$). In addition, blockade of SIRPA rescue induction of IL18 mRNA in ID3 deficient KC (Revised Figure 5b, $p = 0.001$), while Dectin-1 blockade abrogate IL18 mRNA induction in wt KC (Revised Figure 5f, $p = 0.001$, $p < 0.001$). In addition, new results shown in the revised manuscript show that expression of ID3 by human macrophages increases their IL18 production in response to tumor cells (Revised Figure 6c, $p = 0.0022$). Among other cytokines, the same remarks apply to IL12, Dectin1, CCL4 with relatively low p values in Revised Figure 2c ($p = 0.03$, $p = 0.03$, $p = 0.04$), but confirmed later in the manuscript as for IL18.

3) all comparisons were performed by either t-test or ANOVA. Both rely on the assumption of data being normally distributed. Most of the data look at least symmetric, however, there are few data types that look quite skewed, including those in Fig 1a-d.

Author response:

We thank the reviewer for this comment. In the revised manuscript, we used the Mann-Whitney U to compare the tumoral load between organs of control and macrophage/KC deficient mice when they were continuous or skewed (not normally distributed) Fig 1a-d, as well as Fig 1e and Fig 3g. This re-analysis yielded p-values that are more significant.

4) It is not clear in Methods what deep learning methodology was used and what software. The provided code was helpful, but not every reader would dig into the code to understand. So suggest adding more description in the extended data or suppl materials.

We thank the referee for their comment. The methodology used in this study was described as follows: we adapted a strategy of AgentBind⁵ and fine-tuned a pre-trained DeepSEA model⁶ using all active enhancers in Kupffer cells based on previously published ATAC-seq and H3K27ac ChIP-seq data under GEO accession GSE128338⁷. Of note, AgentBind model consists of (1) pre-training convolutional neural networks (CNNs), which infer important sequence context features and learn combinations and orientations of these features that are predictive of binding, using ChIP-sequencing and DNaseI-sequencing profiles collected from ENCODE18 and the Epigenomics Roadmap Project20 across dozens of cell types, and (2) fine-tuning an individual model for each TF to identify bound vs. unbound sequences, detailed in ref: ⁵. DeepSEA model (deep learning-based sequence analyzer), is a fully sequence-based algorithmic framework for noncoding-variant effect prediction, detailed in ref: ⁶. Software used for this methodology are Python 3, Keras 2.3.1, tensorflow 2.1.0, scikit-learn 0.21.3, deeplift 0.6.10.0, biopython 1.76. Comparable analysis approach along with this methodology and code is available at https://github.com/zeyang-shen/macrophage_IL4Response, and has been published, see ref: DOI: 10.1126/sciadv.abf9808).

As suggested, we have added an additional description of the methods in the section: ***Training and interpretation of deep learning model section***.

The modified text is underlined

“The deep learning model was trained and interpreted as described previously⁸. In brief, we adapted a strategy of AgentBind⁵ and fine-tuned a pre-trained DeepSEA model⁶ using all active enhancers in Kupffer cells based on previously published ATAC-seq and H3K27ac ChIP-seq data under GEO accession GSE128338⁷. Of note, AgentBind model consists of (1) pre-training convolutional neural networks (CNNs), which infer important sequence context features and learn combinations and orientations of these features that are predictive of binding, using ChIP-sequencing and DNaseI-sequencing profiles collected from ENCODE18 and the Epigenomics Roadmap Project20 across dozens of cell types, and (2) fine-tuning an individual model for each TF to identify bound vs. unbound sequences, detailed in ref: ⁵. DeepSEA model (deep learning-based sequence analyzer), is a fully sequence-based algorithmic framework for noncoding-variant effect prediction, detailed in ref: ⁶. Software used for this methodology are Python 3, Keras 2.3.1, tensorflow 2.1.0, scikit-learn 0.21.3, deeplift 0.6.10.0, biopython 1.76. Training data were prepared as follows. Positive data labeled as 1 were 300-bp sequences of ATAC-seq peaks associated with strong levels of H3K27ac. We first obtained the processed data file from GEO accession GSE128338, which includes the reproducible ATAC-seq peaks merged from Kupffer cells of both healthy and NASH diet mice and their tag counts of H3K27ac ChIP-seq in the expanded 2000-bp regions⁷. We removed sex chromosomes and filtered the peaks with a minimum cutoff of 32 tags of H3K27ac ChIP-seq. The positive sequences were balanced with the same number of 300-bp negative sequences, which were GC content-matched random genomic regions selected from the mm10 genome and were labeled as 0. During the training, we left out sequences on chromosome 8 for cross validation and those on chromosome 9 for testing. The final model had an area under the receiver operating characteristic curve (auROC) equal to 0.828 on the testing data. Next, we used DeepLIFT⁹ to generate importance scores with single-nucleotide resolution using uniform nucleotide backgrounds. For each input sequence, we generated two sets of scores, one for the original sequence and the other for its reverse complement. The final scores were the absolute maximum at each aligned position. We defined

predicted functional nucleotides by the top 20% (i.e., top 60) positions within each input 300-bp sequence. “

- 1 Grakoui, A. & Crispe, I. N. Presentation of hepatocellular antigens. *Cellular & Molecular Immunology* **13**, 293-300, doi:10.1038/cmi.2015.109 (2016).
- 2 Knolle, P. A. Staying local-antigen presentation in the liver. *Curr Opin Immunol* **40**, 36-42, doi:10.1016/j.coi.2016.02.009 (2016).
- 3 Mass, E. *et al.* Specification of tissue-resident macrophages during organogenesis. *Science* **353**, doi:10.1126/science.aaf4238 (2016).
- 4 Siwicki, M. *et al.* Resident Kupffer cells and neutrophils drive liver toxicity in cancer immunotherapy. *Science Immunology* **6**, eabi7083, doi:doi:10.1126/sciimmunol.abi7083 (2021).
- 5 Zheng, A. *et al.* Deep neural networks identify sequence context features predictive of transcription factor binding. *Nat Mach Intell* **3**, 172-180, doi:10.1038/s42256-020-00282-y (2021).
- 6 Zhou, J. & Troyanskaya, O. G. Predicting effects of noncoding variants with deep learning-based sequence model. *Nat Methods* **12**, 931-934, doi:10.1038/nmeth.3547 (2015).
- 7 Seidman, J. S. *et al.* Niche-Specific Reprogramming of Epigenetic Landscapes Drives Myeloid Cell Diversity in Nonalcoholic Steatohepatitis. *Immunity* **52**, 1057-1074.e1057, doi:10.1016/j.immuni.2020.04.001 (2020).
- 8 Hoeksema, M. A. *et al.* Mechanisms underlying divergent responses of genetically distinct macrophages to IL-4. *Sci Adv* **7**, doi:10.1126/sciadv.abf9808 (2021).
- 9 Shrikumar, A., Greenside, Peyton, & Kundaje, Anshul. Learning Important Features Through Propagating Activation Differences. <http://arxiv.org/abs/1704.02685> (2017).

Reviewer Reports on the Second Revision:

Referees' comments:

Referee #2 (Remarks to the Author):

I thank the authors for the additional revisions of their manuscript. The new data further strengthen the original message and indeed suggest important translational implications of their findings.

Referee #4 (Remarks to the Author):

Thank you for addressing my comments thoroughly. It's helpful to see the responses and know that the results hold with alternative analyses approaches. Congratulations on a well-done manuscript. I have no more questions.

PS. As you revise the manuscript please pay attention to the format related issues raised by our editorial assistants:

Issues:

1. Flagging that the manuscript is not in .docx format. Currently it is in pdf format.
2. The number of main text references should be 60 in total or less - currently there are 105.
3. Flagging that there are no methods references - please create a separate reference list for the methods with continuous numbering.
4. Please remove the main figures from the article file and re-supply them individually in an acceptable format such as EPS, AI, PS, PDF, PPT, CDR, PSD or XLS (for graphs)
5. Please reduce subheadings to 40 characters (with spaces) or less.
6. Flagging that the method section is duplicated between SI and main text
7. Please provide a supplementary information guide.
8. Flagging that there are potential third party rights issues in the figures - please check sources or if permissions are needed for the Mice, Liver (illustrations), petriplates, illustrations in the figures.
9. Please provide a data availability statement in the main text of the manuscript.
10. Please ensure all main figure legends are 300 words or less.
11. Figure 2, 4 are too tall in height when re-sized to 18 cm width, please reduce to 17 cm or less.
12. Please ensure that the text size in all figures is at least 5 pt Arial.